# BLACK-BOX DETECTION OF LLM-GENERATED TEXT USING GENERALIZED JENSEN-SHANNON DIVERGENCE

## ABSTRACT

We study black-box detection of machine-generated text under practical constraints: the scoring model (proxy LM) may mismatch the unknown source model, and per-input contrastive generation is costly. We propose SurpMark, a reference-based detector that summarizes a passage by the dynamics of its token surprisals. SurpMark discretizes surprisals into interpretable states, estimates a state-transition matrix for the test text, and scores it via a generalized Jensen–Shannon (GJS) gap between the test transitions and two fixed references (human vs. machine) built once from existing corpora. Theoretically, we derive design guidance for how the discretization bins should scale with data and provide a principled justification for our test statistic. Empirically, across multiple datasets, source models, and scenarios, SurpMark consistently matches or surpasses baselines; our experiments on hyperparameter sensitivity exhibit trends that our theoretical results help to explain, and are consistent with the method's underlying intuitions.

## 1 INTRODUCTION

Rapid advancements in LLMs have driven their text generation capabilities to near-human levels. This has blurred the boundary between human-written and machine-generated text, posing multiple concerns. These include susceptibility to fabrications (Ji et al. (2023)) and outdated or misleading information, which can spread misinformation, or facilitate plagiarism (Lee et al. (2023)). LLMs are also vulnerable to malicious use in disinformation dissemination (Lin et al. (2022)), fraud(Ayoobi et al. (2023)), social media spam (Mirsky et al. (2021)), and academic dishonesty (Kasneci et al. (2023)). Moreover, the increasing use of LLM-generated content in training pipelines creates a recursive feedback loop (Alemohammad et al. (2023)), potentially degrading data quality and diversity, which poses long-term risks to both society and academia. These concerns motivate the development of detectors that reliably distinguish human-written from machine-generated text and can be deployed at scale across domains.

Prior work on text detection can be grouped into two categories: classifier-based and statistics-based. Classifier-based detectors require training a task-specific model, which in turn hinges on collecting high-quality, domain-balanced labeled data (Guo et al. (2023); Tian (2023); Guo et al. (2024)); this process is costly, time-consuming, and must be repeated when the target domain or generator shifts. Statistics-based methods fall into two categories: global statistics and distributional statistics. The first relies on global statistics such as likelihood or rank (Solaiman et al. (2019); Gehrmann et al. (2019)), which can be inaccurate or unstable under calibration mismatch, text-length variability, and domain shift. The second relies on distributional statistics, which are constructed by regenerating a neighborhood around the test passage, via sampling, perturbation, or continuation, thereby tying the detector to that particular input (Yang et al. (2023); Su et al. (2023b); Mitchell et al. (2023)). Such per-instance pipelines demand substantial compute and latency and are unrealistic when resources are constrained or throughput is high. Black-box constraints exacerbate calibration drift in global-statistic and regeneration-based detectors due to proxy-model mismatch. This motivates the development of detectors that avoid retraining and per-instance regeneration while remaining reliable under distribution shift in the black-box setting.

Accordingly, we pursue a design that sidesteps both training-classifier and per-instance regeneration by focusing on stable, dynamics-aware signals, that can be reused across test samples. Viewed through a black-box perspective, the problem naturally invites a likelihood-free hypothesis testing formulation

Figure 1: SurpMark framework. Offline, we build human/machine reference transition matrices by scoring corpora with a proxy LM, discretizing surprisal via a shared $q_k$, and counting state transitions. Online, a test passage is summarized the same way and assigned a GJS score to measure proximity to human vs. machine references. Details are in Algorithm 1 and 2 in Appendix A1.

(Gutman (1989); Gerber & Polyanskiy (2024)): when the true likelihood is unknown, we compare the empirical summary statistics of a test text against human and machine references. Our summary statistic design is guided by two principles. First, because the references are existing corpora whose contexts differ from the test passage, the summary must be abstract and calibration-robust; second, decisions should exploit token dynamics which expose rich local patterns (Xu et al. (2025)). We therefore quantize token surprisal into interpretable states and summarize texts by their state-transition patterns, allowing decisions to depend on relative structure rather than absolute likelihood levels. This representation captures token dynamics and provides a stable, interpretable basis for likelihood-free comparison to human and machine references.

In this paper, we present SurpMark, a black-box, reference-based detector that frames attribution as a likelihood-free hypothesis test. For each test text, token surprisals from a proxy LM are quantized into $k$ interpretable states. The text is summarized by its state-transition matrix and is then assigned a generalized Jensen-Shannon (GJS) divergence score that measures its proximity to the human or machine reference transitions. These design choices motivate the theoretical analysis: Under an idealized first-order Markov model fitted to the discretized surprisal states, we analyze how discretization affects the estimation of GJS and study the properties of our decision statistic.

### 1.1 MAIN CONTRIBUTIONS

- We propose SurpMark, a reference-based detector that requires no per-instance regeneration, as shown in Figure 1.
- A theoretical analysis, deriving design guidance for how the discretization bins should scale with data and providing a principled justification for the proposed test statistic.
- A comprehensive experimental evaluation of SurpMark demonstrates its effectiveness across multiple models and domains.

## 2 RELATED WORK

Prior work on text detection can be broadly categorized into classifier-based and statistics-based methods. Classifier-based detectors train task-specific classifiers to distinguish between human-written and machine-generated text(Guo et al. (2023); Tian (2023); Guo et al. (2024)). While effective with sufficient training data, they are costly to build and must be retrained whenever the domain or generator shifts.

Statistics-based approaches can be divided into two groups based on their design of decision statistics. The first global-statistic methods rely on overall features of the text such as likelihood (Solaiman et al. (2019)), LogRank (Solaiman et al. (2019)) that measures the log of each token's rank in a model's predicted distribution , or entropy (Gehrmann et al. (2019)) that measures the uncertainty of a model's next-token distribution. Distributional-statistic methods generate a neighborhood around the test passage via perturbation, continuation, or sampling, and then measure divergence between the test instance and this synthetic distribution. DetectGPT (Mitchell et al. (2023)) leverages the local curvature of log-probability function, comparing original passages with perturbed variants to enable detection of machine-generated text. Fast-DetectGPT (Bao et al. (2024)) introduces conditional

probability curvature for faster detection. DNA-GPT (Yang et al. (2023)) truncates passages, and analyzes n-gram divergences of the regeneration. DetectLLM-NPR (Su et al. (2023a)) leverages normalized perturbed log-rank statistics, showing that machine-generated texts are more sensitive to small perturbations. Lastde++ (Xu et al. (2025)) combines global likelihood with local diversity entropy, where discretization of token probabilities stabilizes the entropy feature. In contrast, our framework discretizes token surprisals to build surprisal-state Markov transitions, enabling likelihood-free hypothesis test. Our method lies between global- and distributional-statistic approaches: it scores each text in a single pass without regeneration, yet makes comparative decisions by measuring alignment with fixed human and machine references.

Recent work has explored kernel-based statistical tests for machine-generated text detection (Zhang et al. (2024),Song et al. (2025)). Song et al. (2025) introduced R-Detect, a relative test framework that reduces false positives by comparing whether a test text is closer to human-written or machine-generated distributions. Our method shares a common foundation with Song et al. (2025) in that it can also be viewed as a relative test framework. Notably, while the decision rules of these kernel-based approaches are non-parametric and do not rely on supervised classifiers, their optimized variants require training kernel parameters on reference corpora, which increases computational cost. Our approach only requires an lightweight data discretization stage.

## 3 SURPMARK: DETAILED METHODOLOGY

In this section, we introduce the proposed detector SurpMark.

**Surprisal Sequence Estimation via Proxy Model.** Given a fixed text passage $\mathbf{t}$ and a proxy model $F_\theta$, we first tokenize $\mathbf{t}$ with the tokenizer associated with $F_\theta$ to obtain a token sequence $\mathbf{x} = (x_1, \ldots, x_n)$ of length $n$. We then run a single forward pass of $F_\theta$ on this fixed sequence to compute the token-level surprisal sequence $\{s_t\}_{t=1}^n$.

$$\begin{aligned}
\{s_t\}_{t=1}^n &= \{s_1, s_2, \ldots, s_n\} \\
&= \{-\log p_\theta(x_2|x_1), -\log p_\theta(x_3|\{x_t\}_{t=1}^2), \ldots, -\log p_\theta(x_n|\{x_t\}_{t=1}^{n-1})\}
\end{aligned}$$

where $p_\theta(\cdot \mid \cdot)$ is the conditional probability estimated by the proxy model $F_\theta$.

**Surprisal Discretization by K-means.** Since surprisal values from the proxy model are continuous, we discretize them into a finite set of surprisal states to enable robust statistical modeling. We employ $k$-means clustering to partition the surprisal distribution into $k$ levels, denoted as $\mathcal{A} = \{1, \ldots, k\}$. For example, when $k = |\mathcal{A}| = 4$, the clusters correspond to interpretable states such as "Predictable," "Slightly Surprising," "Significantly Surprising," and "Highly Surprising." This abstraction simplifies modeling while preserving the essential structure of predictive uncertainty.

Effectively, this step converts the initial sequence of continuous surprisal values, $\{s_t\}_{t=1}^n$, into a discrete state sequence, $\{a_t\}_{t=1}^n$, where $a_t \in \mathcal{A}$.

**Modeling State Transitions as Markov Chain.** After discretizing surprisal values into finite states, we model the resulting sequence as a Markov chain. Notably, LLMs often produce a highly predictable token after a highly surprising one, a recovery effect driven by perplexity minimization, as illustrated in Figure 2(a). To capture this structure, we summarize each text by its empirical first-order transition frequencies. Formally, given a discretized surprisal state sequence $\{a_1, a_2, \ldots, a_n\}$, we estimate a transition probability matrix $\hat{M}$, where each entry $\hat{M}(j|i)$ represents the empirical probability of transitioning from state $i$ to state $j$, with $i, j \in \mathcal{A}$.

$$\hat{M}(j|i) = \frac{\sum_{t=1}^{n-1} \mathbf{1}\{a_t = i, \ a_{t+1} = j\}}{\sum_{t=1}^{n-1} \mathbf{1}\{a_t = i\}}, \quad i, j \in \mathcal{A} \tag{1}$$

Here, $\mathbf{1}\{\cdot\}$ is the indicator function.

Figure 2(b) varies the order of the state-transition summary while keeping the reference and test sets fixed. AUROC deteriorates as the order increases, which we attribute to state-space explosion with limited data: higher-order transition counts on both the reference and test side become extremely sparse, so higher-order summaries bring no notable gains over the first-order one. More details and further empirical and theoretical justification are in Section 4.1 and Appendix A2.

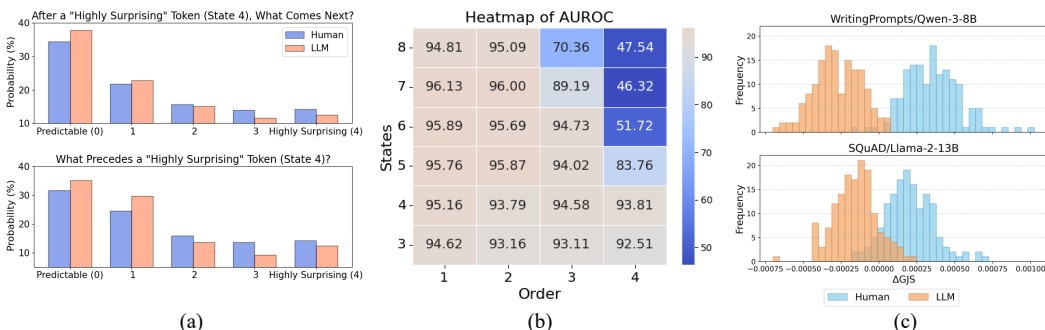

Figure 2: (a) Visualizes the key feature driving our detector by comparing the conditional probabilities of transitioning into and out of the "Highly Surprising" state under a 4-bin discretization. This reveals distinct dynamic patterns, including a stronger recovery tendency and a more pronounced spiking tendency from low-surprisal contexts in LLM-generated text. (b) A heatmap illustrating the detector's performance (AUROC) on SQuAD across different hyperparameter settings, using a fixed amount of reference and test data. Higher orders suffer from state-space explosion and sparse transitions, yielding no notable gains beyond the first-order model. (c) The final score distributions of our detector.

**Reference-based Detection with Generalized JS Divergence.** We frame the task of distinguishing between human-written and LLM-generated text as a binary likelihood-free hypothesis testing (LFHT) problem (Gutman (1989); Gerber & Polyanskiy (2024)). To adapt LFHT for this specific domain, we introduce three key methodological modifications: (1) we utilize token-level surprisals from a fixed proxy LM as observable features; (2) we employ $k$-means quantization to transform continuous values into statistically tractable discrete state sequences; and (3) we propose a novel test statistic, $\Delta\text{GJS}_n$. Crucially, unlike standard LFHT which typically evaluates divergence from a single reference distribution Gutman (1989), $\Delta\text{GJS}_n$ leverages a two-sided comparison against both human and machine references to enhance discriminative power. In this framework, the null hypothesis $H_0$ posits that the text is machine-generated, while the alternative $H_1$ suggests it is human-written. Since the true source distributions ($P$ and $Q$) are unknown, our approach remains strictly reference-based, relying on historical corpora to approximate the underlying statistics.

Specifically, given reference texts $\mathbf{t}_P, \mathbf{t}_Q$ from both model source $P$ and human source $Q$, we first compute their empirical surprisal transition probability matrices, denoted by $\hat{M}_P$ and $\hat{M}_Q$, respectively. For a given test text $\mathbf{t}$ coming from either $P$ or $Q$, we similarly compute its surprisal transition probability matrix $\hat{M}_T$ using the surprisal state levels estimated from reference texts. We then calculate two separate divergence scores using the generalized Jensen-Shannon Divergence (GJS): one measuring the distance between the test text and the machine reference model $\text{GJS}(\hat{M}_P, \hat{M}_T, \alpha)$ and another measuring the distance to the human reference model $\text{GJS}(\hat{M}_Q, \hat{M}_T, \alpha)$, where $\alpha$ denotes the reference–test length ratio. The GJS divergence between $M_A$ and $M_B$ with weight $\alpha$ is defined as

$$\text{GJS}(M_A, M_B, \alpha) = \frac{\alpha}{1+\alpha} D_{\text{KL}}(M_A, M_\alpha) + \frac{1}{1+\alpha} D_{\text{KL}}(M_B, M_\alpha), \quad M_\alpha = \frac{\alpha}{1+\alpha}M_A + \frac{1}{1+\alpha}M_B,$$

where $D_{\text{KL}}$ denotes the Kullback–Leibler divergence. We score each test passage with $\Delta\text{GJS}_n = \text{GJS}\left(\hat{M}_P, \hat{M}_T, \alpha\right) - \text{GJS}\left(\hat{M}_Q, \hat{M}_T, \alpha\right)$. We classify via a tunable threshold $\tau$.

$$\Omega = \begin{cases} H_0 & \text{if } \Delta\text{GJS}_n \leq \tau, \\ H_1 & \text{if } \Delta\text{GJS}_n > \tau \end{cases} \tag{2}$$

See Algorithm 1 and 2 in Appendix A1 for details.

## 4 ANALYSIS

This section theoretically grounds our design choices. First, we justify the first-order Markov modeling of discretized surprisals, supported by Gray's approximation theory and empirical evidence that second-order gains are negligible. Second, we justify $\Delta\text{GJS}_n$ to be a principled choice and analyze it under this idealized framework to characterize how discretization and sample size influence its behavior.

### 4.1 THEORETICAL JUSTIFICATION FOR FIRST-ORDER MARKOV MODELING

Our detector models the discretized surprisal sequence with a first-order Markov chain. Gray's Markov approximation theory (Gray, 2011) shows that, for any stationary discrete-time process, the canonical $K$-th order Markov chain is the best $K$-th order approximation in relative-entropy-rate sense. Specializing this to our discretized surprisal process and $K = 1$, the gain from moving from first order to second order is governed by the conditional mutual information term $I(a_t; a_{t-2} \mid a_{t-1})$. Empirically, we observe this term to be negligible in our data—at most 0.0076 bits/token (perplexity reduction $\approx 0.5\%$; see Appendix A2.2). This indicates that a first-order chain already captures almost all useful temporal dependence for our purposes.

### 4.2 ANALYSIS UNDER IDEALIZED FIRST-ORDER MARKOV MODELING

Let $\{s_t^P\}_{t=1}^N$ and $\{s_t^Q\}_{t=1}^N$ be the surprisal sequences produced by a fixed proxy LM on reference corpora from the model source $P$ and the human source $Q$, respectively. For the purpose of analysis, we work with a *first-order Markov approximation* to these surprisal processes on the real line $\mathbb{R}$, and write $\mathcal{S}_P$ and $\mathcal{S}_Q$ for the corresponding transition kernels. This should be viewed as a stylized model for the discretized surprisal dynamics, rather than a literal assumption that the underlying language model is exactly first-order Markov. For an integer $k \geq 2$, let $q_k : \mathbb{R} \to \mathcal{A} = \{1, \dots, k\}$ be a shared quantizer with boundaries $b_1 < \cdots < b_{k-1}$, and define discretized states $a_t^P = q_k(s_t^P)$ and $a_t^Q = q_k(s_t^Q)$. The induced $k$-state Markov chains have transition kernels

$$M_P(j \mid i) = \Pr[a_{t+1}^P = j \mid a_t^P = i], \qquad M_Q(j \mid i) = \Pr[a_{t+1}^Q = j \mid a_t^Q = i],$$

and their plug-in estimators $\hat{M}_P, \hat{M}_Q$ are formed from transition counts as in Eq. 1. We assume the discretized chains are ergodic with well-behaved mixing, which is standard in Markov-chain analyses and matches our empirical observations on the surprisal-state sequences. Finally, we observe an independent test surprisal-state sequence $a_{1:n}^T = \{a_t^T\}_{t=1}^n$ whose source $M_T$ is either $M_P$ (null $H_0$) or $M_Q$ (alternative $H_1$), and all three sequences share the same quantizer $q_k$.

#### 4.2.1 DISCRETIZATION EFFECT

How should we choose the number of bins $k$? Too few bins lose structural information, while too many, given a fixed-length reference, lead to sparse counts, higher estimation noise, and bias from zero-count corrections. Thus, $k$ must balance information preservation and statistical reliability.

Following Pillutla et al. (2023), we analyze discretization through a two-term decomposition. Discretization error is a deterministic bias from projecting the continuous object onto $k$ bins, while the statistical error is the finite-sample discrepancy when estimating the discretized object. Pillutla et al. (2023) study IID samples, and control the statistical error by splitting observed vs. unobserved mass and derive non-asymptotic bounds when balanced with their quantization error. Rather than assuming IID samples, we focus on Markov sources and examine empirical transition counts from their sequences.

For a divergence functional $\mathcal{D}_f$ (we use row-wise GJS), the empirical estimator is $\mathcal{D}_f(\hat{M}_P, \hat{M}_Q)$. Our goal is to develop a non-asymptotic bound on the absolute error of the empirical estimator relative to the true target, decomposed as

$$\underbrace{|\mathcal{D}_f(\mathcal{S}_P, \mathcal{S}_Q) - \mathcal{D}_f(M_P, M_Q)|}_{\text{discretization error}} + \underbrace{|\mathcal{D}_f(\hat{M}_P, \hat{M}_Q) - \mathcal{D}_f(M_P, M_Q)|}_{\text{statistical error}} \tag{3}$$

where $\mathcal{S}_P, \mathcal{S}_Q$ denote the underlying Markov transition kernels. For simplicity we take both references to have the same total transitions $N$. $C$ denotes an absolute constant that may change from line to line.

**Discretization Error and Statistical Error.** At a high level, we decompose the total error into a discretization bias and a finite-sample statistical term. The discretization bias is controlled by adapting Pillutla et al. (2023) to our Markov setting and yields an $O(1/k)$ bound. Theorem 4.2 then bounds the statistical error by tracking three sources—row-wise transition noise, missing-mass from unseen transitions, and an additional stationary-weight estimation error specific to Markov chains—showing how the overall error trades off $k$ and $N$.

**Proposition 4.1.** *Let $\mathcal{S}_P, \mathcal{S}_Q$ be the population first-order Markov transition kernels on the continuous surprisal space $\mathbb{R}$. Consider a shared $k$-bin quantizer $q_k : \mathbb{R} \to \mathcal{A}$ and, from it, form the discretized*

$k$-state Markov chains $M_P, M_Q$. *For any row-aggregated $f$-divergence functional $\mathcal{D}$, there exists such a shared $k$-bin partition satisfying*

$$|\mathcal{D}_f(\mathcal{S}_P, \mathcal{S}_Q) - \mathcal{D}_f(M_P, M_Q)| \leq \frac{C}{k} \tag{4}$$

See AppendixA3.2.4 for the proof.

**Theorem 4.2.** *Suppose we are in the setting described in Section 4.2. Assume each discretized chain is ergodic with bounded mixing time, $\pi_{\min} \gtrsim 1/k$, and maximum hitting time $\max\{T(M_P), T(M_Q)\} = O(1)$. It holds that*

$$|\mathcal{D}_f(\hat{M}_P, \hat{M}_Q) - \mathcal{D}_f(M_P, M_Q)| \leq C\Big( \log N \cdot \sqrt{\frac{k^3 \log(kN)}{N}} + \frac{k^3}{N} \log \big(1 + \frac{N}{k}\big) + \frac{k}{\sqrt{N}} \Big) \tag{5}$$

See Appendix A3.2.3 for the proof.

**Balancing Two Errors.** We balance $k$ by trading off the discretization bias against the finite-sample statistical error. The discretization term decays as $O(k^{-1})$, while the leading statistical term from row-wise transition estimation grows like $k^{\frac{3}{2}}/\sqrt{N}$ up to logs, with smaller contributions $O(k/\sqrt{N})$ and $O(k^3/N)$ for $k \ll N^{\frac{1}{3}}$. Neglecting logs and lower-order terms, the dominant balance is between $c_1 k^{\frac{3}{2}}/\sqrt{N}$ and $c_2/k$, yielding $k^* = \Theta(N^{\frac{1}{5}})$.

### 4.2.2 Decision Statistic Analysis

Building on this discretized Markov approximation, we next analyze the decision statistic $\Delta\text{GJS}_n$. The goal is to understand why the proposed score behaves well under this idealized model, providing intuition for the empirical results in Section 5. Our detector extends Gutman's universal hypothesis test (Gutman (1989)) from a single-reference setting to a two-reference setting. In Gutman's test, the test sequence is compared against one reference source; here we leverage two calibrated references $P$ (LM) and $Q$ (human) and decide by $\Delta\text{GJS}_n$. Our choice of GJS is not ad hoc. Algebraically, $\Delta\text{GJS}_n$ is the log–likelihood ratio (LLR) between the two hypotheses.

**$\Delta\text{GJS}_n$ as Log-Likelihood Ratio.** Proposition 4.3 shows that $\Delta\text{GJS}_n$ exactly equals the normalized log-likelihood ratio $\Lambda_{n,N}$. Here, the log-likelihood ratio represents the maximized data likelihood under the two hypotheses $H_0$ and $H_1$. See Appendix A3.3.2 for the proof.

**Proposition 4.3.** *Assume the setting of Section 4.2. Let $\mathcal{F}_k$ be the family of stationary first-order Markov models on $\mathcal{A} := [k]$. For sequences $a_{1:N}^P$, $a_{1:N}^Q$, and $a_{1:n}^T$, define the concatenations $(a_{1:N}^P, a_{1:n}^T)$ and $(a_{1:N}^Q, a_{1:n}^T)$. Consider the generalized log-likelihood ratio $\Lambda_{n,N}$*

$$\Lambda_{n,N} = \frac{1}{n} \log \frac{\sup\limits_{M,M' \in \mathcal{F}_k} M\big((a_{1:N}^P, a_{1:n}^T)\big)\, M'(a_{1:N}^Q)}{\sup\limits_{M,M' \in \mathcal{F}_k} M(a_{1:N}^P)\, M'\big((a_{1:N}^Q, a_{1:n}^T)\big)} \tag{6}$$

*where the suprema are attained at the empirical Markov models on the respective concatenated sequences. Then, $\Delta\text{GJS}_n = \Lambda_{n,N}$.*

In Appendix A3.3.3, we further prove the asymptotic normality of our statistics $\Delta\text{GJS}_n$, and empirically verify it through experiments in Appendix A4.2.3.

## 5 Experiments

**Datasets, Configurations and Models.** We evaluate our method on XSum (Narayan et al. (2018)), WritingPrompts (Fan et al. (2018)), SQuAD (Rajpurkar et al. (2016)), WMT19 (Barrault et al. (2019)), and HC3 (Guo et al. (2023)). Unless otherwise noted, we construct the reference corpora and test set as follows. For each dataset, we randomly sample 300 human-written texts to form the human reference, then generate paired machine outputs by prompting the source model with the first 30 tokens of each human text. For the test set, we sample another 150 human-written texts and create their machine-generated counterparts using the same procedure. We select 9 open-source models and 3 closed-source models as our source model. More details are in Appendix A4.1. Unless otherwise specified, we use GPT2-Large as our proxy model.

**Baselines.** We benchmark against 13 detectors, including Likelihood (Solaiman et al. (2019)), LogRank (Solaiman et al. (2019)), Entropy (Gehrmann et al. (2019); Ippolito et al. (2020)), DetectLRR (Su et al. (2023a)), and Lastde (Xu et al. (2025)), DetectGPT (Mitchell et al. (2023)), Fast-DetectGPT (Bao et al. (2024)), DNA-GPT (Yang et al. (2023)), DetectNPR (Su et al. (2023a)), Lastde++ (Xu et al. (2025)), R-Detect (Song et al. (2025)), Binoculars Hans et al. (2024), and FourierGPT Xu et al. (2024).

| | GPT2-XL | GPT-J-6B | GPT-Neo-2.7B | GPT-NeoX-20B | OPT-2.7B | Llama-2-13B | Llama-3-8B | Llama-3.2-3B | Gemma-7B | Avg |
|---|---|---|---|---|---|---|---|---|---|---|
| Likelihood | 85.02 | 74.82 | 73.32 | 72.03 | 77.22 | 94.39 | 93.93 | 65.22 | 65.8 | 77.97 |
| LogRank | 88.2 | 79.25 | 78.29 | 75.37 | 81.99 | 95.9 | 95.05 | 71.04 | 69.18 | 81.59 |
| Entropy | 51.1 | 47.15 | 50.94 | 45.94 | 48.88 | 29.03 | 29.31 | 53 | 46.85 | 44.69 |
| DetectLRR | 91.07 | 85.81 | 87.12 | 80.27 | 88.48 | 96.43 | 94.85 | 81.54 | 75.5 | 86.79 |
| Lastde | 95.97 | 85.88 | 89.09 | 80.16 | 88.89 | 93.29 | 94.29 | 72.99 | 69.48 | 85.56 |
| Lastde++ | **99.46** | 91.54 | 94.29 | 85.13 | 94.15 | 95.5 | 95.9 | 77.47 | 76.9 | 90.04 |
| DNA-GPT | 81.98 | 70.68 | 72.69 | 70.42 | 73.86 | 95.91 | 96.54 | 64.79 | 65.32 | 76.91 |
| Fast-DetectGPT | 97.94 | 86.83 | 89.15 | 83.17 | 90.55 | **98.21** | **97.98** | 74.32 | 73.95 | 88.01 |
| DetectGPT | 94.45 | 79.55 | 84.71 | 75.71 | 82.88 | 86.51 | 86.28 | 64.23 | 69.05 | 80.37 |
| DetectNPR | 94.93 | 81.91 | 86.4 | 77.93 | 84.06 | 95.19 | 93.67 | 69.45 | 71.49 | 83.89 |
| R-Detect | 74.38 | 63.58 | 67.7 | 63.35 | 65.83 | 79.98 | 81.64 | 62.22 | 55.46 | 68.24 |
| Binoculars | 99.19 | 85.76 | 87.5 | 82.02 | 86.9 | 96.93 | 96.41 | 68.36 | 73.57 | 86.19 |
| FourierGPT | 54.72 | 54.28 | 56.5 | 56.51 | 52.47 | 72.43 | 72.06 | 54.83 | 55.84 | 59.07 |
| SurpMark$_{k=6}$ | 98.07 | 92.96 | 95.19 | **86.78** | 94.49 | 97.41 | 97.06 | **81.74** | **77.40** | **91.23** |
| SurpMark$_{k=7}$ | 98.35 | **93.1** | **95.42** | 86.40 | **94.88** | 97.58 | 97.17 | 80.74 | 76.89 | 91.17 |

Table 2: Detection results for text generated by 9 open-source models under the black-box setting. The AUROC reported for each model are averaged across three datasets: Xsum, WritingPrompts, and SQuAD. See Table 12, 13, 14 in Appendix for details.

## 5.1 MAIN RESULTS

Table 1 and 2 present the detection results under black-box scenario. Table 1 shows that SurpMark achieves the best performance on 3 commercial, closed-source LLM. Performance is especially strong on GPT-5-Chat. Table 2 shows that SurpMark ranks first on 6 of 9 open-source models and within the top two on 7 of 9. These results highlight SurpMark's robustness on proprietary systems and its suitability for real-world commercial deployments. Please note that compared with distribution-based detectors that generate a neighborhood per input at test time, SurpMark builds reference corpora once and reuses them for all test passages. Under a reference-per-test budget $B = \frac{\#\text{references}}{\#\text{tests}}$, in Table 1 and 2, SurpMark operates at

| | Gemini-1.5-Flash | GPT-4.1-mini | GPT-5-Chat | Avg |
|---|---|---|---|---|
| Likelihood | 56.49 | 66.77 | 49.62 | 57.63 |
| LogRank | 53.87 | 66.8 | 49.83 | 46.53 |
| Entropy | 58.36 | 38.72 | 46.99 | 48.02 |
| DetectLRR | 44.51 | 63.29 | 49.83 | 62.11 |
| Lastde | 48.13 | 57.28 | 41.96 | 49.12 |
| Lastde++ | 71.72 | 68.23 | 43.51 | 61.15 |
| DNA-GPT | 62.06 | 56.71 | 49.82 | 56.2 |
| Fast-DetectGPT | 72.49 | 68.32 | 43.39 | 61.4 |
| DetectGPT | 69.19 | 70.08 | 54.6 | 64.75 |
| DetectNPR | 64.96 | 70.83 | 54.99 | 63.59 |
| R-Detect | 69.25 | 71.64 | 67.75 | 69.55 |
| Binoculars | 74.51 | 71.12 | 49.65 | 65.09 |
| FourierGPT | 61.25 | 63.05 | 64.82 | 63.04 |
| SurpMark$_{k=6}$ | 74.57 | **80.25** | 78.33 | 77.72 |
| SurpMark$_{k=7}$ | **75.14** | 78.48 | **81.33** | **78.32** |

Table 1: Detection results for text generated by 3 closed-source models under the black-box setting. The AUROC reported for each model are averaged across three datasets: Xsum, WritingPrompts, and SQuAD. See Table 9, 10, 11 in Appendix for details.

$B = 2$, whereas DNA-GPT uses $B = 10$, DetectGPT, DetectNPR require $B = 100$. Thus SurpMark's reference cost is $5\times$–$50\times$ lower, while avoiding any per-input contrastive generation at test time, enabling real-time detection as discussed later.

We attribute SurpMark's superior performance on closed-source models (e.g., GPT-5-chat) to its ability to capture transitional dynamics. As detailed in Appendix A4.2.12, stronger models exhibit a vanishingly small gap in marginal surprisal distributions compared to humans, rendering marginal statistics ineffective. However, the 'transition gap' remains significant, which SurpMark effectively exploits.

## 5.2 ABLATION AND SENSITIVITY ANALYSIS

**Effect of bins $k$.** Figure 3 shows the effect of the number of bins $k$. Across both models, increasing the number of bins $k$ leads to clear improvements in AUROC up to a moderate range, after which the gains saturate or slightly decline. The best results across datasets are generally observed at $k = 6 - 7$. Our theory yields an optimal bin count of the form $k^* = CN^{1/5}$ for some constant $C$, where $N$ is

the total number of transitions of the reference samples. We empirically calibrate the constant $C$ in Appendix A4.2.2. Next, we further investigate how varying $N$ shift the empirical optimum $k^*$.

**Effect of Number of Reference Samples.**
Figure 4 (a) shows that AUROC improves sharply as the reference grows from very small number of reference samples to 100 reference samples; beyond 100 reference samples the gains are minor. The $k$-optimized curve picks the best $k \in \{4, \ldots, 12\}$ at each number of reference. The annotated $k$ values grow mildly with the number of reference samples, and using large $k$ for small number of reference hurts performance. This trend aligns with our theoretical intuition: a larger number of reference samples reduces reference-side estimation error and thus allows for a slightly larger $k$.

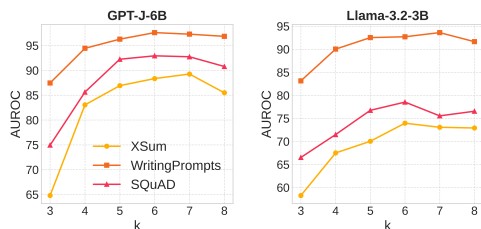

Figure 3: Effect of the number of bins $k$ on detection performance for source models including GPT-J-6B (left) and Llama-3.2-3B (right).

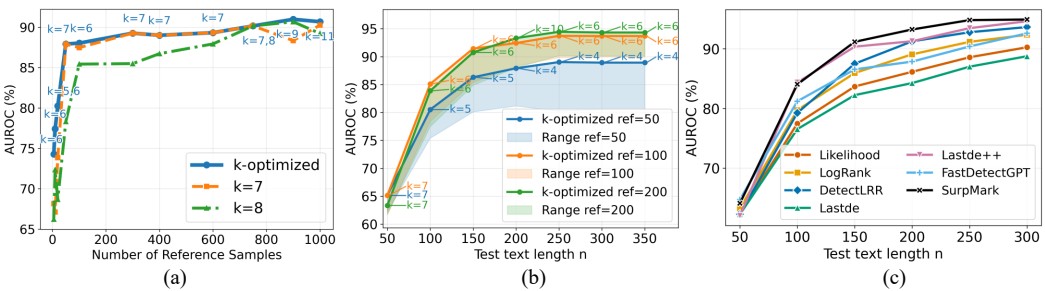

Figure 4: (a) AUROC vs. number of reference samples. The blue curve ("$k$-optimized") picks the best $k$ at each number of reference. orange/green curves fix $k \in \{7, 8\}$. (b) AUROC vs. test length $n$ under different reference lengths. Solid lines are k-optimized for each reference sample truncated to 50/100/200 tokens; shaded bands show the attainable range across $k$ at each $n$. (c) Detection results of 7 detection methods on 6 test lengths.

**Effect of Length of Test Sample.** In Figure 4 (b), we fix the number of reference samples and study the effect of sample length. AUROC climbs rapidly as test length $n$ grows from 50 to about 150–200. Longer reference lift the curves and make the bands across $k \in \{4, \ldots, 12\}$ tighter, indicating greater stability. The $k$-optimized curves show that the optimal $k$ is driven more by reference length than by test length. In Figure 4 (c), we evaluated detection performance of baselines across varying test length (tokens), focusing on WritingPrompts generated by Gemma-7B. All methods improve with longer texts. SurpMark is competitive at short lengths and becomes the top method for test length larger than 150. Comparison on more source models are presented in Figure 8 in Appendix.

**Reference–Test Length Trade-Offs.** Figure 5 (a) and (b) show AUROC contours over reference length and test length $n$ at fixed bins $k$. Performance improves toward the upper-right, and the up-right tilt shows a reference-test length trade-off: larger reference length can compensate for smaller test length at similar accuracy.

**Effect of Proxy Model.** In Figure 5 (c), x-axis lists the proxy LM used to compute scores. Across both datasets, most baselines improve with stronger proxy models, especially on WritingPrompts with GPT-5-Chat as the source model. SurpMark is consistently top and stable across proxy models. It already performs strongly with the smallest proxy and improves only modestly with larger ones, whereas several baselines are highly sensitive to the proxy choice, some even degrade when the proxy changes. In short, SurpMark achieves strong and reliable performance without expensive proxy models, making it a better default in low-resource deployments.

**Throughput.** Figure 6 (Left) plots throughput (items/s) against the number of test texts. Baseline methods appear as horizontal lines because their per-item latency is constant. SurpMark improves monotonically as the one-time preprocessing cost is amortized. The curve crosses the Fast-DetectGPT line at roughly $n \approx 298$, after which SurpMark maintains higher throughput.

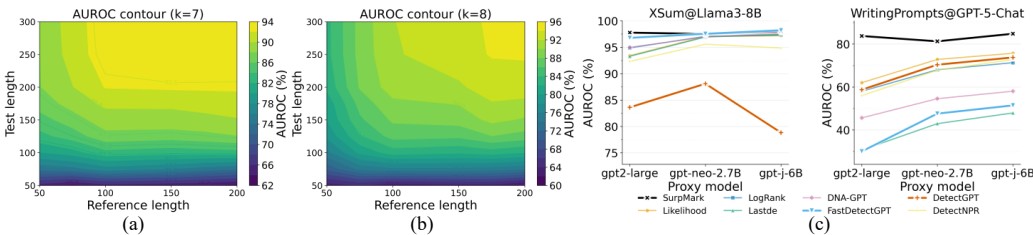

(a)  (b)  (c)

Figure 5: (a-b) AUROC contour maps (WritingPrompts/Gemma-7B). Left: $k = 7$; right: $k = 8$. The x-axis is reference length (tokens) and the y-axis is test length (tokens). Colors encode AUROC. In both panels, contours tilt up-right, indicating a trade-off: larger reference length allows smaller test length at similar performance. (c) AUROC vs. proxy model.

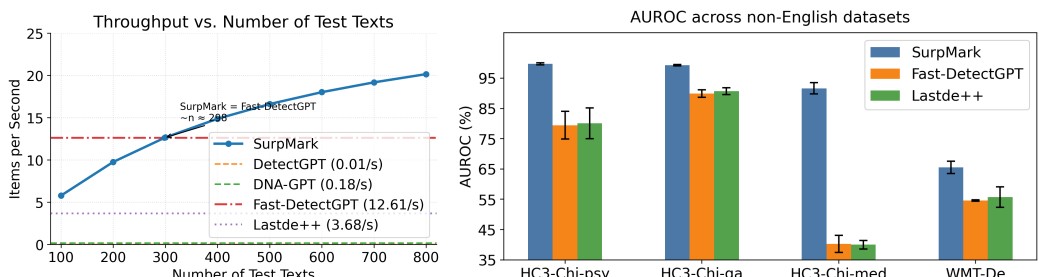

Figure 6: Left: Throughput (items per second) versus the number of test texts for SurpMark compared to baseline methods (proxy LM: GPT-2 Large; GPU: NVIDIA RTX 4090). Right: AUROC on non-English datasets (HC3-Chi-psy/qa/med and WMT-De). Error bars denote standard deviation. Higher is better.

**Non-English Scenarios.** In Figure 6 (Right), we evaluate on German and Chinese corpora. For German, we use WMT19 with GPT-4o-mini as the source model and Llama-3.2-1B as the proxy model. For Chinese, we use HC3 across multiple domains (psychology, medicine, openqa), which provides paired human and ChatGPT answers to the same questions, and adopt Qwen-2.0-0.5B as the proxy model. SurpMark ranks first on all four datasets, with large margins on HC3-Chi-med.

**Necessity of $k$-means.** In Table 3, we evaluate the effect of different discretization schemes, including $k$-means, equal-width, and equal-mass binning. Across all datasets and $k$ values, $k$-means is the most robust quantization scheme: it consistently reaches or matches the best AUROC, while equal-mass can degrade sharply on XSum@GPT-4.1-mini and equal-width is unstable and often much worse.

**Necessity of GJS.** In Table 4, we evaluate the effect of different distance metrics including GJS divergence, $L_1$ and $L_2$ norm distance. GJS achieves the best AUROC on most dataset and source model. This suggests that GJS is a more robust similarity measure than $L_1$ and $L_2$.

### 5.3 GENERALIZATION

**Cross-Domain Generalization.** We evaluated cross-domain generalization og detector in Table 5. We compared self-ref (in-domain reference) to Out-Of-Domain (OOD) reference (associate with the corresponding generator). Even when we deliberately use out-of-domain reference corpora to estimate transition probabilities, the impact is small. Changes remain moderate and can even be positive sometimes. See Table 19 for more comparison.

**Cross-Generator Generalization.** To evaluate robustness without recomputing transitions per model, we propose SurpMark-MC, a multi-class variant using a single shared quantizer (derived from pooled corpora) and distinct Markov chains for human and each reference generator. We classify test texts by assigning them to the source with the minimal GJS divergence (i.e., a nearest-neighbor rule). In a multi-class setting with GPT-J-6B, GPT-4.1-mini, and LLaMA2-13B, SurpMark-MC achieves 82.3% overall accuracy, with 78.7% accuracy for human texts, 78.0% for GPT-J-6B, 84.0% for GPT-4.1-mini, and 96.0% for LLaMA2-13B. We further collapse the multi-generator setting into

| Dataset@Source model | Method | $k = 7$ | $k = 8$ | Best |
|---|---|---|---|---|
| XSum@GPT-4.1-mini | k-means | 80.42 | 79.32 | **81.79** |
| | equal-width | 80.41 | 76.20 | 80.41 |
| | equal-mass | 71.40 | 74.22 | 74.22 |
| WritingPrompts@GPT5-chat | k-means | 82.05 | 84.86 | **84.86** |
| | equal-width | 62.90 | 72.06 | 72.06 |
| | equal-mass | 84.42 | 83.59 | 84.42 |

Table 3: AUROC of different discretization schemes under varying number of states $k$.

| | | GPT-4.1 mini | |
|---|---|---|---|
| | XSum | WritingPrompts | SQuAD |
| GJS | 82.52 | 83.64 | 69.27 |
| $L_1$ | 73.51 | 82.17 | 62.28 |
| $L_2$ | 73.58 | 83.04 | 59.14 |

Table 4: Comparison of different distance metrics across datasets.

| Test | self-ref | WritingPrompts-as-ref | XSum-as-ref | SQuAD-as-ref |
|---|---|---|---|---|
| XSum@Llama2-13b | 97.09 | 97.22 | – | 96.45 |
| WritingPrompts@Llama2-13b | 99.53 | – | 99.71 | 99.33 |
| SQuAD@Llama2-13b | 96.13 | 94.36 | 95.67 | – |
| XSum@Llama3-8b | 97.09 | 97.25 | – | 95.13 |
| WritingPrompts@Llama3-8b | 99.86 | – | 99.95 | 99.88 |
| SQuAD@Llama3-8b | 94.17 | 93.77 | 93.90 | – |

Table 5: AUROC of SurpMark under different reference choices across datasets and models.

a binary classification task (LM-generated vs. human-written), defining the score as the difference between the GJS to the human reference and the minimum GJS to any machine reference. As shown in Table 6, this approach generalizes effectively even to unseen models (e.g., Llama-3-8B, GPT-5-chat, GPT-neo-20B), matching or surpassing baselines like Lastde++ and Fast-DetectGPT.

| WritingPrompts | LLaMA2-13B | GPT-4.1-mini | GPT-J-6B | LLaMA3-8B | GPT-5-chat | GPT-neo-20B |
|---|---|---|---|---|---|---|
| Lastde++ | 99.14 | 68.49 | **95.96** | 99.56 | 30.64 | 92.68 |
| Fast-DetectGPT | 99.56 | 70.23 | 93.80 | 99.84 | 30.01 | 92.22 |
| SurpMark self-ref | 99.59 | 87.27 | 96.85 | 99.87 | 83.56 | 93.93 |
| SurpMark-MC | **99.72** | **90.01** | 95.61 | **99.77** | 64.27 | 92.84 |

Table 6: AUROC on WritingPrompts when the reference transition probabilities (LLaMA2-13B, GPT-4.1-mini and GPT-J-6B) are held fixed and not re-estimated for each test-time generator

**More Results.** We provide additional experimental results in the Appendix, including: (1) ablation study on decoding strategies (Appendix A4.2.8) (2) evaluations under paraphrasing attack (Appendix A4.2.9) (3) evaluations under prompt-engineered adversarial attacks (Appendix A4.2.10) (4) ablation on the necessity of first-order markov chain (Appendix A4.2.11) (5) discussion for threshold $\tau$ selection (Appendix A4.2.13).

# 6 CONCLUSION

We presented SurpMark, a reference-based detector for black-box detection of machine-generated text. By quantizing token surprisals into interpretable states and modeling their dynamics as a Markov chain, SurpMark reduces each passage to a transition matrix and scores it via a GJS score against fixed human/machine references. It avoids per-instance regeneration and enabling fast, scalable deployment. Our analysis establishes a principled discretization criterion and proves asymptotic normality of the decision statistic. Empirically, across diverse datasets, source models, and scenarios, SurpMark consistently matches or surpasses strong baselines.

## ETHICS STATEMENT

This work focuses on developing methods for the detection of large language model (LLM)-generated text. Our aim is to enhance transparency and accountability in AI systems rather than to enable misuse. All datasets used in this study are publicly available benchmark corpora, and no personally identifiable or sensitive information was included. we consider our framework as a tool for improving the responsible development and governance of generative AI.

## REPRODUCIBILITY STATEMENT

All experimental projects in this paper are reproducible. The details of experiments are in Section 5 and Appendix A4.1

## LLM USAGE

Large Language Models (LLMs) were employed solely for paraphrasing and minor language polishing. They were not used for idea generation, proof writing, data analysis, or experiment design. All technical contributions, theoretical results, and empirical evaluations in this paper are original and independently produced by the authors.

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

APPENDIX

## A1 ALGORITHM

---

**Algorithm 1** SurpMark (Offline): Build Human/Machine Reference Transitions

---

**Require:** Proxy LM $F_\theta$; human corpus $\mathcal{D}_Q$; machine/LLM corpus $\mathcal{D}_P$; number of bins $k$

**Ensure:** Shared surprisal quantizer $q_k$; reference transition matrices $\hat{M}_P, \hat{M}_Q$; total reference length $N$

1: **Score references.** For every $t \in \mathcal{D}_Q \cup \mathcal{D}_P$, run $F_\theta$ to obtain token sequence $x_{1:N}$ and surprisals $s_{1:N}$ with

$$s_t = -\log p_\theta(x_t \mid x_{1:t-1}).$$

2: **Fit shared quantizer.** Pool all reference surprisals and fit $k$-means to obtain $q_k : \mathbb{R} \to \{1, \ldots, k\}$.

3: **Discretize to states.** Map each reference sequence to the corresponding state sequence $a_t = q_k(s_t), t \in \{1, \ldots, N\}$.

4: **Estimate transitions.** For each corpus $C \in \{P, Q\}$, estimate the empirical first-order transition matrix $\hat{M}_C$ by counts:

$$\hat{M}_C(j \mid i) = \frac{\sum_{t=1}^{n-1} \mathbf{1}\{a_t = i, \, a_{t+1} = j\}}{\sum_{t=1}^{n-1} \mathbf{1}\{a_t = i\}}, \quad i, j \in \{1, \ldots, k\}.$$

5: **Record length.** Let $N$ be the total number of *reference* transitions used to form $\hat{M}_P$ and $\hat{M}_Q$ (sum over sequences).

6: **return** $q_k, \; \hat{M}_P, \; \hat{M}_Q, \; N$.

---

**Algorithm 2** SurpMark (Online): Decision via GJS score against References

---

**Require:** Proxy LM $F_\theta$; test text $t$; shared quantizer $q_k$; reference transitions $\hat{M}_P, \hat{M}_Q$; reference length $N$

**Ensure:** Score $\Delta\text{GJS}_n$ and label $\Omega \in \{\text{MACHINE}, \text{HUMAN}\}$

1: **Score test text.** Run $F_\theta$ on $t$ to get tokens $x_{1:n}$ and surprisals $s_{1:n}$.

2: **Discretize.** Map to surprisal states $a_t = q_k(s_t), t \in \{1, \ldots, n\}$ and estimate the test transition matrix $\hat{M}_T$ using the same formula as Offline.

3: **Set mixing weight.** $\alpha \leftarrow N/n$.

4: **Compute divergence.**

$$\Delta\text{GJS}_n \; = \; \text{GJS}(\hat{M}_P, \hat{M}_T, \alpha) \; - \; \text{GJS}(\hat{M}_Q, \hat{M}_T, \alpha).$$

5: **Decision rule.**

$$\Omega \; = \; \begin{cases} \text{MACHINE}, & \Delta\text{GJS}_n \leq \tau, \\ \text{HUMAN}, & \Delta\text{GJS}_n > \tau. \end{cases}$$

6: **return** $\Delta\text{GJS}_n, \; \Omega$.

---

## A2 JUSTIFICATION FOR FIRST-ORDER MODELING

### A2.1 EMPIRICAL FINDINGS

All AUROC values in Figure 2(b) are computed using exactly the same amount of reference data and test data. As in our main experiments, we use 300 human paragraphs and 300 machine-generated paragraphs, each with length about 100-200 tokens as reference data. We use 150 human paragraphs and 150 machine paragraphs as test data. Intuitively, increasing the Markov order makes the state space explode while the amount of reference data is fixed, so transition estimates become very sparse and noisy.

The degradation with larger order is a sparsity effect that arises both on the reference side and on the test side: (i) state-space explosion: A first-order chain with $k$ bins has $k^2$ transitions; an order-$v$ chain effectively has $k^{v+1}$ transitions. With only 300 human + 300 machine paragraphs of 100-200

| Order | Reference size | | | |
|---|---|---|---|---|
| | 300 | 600 | 900 | 1200 |
| 1 | 86.51 | 86.17 | 86.69 | 86.89 |
| 2 | 81.49 | 82.08 | 82.77 | 82.80 |
| 3 | 74.98 | 74.74 | 75.84 | 75.85 |
| 4 | 64.72 | 69.51 | 71.20 | 73.00 |

(a) AUROC vs. reference size

| Order | Test length | | | |
|---|---|---|---|---|
| | 150 | 200 | 250 | 300 |
| 1 | 90.59 | 92.66 | 94.60 | 94.58 |
| 2 | 89.42 | 90.67 | 92.21 | 92.60 |
| 3 | 88.67 | 90.57 | 91.40 | 91.40 |
| 4 | 62.29 | 65.59 | 68.46 | 68.33 |

(b) AUROC vs. test length

Table 7: Effect of reference size and test length on AUROC for different Markov orders.

tokens, many high-order contexts in the reference data are observed only a few times or not at all, so the estimated transitions become extremely noisy. (ii) Short test sequences. Each test paragraph is itself only 100-200 tokens long. Even if the reference transitions were perfectly estimated, an order-$v$ model on a 100-200-token sequence can observe only a very small number of distinct $v$-length contexts. The higher-order model is severely under-sampled on each individual test example.

To isolate the effect of reference sparsity, in the XSum@GPT-J-6B dataset, we fixed $k = 6$ and the test set (150 human + 150 machine paragraphs, 100-200 tokens each), and increased the reference size from 300 to 1200 paragraphs per side. As shown in Table 7(a), AUROC for higher-order models improves only slightly and remains clearly below the first-order model.

To further evaluate the effect of test sparsity, in another WritingPrompts@Genmma-7B dataset, we vary the test passage length from 150 to 300 tokens while keeping the reference size fixed (300 passages per side, each with fixed 300 tokens). As in Table 7(b), AUROC consistently increases for all orders, but the first-order model remains clearly best, and higher orders still lag behind by several points.

Taken together, these ablations reflect practical text-detection settings with limited reference data and short passages. In this regime, the first-order model offers the best bias-variance tradeoff, so we believe it is the most reasonable default choice.

### A2.2 THEORETICAL JUSTIFICATION

We clarify our reasoning by (i) starting from Gray's Markov approximation theory, (ii) explaining how the gain from order $K$ to $K + 1$ is governed by conditional mutual information, (iii) mapping this theory to our discretized surprisal process, and (iv) presenting empirical measurements showing that the additional benefit of a second-order approximation over a first-order one is very small.

(1) Best finite-order Markov approximation in Gray's theory.

Following Gray's Entropy and Information Theory [Gray (2011), Sec. 6.4, Cor. 6.4.1–6.4.2; Sec. 7.4, Cor. 7.4.2–7.4.3], consider a stationary discrete-time source $\{X_n\}$. Gray constructs, for each order $K$, a canonical $K$-th order Markov chain $M_K$ whose conditional distributions match those of the source given the last $K$ symbols. He shows that $M_K$ is optimal in the sense that it uniquely minimizes the relative entropy rate between the true source and any $K$-th order Markov chain on the same alphabet. In other words, the family of finite-order Markov chains $\{M_K\}$ provides a sequence of best approximations to the stationary process in the relative-entropy-rate sense. Formally,

$$H_{p\|p^K}(\{X_n\}) = \inf_{M_K \in \mathcal{M}_K} H_{p\|M_K}(\{X_n\}) = I(X_0; X_{-\infty}^{-K-1} | X_{-K}^{-1}) \tag{7}$$

where $p$ is the true stationary source, $p^K$ is the canonical $K$-th order Markov approximation to $p$, $\mathcal{M}_K$ is the class of stationary $K$-th order Markov sources on the same alphabet, $H_{p\|q}(\{X_n\})$ is the relative entropy rate of $p$ with respect to $q$, $X_{-\infty}^{-K-1} = (X_{-\infty} \ldots, X_{-K-1})$ is the infinite past, $X_{-K}^{-1} = (X_{-K}, \ldots, X_{-1})$ is the block of the last $K$ symbols, and $I(\cdot; \cdot|\cdot)$ conditional mutual information. Applying the above indentity with $K + 1$ instead of $K$, we get

$$H_{p\|p^{K+1}}(\{X_n\}) = I(X_0; X_{-\infty}^{-K-2} | X_{-K-1}^{-1}) \tag{8}$$

We are interested in the gain from going from order $K$ to $K+1$, so

$$\Delta_K = H_{p\|p^K}(\{X_n\}) - H_{p\|p^{K+1}}(\{X_n\}) \tag{9}$$

$$= I(X_0; X_{-\infty}^{-K-1}|X_{-K}^{-1}) - I(X_0; X_{-\infty}^{-K-2}|X_{-K-1}^{-1}) \tag{10}$$

$$= I(X_0; X_{-K-1}|X_{-K}^{-1}) \tag{11}$$

(2) Mapping to our discretized surprisal process

In our setting, $\{X_n\}$ is instantiated by the discretized surprisal process $\{a_t\}$, where $a_t$ corresponds to $X_0$, $X_{-K}^{-1}$ corresponds to $a_{t-K}^{t-1} == (a_{t-K}, \cdots, a_{t-1})$, $X_{-K-1}$ corresponds to $a_{t-K-1}$. For the case $K = 1$, the gain from first-order to second order is precisely $I(a_t; a_{t-2}|a_{t-1})$. We directly estimate the relevant conditional mutual information term on our data. We first fit a first-order $\hat{P}_1(a_t|a_{t-1})$ and a second-order model $\hat{P}_2(a_t|a_{t-1}, a_{t-2})$ from transition counts on the reference set. We then compute plug-in estimates on test set

$$\hat{H}_1 = -\frac{1}{n-1}\sum_{t=2}^{n} \log_2 \hat{P}_1(a_t|a_{t-1}) \tag{12}$$

$$\hat{H}_2 = -\frac{1}{n-2}\sum_{t=3}^{n} \log_2 \hat{P}_2(a_t|a_{t-1}, a_{t-2}) \tag{13}$$

Their difference is the plug-in estimate of the conditional mutual information $\hat{I} = \hat{H}_1 - \hat{H}_2 = I(a_t; a_{t-2}|a_{t-1})$, in bits per token, i.e., the extra predictive information contributed by the second-order context beyond the immediate past. On our data, we obtain empirical estimates of conditional mutual information and perplexity in Table 8. In our experiments, $\hat{I}$ is at most 0.0076 bits/token, which corresponds to a perplexity reduction around 0.5%. Thus, in terms of average predictive performance, the second-order Markov model brings only a sub-percent improvement over the first-order model. Combined with Gray's best Markov approximation theory, this indicates that a first-order Markov chain already captures most useful temporal dependence in the discretized surprisal dynamics, and provides a theoretically justified and empirically sufficient model for the discretized surprisal dynamics in our detector.

| Source | Order pair | $\hat{H}_k$ (bits/token) | $\hat{H}_1 - \hat{H}_2$ (bits/token) | Perplexity | Rel. PP change vs. 1st |
|---|---|---|---|---|---|
| GPT-5-chat | 1st (baseline) | 2.7882 | 0.0000 | 6.9075 | 0.000% |
| GPT-5-chat | 2nd order | 2.7805 | 0.0076 | 6.8711 | +0.528% |
| Human | 1st (baseline) | 2.8089 | 0.0000 | 7.0074 | 0.000% |
| Human | 2nd order | 2.8043 | 0.0045 | 6.9854 | +0.314% |

Table 8: Conditional entropies and perplexities for discretized surprisal states.

## A3 THEORETICAL ANALYSIS

### A3.1 PROBLEM SETUP

Let $\{s_t^P\}_{t=1}^{N}$ and $\{s_t^Q\}_{t=1}^{N}$ be the surprisal sequences produced by a fixed proxy LM on reference corpora from $P$ and $Q$. Each sequence is modeled as an ergodic first-order Markov process on $\mathbb{R}$. For an integer $k \geq 2$, let $q_k : \mathbb{R} \to \mathcal{A} = \{1, \ldots, k\}$ be a shared quantizer with boundaries $b_1 < \cdots < b_{k-1}$, and discretized states $a_t^P = q_k(s_t^P)$ and $a_t^Q = q_k(s_t^Q)$. Let $\mathcal{S}_P, \mathcal{S}_Q$ denote the underlying Markov transition kernels on the real-valued surprisal sequences before discretization. The induced transition kernels on the $k$-state alphabet are $M_P(j|i) = \Pr[a_{t+1}^P = j|a_t^P = i]$ and likewise $M_Q$. Their plug-in estimators $\hat{M}_P, \hat{M}_Q$ are formed from transition counts with $\hat{M}_P(a|s) = \frac{N_P(s,a)}{N_P(s)}$, where $N_P(s)$ is the number of occurrences of state $s$ in $a_{1:N}^P$, and $N_P(s, a)$ is the number of times $s$ is followed by $a$; analogously for $Q$. Let $\pi_P, \pi_Q$ are stationary distributions of $M_P$ and $M_Q$, we define $\pi_{\min} := \min\{\min_{s\in\mathcal{A}} \pi_P(s), \min_{s\in\mathcal{A}} \pi_Q(s)\}$.

We observe an independent test surprisal-state sequence $a_{1:N}^T := \{a_t^T\}_{t=1}^{n} \sim M_T$, where the test source $M_T$ is either $M_P$ (null $H_0$) or $M_Q$ (alternative $H_1$). All three sequences are discretized by the same $q_k$.

Throughout the analysis we impose the following conditions on the induced chains $M_P$ and $M_Q$. These assumptions are standard in the study of Markov concentration inequalities and are required in order to apply the auxiliary results recalled below.

**Assumption A3.1.** We impose the following standing conditions on the induced chains $M_P, M_Q$. $M_P$ and $M_Q$ are irreducible, aperiodic Markov chain on the finite alphabet $\mathcal{A}$ with unique stationary distribution $\pi_P$ and $\pi_Q$ and maximum hitting time $T(M_P)$ and $T(M_Q)$ respectively. We assume $\pi_{\min} := \min\{\min_{s \in \mathcal{A}} \pi_P(s), \min_{s \in \mathcal{A}} \pi_Q(s)\} \gtrsim 1/k$, and $T(M_\bullet) = O(1)$.

### A3.2 DISCRETIZATION EFFECT

#### A3.2.1 AUXILIARY RESULTS FROM LITERATURE

**The GJS Divergence as $f$-divergence.** The GJS divergence is a specific instance of a broader class of divergences known as $f$-divergences. An $f$-divergence between two discrete probability distributions $p$ and $q$ is defined by a convex generator function $f$ where $f(1) = 0$. The GJS divergence is equivalent to the $w$-skew Jensen-Shannon Divergence with $w = \alpha/(1 + \alpha)$, which is an $f$-divergence generated by the function $f_{JS}^w(t)$.

$$f_{JS}^w(t) = \alpha t \log(\frac{t}{\alpha t + 1 - \alpha}) + (1 - \alpha) \log(\frac{1}{\alpha t + 1 - \alpha}) \tag{14}$$

For notational convenience, we abbreviate $f_{JS}^\alpha$ as $f$. This connection allows us to leverage the following theoretical tools developed for general $f$-divergences.

**Assumption A3.2** (Assumption 9 in Pillutla et al. (2023)). We assume that the generator function $f$ of the $f$-divergence must satisfy the following three conditions:

- (A1) The function $f$ and its conjugate generator $f^*$ must be bounded at zero. Formally, $f(0) < \infty$ and $f^*(0) < \infty$.

- (A2) The first derivatives of $f$ and $f^*$ must not grow faster than a logarithmic function. For any $t \in (0,1)$, there must exits constants $C_1$ and $C_1^*$ such that $|f'(t)| \leq C_1(\max(1, \log(1/t)))$ and $|(f^*)'(t)| \leq C_1^*(\max(1, \log(1/t)))$.

- The second derivatives of $f$ and $f^*$ must not grow faster than $\frac{1}{t}$ as $t \to 0$. Formally, there must exist constants $C_2$ and $C_2^*$ such that for any $t \in (0, \infty)$, $\frac{t}{2}f''(t) \leq C_2$, and $\frac{t}{2}(f^*)''(t) \leq C_2^*$.

**Lemma A3.3** (Approximate Lipschitz Property of the $f$-divergence, Lemma 20 in Pillutla et al. (2023)). *Let $f$ be a generator function satisfying Assumption A3.2. Consider the bivariate scalar function $\psi : [0,1] \times [0,1] \to [0,\infty)$ defined as $\psi(p,q) = qf(\frac{p}{q})$. For all probability values $p, p', q, q' \in [0,1]$ with $\max(p, p') > 0$ and $\max(q, q') > 0$, the following inequalities hold:*

$$|\psi(p',q) - \psi(p,q)| \leq \left( C_1 \max\left(1, \log \frac{1}{\max(p,p')}\right) + \max(C_0^*, C_2) \right) |p - p'| \tag{15}$$

$$|\psi(p,q') - \psi(p,q)| \leq \left( C_1^* \max\left(1, \log \frac{1}{\max(q,q')}\right) + \max(C_0, C_2^*) \right) |q - q'| \tag{16}$$

**Assumption A3.4** (Assumption 3(b) in Kara et al. (2023)). *Let $P(\cdot|x)$ be a probability measure on $(\mathcal{X}, \mathcal{F})$. There exit $L_P < \infty$ such that*

$$\text{TV}(P(\cdot|x) - P(\cdot|x')) \leq L_P|x - x'|, \quad \forall x, x' \in \mathcal{X}. \tag{17}$$

**Proposition A3.5** (Quantization Error of f-Divergence, Proposition 13 in Pillutla et al. (2023)). *Let $P$ and $Q$ be two probability distributions over a common sample space $\mathcal{X}$.*

*Let $S = \{S_1, S_2, \ldots, S_m\}$ be a partition of the space $\mathcal{X}$ into $m$ disjoint sets. The corresponding quantized distributions, $P_S$ and $Q_S$, are defined as multinomial distributions over the indices $\{1, \ldots, m\}$.*

*Then, for any integer $k \geq 1$, and $f$-divergence functional $\mathcal{D}_f$, there exists a partition $S$ of size $m \leq 2k$ such that the absolute difference between the original and the quantized $f$-divergence is bounded as follows:*

$$|\mathcal{D}_f(P,Q) - \mathcal{D}_f(P_S, Q_S)| \leq \frac{f(0) + f^*(0)}{k}$$

Theorem A3.6 is adapted from Theorem 3.1 and Lemma 3.1 of Wolfer (2023), which provide high-probability bounds on the row-wise total variation error of the empirical transition matrix for a finite-state, irreducible, aperiodic Markov chain observed over a single trajectory. The bound holds uniformly over all states and depends explicitly on the number of states and the trajectory length, while accounting for the chain's dependence structure.

**Lemma A3.6** (Row-wise TV bound, Wolfer (2023)). *Let $(X_1, \ldots, X_N)$ be an irreducible, aperiodic, stationary Markov chain on a finite state space $\mathcal{A}$ with $|\mathcal{A}| = k$, transition matrix $M$ and stationary distribution $\pi$. Then there exists a universal constant $C > 0$ such that, for any $0 < \delta < 1$, the following holds with probability at least $1 - \delta$:*

$$\max_{s \in \mathcal{A}} \sum_{a \in \mathcal{A}} \left| \hat{M}(a|s) - M(a \mid s) \right| \leq C \sqrt{\frac{\tau_{\mathrm{mix}} k \log \left( \frac{kN}{\delta} \right)}{N}},$$

*where $\tau_{\mathrm{mix}}$ is a mixing-time–type constant depending only on $M$ (for reversible chains one has $\tau_{\mathrm{mix}} \asymp 1/\gamma_{\mathrm{ps}}$, with $\gamma_{\mathrm{ps}}$ denoting the pseudo–spectral gap).*

We will use the missing-mass bound from Skorski (2020) to handle unseen transitions.

**Lemma A3.7** (Missing Mass Bound, Theorem 1 in Skorski (2020)). *Let $(X_1, \ldots, X_N)$ be an irreducible Markov chain over a finite state space $\mathcal{A}$ with stationary distribution $\pi_P$ and true transition matrix $M_P$. Define the* transition missing mass *as*

$$\mathrm{Mmass} = \sum_{s \in \mathcal{A}} \sum_{s \in \mathcal{A}} \pi_P(s) M_P(a|s) \cdot \mathbf{1}\{\hat{M}_P(a|s) = 0\}.$$

*Let $T$ be the maximum hitting time of any set of states with stationary probability at least $0.5$. Then there exists an absolute constant $c > 0$ and independent Bernoulli random variables*

$$Q_{s,a} \sim \mathrm{Bernoulli} \left( e^{-c \cdot N \cdot \pi_P(s) M_P(a|s)/T} \right)$$

*such that for any subset $\mathcal{E} \subseteq \{(s,a) : s, a \in \mathcal{A}\}$ and any $n \geq 1$,*

$$\Pr \left[ \bigwedge_{(s,a) \in \mathcal{E}} \{\hat{M}_P(a|s) = 0\} \right] \leq \prod_{(s,a) \in \mathcal{E}} \Pr[Q_{s,a} = 1].$$

*In particular, for any $t > 0$ it holds that*

$$\mathbb{E} \exp (t \cdot \mathrm{Mmass}) \leq \mathbb{E} \exp \left( t \cdot \sum_{s \in \mathcal{A}} \sum_{s \in \mathcal{A}} \pi_P(s) M_P(a|s) Q_{s,a} \right).$$

*For bounding deviations of weighted sums over Markov chains, we rely on the inequality of Chung et al. (2012).*

**Lemma A3.8** (Theorem 3.1 of Chung et al. (2012)). *Let $M$ be an ergodic Markov chain on state space $\mathcal{A}$ with stationary distribution $\pi$. For $\varepsilon \leq 1/8$, let $T(\varepsilon)$ denote its total-variation mixing time. Consider a length-$N$ chain $(X_1, \ldots, X_N)$ on $M$ with $X_1 \sim \varphi$. For each $s \in \mathcal{A}$, let $f_s : \mathcal{A} \to [0, 1]$ be a weight function with $\mathbb{E}_{X \sim \pi}[f_s(X)] = \pi(s)$. Define the total weight $N(s) = \sum_{i=1}^{N} f_s(X_i)$. Then there exists an absolute constant $c$ such that:*

$$\Pr\big[N(s) \geq (1+\delta)\pi(s)N\big] \leq c \|\varphi\|_\pi \times \begin{cases} \exp\big(-\delta^2 \pi(s)N/(72\,T(\epsilon))\big), & 0 \leq \delta \leq 1, \\ \exp\big(-\delta\,\pi(s)N/(72\,T(\epsilon))\big), & \delta > 1, \end{cases}$$

*and, for $0 \leq \delta \leq 1$,*

$$\Pr\big[N(s) \leq (1-\delta)\pi(s)N\big] \leq c \|\varphi\|_\pi \exp\big(-\delta^2 \pi(s)N/(72\,T(\epsilon))\big).$$

*Here $\langle u, v \rangle_\pi = \sum_x u_x v_x / \pi(x)$ and $\|u\|_\pi = \sqrt{\langle u, u \rangle_\pi}$.*

### A3.2.2 Auxiliary Results

**Lemma A3.9.** *For all $\alpha > 0$ and $p \in (0, 1]$, it holds that*

$$p \max\{1, \log(1/p)\} e^{-\alpha p} \leq \frac{2 + \log(1 + \alpha)}{e\,\alpha}. \tag{18}$$

*Proof.* Let $y = \alpha p \in (0, \alpha]$ and $A = \log \alpha$. Then we can rewrite

$$p \max\{1, \log(1/p)\} e^{-\alpha p} = \frac{1}{\alpha}\, y e^{-y}\, \max\{1, A - \log y\}.$$

Next observe the inequality

$$\max\{1, A - \log y\} \leq 1 + A_+ + (-\log y)_+,$$

where $x_+ = \max\{0, x\}$ and $A_+ = \max\{0, A\}$.

Therefore,

$$y e^{-y} \max\{1, A - \log y\} \leq (1 + A_+) \cdot y e^{-y} + y e^{-y}(-\log y)_+.$$

Now use the following standard bounds:

$$\sup_{y > 0} y e^{-y} = \frac{1}{e}, \qquad \sup_{0 < y \leq 1} y(-\log y) = \frac{1}{e}.$$

Hence

$$\sup_{y > 0} y e^{-y} \max\{1, A - \log y\} \leq \frac{1 + A_+}{e} + \frac{1}{e} = \frac{2 + \log \alpha_+}{e},$$

where $\log \alpha_+ = \max\{0, \log \alpha\} \leq \log(1 + \alpha)$.

Substituting back into the expression, we obtain

$$p \max\{1, \log(1/p)\} e^{-\alpha p} \leq \frac{1}{\alpha} \cdot \frac{2 + \log(1 + \alpha)}{e}.$$

This proves Eq. 18. $\qquad\square$

**Lemma A3.10** (Stationarity of Quantized Kernels)**.** *Let $S_P$ be the population first-order Markov transition kernel on the continuous surprisal space $\mathbb{R}$ with stationary law $\rho_P$. Fix a shared $k$-bin quantizer $q_k : \mathbb{R} \to \mathcal{A} = \{1, \ldots, k\}$ with boundaries $b_1 < \cdots < b_{k-1}$ partitions space into bins $B_i = [b_i, b_{i+1})$. Define the row-stationary weights and the edge measure*

$$\pi_P(i) := \rho_P(B_i), \qquad Z_P(i, j) := \int_{B_i} \rho_P(\mathrm{d}x)\, S_P(B_j|x), \quad i, j \in \mathcal{A},$$

*and the induced $k$-state transition kernel*

$$M_P(j \mid i) := \frac{Z_P(i, j)}{\pi_P(i)} \qquad (\text{for } \pi_P(i) > 0).$$

*Then $\pi_P$ is a stationary distribution of $M_P$, i.e. $\sum_i \pi_P(i) M_P(j \mid i) = \pi_P(j)$ for all $j \in \mathcal{A}$.*

*Proof.* By definition,

$$\sum_{i \in \mathcal{A}} \pi_P(i) M_P(j \mid i) = \sum_{i \in \mathcal{A}} Z_P(i, j) = \int_{\mathbb{R}} \rho_P(\mathrm{d}x)\, S_P(B_j|x) = \rho_P(B_j) = \pi_P(j),$$

where the penultimate equality uses the stationarity of $\rho_P$ for $S_P$. $\qquad\square$

### A3.2.3 Proof of Theorem 4.2

In this step, we aim to bound the expected absolute difference between the estimated GJS divergence and the GJS divergence for the induced Markov kernels after discretization. The statistical error of our estimator is:

$$E_1 = |\mathcal{D}_f(\hat{M}_P, \hat{M}_Q) - \mathcal{D}_f(M_P, M_Q)| \tag{19}$$

The analysis will reveal how this error depends on the number of bins $k$ and the sequence length $N$. To analyze the statistical error, we will extend the logic used in Pillutla et al. (2023). We will apply Lemma A3.3 (Lemma 20 in Pillutla et al. (2023)), which establishes an approximate Lipschitz property for the core component of any $f$-divergence.

*Proof of Theorem 4.2.* To bound the statistical error $E_1$, we first decompose it and then expand the GJS function into a sum of its core components, allowing for the application of Lemma A3.3. Using the triangle inequality, we can bound the total statistical error by the sum of the errors arising from the estimation of each matrix individually:

$$E_1 \le \underbrace{|\mathcal{D}_f(\hat{M}_P, \hat{M}_Q) - \mathcal{D}_f(M_P, \hat{M}_Q)|}_{=:\mathcal{T}_1} + \underbrace{|\mathcal{D}_f(M_P, \hat{M}_Q) - \mathcal{D}_f(M_P, M_Q)|}_{=:\mathcal{T}_2} \tag{20}$$

The f-divergence between two Markov chains, $M_A$ and $M_B$, is defined as the expected divergence of their row-wise conditional probability distributions, weighted by the stationary distribution of the second chain. Let $\pi_B(s)$ be the stationary probability of state $s$ for chain $M_B$. The f-divergence is:

$$D_f(M_A, M_B) = \sum_{s \in \mathcal{A}} \pi_B(s) \sum_{a \in \mathcal{A}} \psi(M_A(a|s), M_B(a|s)) \tag{21}$$

Applying this to the first term of our decomposed error Eq. equation 20, with $f = f_{JS}^w$, we get

$$\mathcal{T}_1 = |\mathcal{D}_f(\hat{M}_P, \hat{M}_Q) - \mathcal{D}_f(M_P, \hat{M}_Q)| \tag{22}$$

$$= \left| \sum_{s \in \mathcal{A}} \hat{\pi}_Q(s) \sum_{a \in \mathcal{A}} \psi(\hat{M}_P(a|s), \hat{M}_Q(a|s)) - \sum_{s \in \mathcal{A}} \hat{\pi}_Q(s) \sum_{a \in \mathcal{A}} \psi(M_P(a|s), \hat{M}_Q(a|s)) \right| \tag{23}$$

$$= \left| \sum_{s \in \mathcal{A}} \hat{\pi}_Q(s) \sum_{a \in \mathcal{A}} \left[ \psi(\hat{M}_P(a|s), \hat{M}_Q(a|s)) - \psi(M_P(a|s), \hat{M}_Q(a|s)) \right] \right| \tag{24}$$

$$\le \left| \sum_{s \in \mathcal{A}} \sum_{a \in \mathcal{A}} \left[ \psi(\hat{M}_P(a|s), \hat{M}_Q(a|s)) - \psi(M_P(a|s), \hat{M}_Q(a|s)) \right] \right| \tag{25}$$

$$\le \sum_{s \in \mathcal{A}} \sum_{a \in \mathcal{A}} \left| \psi(\hat{M}_P(a|s), \hat{M}_Q(a|s)) - \psi(M_P(a|s), \hat{M}_Q(a|s)) \right| \tag{26}$$

**Case 1: Observed Transitions** For a transition that appears in the human-written text sample, its empirical probability is $\hat{M}_P(a|s) = \frac{N_P(s,a)}{N_P(s)} \ge \frac{1}{N_P(s)}$, where $N_P(s)$ is the number of times state $s$ was visited in the sequence of length $N$. We apply the first inequality of Lemma A3.3 with $p' = \hat{M}_P(a|s), p = M_P(a|s)$, and $q = \hat{M}_Q(a|s)$. The term $\max\left(1, \log \frac{1}{\max(p,p')}\right)$ is bounded by $\log N_P(s)$ as long as $N_P(s) \ge 3$. Thus, the error for a single observed transition is bounded by:

$$\left| \psi(\hat{M}_P(a|s), \hat{M}_Q(a|s)) - \psi(M_P(a|s), \hat{M}_Q(a|s)) \right| \le (C_1 \log N_P(s) + C') \left| \hat{M}_P(a|s) - M_P(a|s) \right| \tag{27}$$

$$\le (C_1 \log N + C') \left| \hat{M}_P(a|s) - M_P(a|s) \right| \tag{28}$$

where $C'$ is a constant absorbing $C_0^*$ and $C_2$. Summing over all observed transitions gives a bound proportional to the Total Variation (TV) distance between the estimated and true transition matrices, multiplied by a logarithmic factor.

**Case 2: Missing Transitions** This case addresses transitions that have a non-zero true probability ($M_P(a|s)$) but were not observed in the finite sample, resulting in an empirical probability of $\hat{M}_P(a|s) = 0$. This scenario is formally known as the missing mass problem for Markov chains, a non-trivial extension of the classic IID case due to the dependencies between samples. To analyze the error contribution, we directly bound the error for a single missing transition using Lemma A3.3. Let $p' = \hat{M}_P(a|s) = 0$ and $p = M_P(a|s)$. The error is now $\left|\psi(0, \hat{M}_Q(a|s)) - \psi(M_P(a|s), \hat{M}_Q(a|s))\right|$. Applying the first inequality of Lemma A3.3, we get:

$$\left|\psi(0, \hat{M}_Q(a|s)) - \psi(M_P(a|s), \hat{M}_Q(a|s))\right| \leq \left(C_1 \max\left(1, \log\frac{1}{M_P(a|s)}\right) + C'\right)\left|0 - M_P(a|s)\right| \tag{29}$$

$$= \left(C_1 \max\left(1, \log\frac{1}{M_P(a|s)}\right) + C'\right)M_P(a|s) \tag{30}$$

This bound shows that the error from a missing transition is proportional to its true probability $M_P(a|s)$, scaled by its information content. The total error from this case is the sum of these individual bounds over all unobserved transitions. This sum constitutes the missing transition mass of the Markov chain.

We summarize the following:

$$\mathbb{E}[\mathcal{T}_1] \leq \left(C_1 \log N + C'\right) \cdot \sum_{s\in\mathcal{A}} \alpha_{N_P(s)}(M_P(\cdot|s)) + \left(C_1 + C'\right)\sum_{s\in\mathcal{A}}\beta_{N_P(s)}(M_P(\cdot|s)) \tag{31}$$

where $M_P(\cdot|s)$ is a $k$-dimensional probability distribution corresponding to state $s$, and we formally define the row-wise error terms:

- Row-wise TV term $\alpha_{N_P(s)}(M_P(\cdot|s))$: This term sums the error from observed transitions in state $s$.

$$\mathbb{E}[\alpha_{N_P(s)}(M_P(\cdot|s))] = \mathbb{E}\left[\sum_{\substack{a\in\mathcal{A}, \\ \text{s.t.}\hat{M}_P(a|s)>0}} \left|\hat{M}_P(a|s) - M_P(a|s)\right|\right] \tag{32}$$

- Row-wise Missing Mass term $\beta_{N_P(s)}(M_P(\cdot|s))$ This term sums the error from unobserved transitions in state $s$.

$$\mathbb{E}[\beta_{N_P(s)}(M_P(\cdot|s))] = \mathbb{E}\left[\sum_{\substack{a\in\mathcal{A}, \\ \text{s.t.}\hat{M}_P(a|s)=0}} M_P(a|s) \cdot \max\left(1, \log\frac{1}{M_P(a|s)}\right)\right] \tag{33}$$

Then we use Lemma A3.6 to upper bound Eq 32.

$$\mathbb{E}[\alpha_{N_P(s)}(M_P(\cdot|s))] = \mathbb{E}\left[\sum_{\substack{a\in\mathcal{A}, \\ \text{s.t.}\hat{M}_P(a|s)>0}} \left|\hat{M}_P(a|s) - M_P(a|s)\right|\right] \tag{34}$$

$$\leq \mathbb{E}\left[\sum_{a\in\mathcal{A}} \left|\hat{M}_P(a|s) - M_P(a|s)\right|\right] \tag{35}$$

$$= O\left(\sqrt{\frac{k\log(kN)}{N}}\right) \tag{36}$$

where Eq. 36 follows Lemma A3.6 by inverting its tail bound and integrating to expectation; the mixing-time constant is absorbed into $O(1)$ under Assumption A3.1.

Lemma A3.7 gives an exponential tail for the event $\hat{M}_P(a|s) = 0$: for some absolute constant $c > 0$ and $T$ the maximum hitting time of any set with stationary probability at least 0.5,

$$\mathbb{P}[\hat{M}_P(a|s) = 0] \leq \exp\left(-\frac{cN}{T}\pi_P(s)M_P(a|s)\right) \tag{37}$$

Then we upper bound the missing mass term $\mathbb{E}[\beta_{N_P(s)}(M_P(\cdot|s))]$. Let $p_a = M_P(a|s)$ and $\Gamma = \frac{cN}{T}\pi_P(s)$.

$$\mathbb{E}[\beta_{N_P(s)}(M_P(\cdot|s))] = \sum_{a\in\mathcal{A}} p_a \max\left(1, \frac{1}{p_a}\right)\mathbb{P}[\hat{M}_P(a|s) = 0] \tag{38}$$

$$= \sum_{a\in\mathcal{A}} p_a \max\left(1, \frac{1}{p_a}\right)e^{-\Gamma p_a} \tag{39}$$

$$\leq \sum_{a\in\mathcal{A}} \frac{2 + \log(1+\Gamma)}{e\Gamma} \tag{40}$$

$$= \frac{kT}{ecN\pi_P(s)}\left(2 + \log\left(1 + \frac{cN\pi_P(s)}{T}\right)\right) \tag{41}$$

where Eq. 40 follows Lemma A3.9 for all $\Gamma > 0$ and $p_a \in (0,1]$. Assuming $\pi_P(s) \geq \frac{c_0}{k}$ for some constant $c_0 > 0$ and $T = O(1)$, we obtain

$$\mathbb{E}[\beta_{N_P(s)}(M_P(\cdot|s))] = O\left(\frac{k^2}{N}\log(1 + \frac{N}{k})\right) \tag{42}$$

By Eq. 31, Eq. 36, and Eq. 42 we obtain

$$\mathcal{T}_1 = O\left(\log N \cdot \sqrt{\frac{k^3\log(kN)}{N}} + \frac{k^3}{N}\log\left(1 + \frac{N}{k}\right)\right) \tag{43}$$

Next we bound $\mathcal{T}_2$.

$$\mathcal{T}_2 = |\mathcal{D}_f(M_P, \hat{M}_Q) - \mathcal{D}_f(M_P, M_Q)| \tag{44}$$

$$= \left|\sum_{s\in\mathcal{A}}\hat{\pi}_Q(s)\sum_{a\in\mathcal{A}}\psi(M_P(a|s), \hat{M}_Q(a|s)) - \sum_{s\in\mathcal{A}}\pi_Q(s)\sum_{a\in\mathcal{A}}\psi(M_P(a|s), M_Q(a|s))\right| \tag{45}$$

$$= \left|\sum_{s\in\mathcal{A}}\hat{\pi}_Q(s)\sum_{a\in\mathcal{A}}\psi(M_P(a|s), \hat{M}_Q(a|s)) - \sum_{s\in\mathcal{A}}\hat{\pi}_Q(s)\sum_{a\in\mathcal{A}}\psi(M_P(a|s), M_Q(a|s))\right.$$
$$\left. + \sum_{s\in\mathcal{A}}\hat{\pi}_Q(s)\sum_{a\in\mathcal{A}}\psi(M_P(a|s), M_Q(a|s)) - \sum_{s\in\mathcal{A}}\pi_Q(s)\sum_{a\in\mathcal{A}}\psi(M_P(a|s), M_Q(a|s))\right| \tag{46}$$

$$\leq \left|\sum_{s\in\mathcal{A}}\hat{\pi}_Q(s)\sum_{a\in\mathcal{A}}\psi(M_P(a|s), \hat{M}_Q(a|s)) - \sum_{s\in\mathcal{A}}\hat{\pi}_Q(s)\sum_{a\in\mathcal{A}}\psi(M_P(a|s), M_Q(a|s))\right|$$
$$+ \left|\sum_{s\in\mathcal{A}}\hat{\pi}_Q(s)\sum_{a\in\mathcal{A}}\psi(M_P(a|s), M_Q(a|s)) - \sum_{s\in\mathcal{A}}\pi_Q(s)\sum_{a\in\mathcal{A}}\psi(M_P(a|s), M_Q(a|s))\right| \tag{47}$$

$$\leq \left|\sum_{s\in\mathcal{A}}\sum_{a\in\mathcal{A}}[\psi(M_P(a|s), \hat{M}_Q(a|s)) - \psi(M_P(a|s), M_Q(a|s))]\right|$$
$$+ \left|\sum_{s\in\mathcal{A}}\left(\hat{\pi}_Q(s) - \pi_Q(s)\right)\sum_{a\in\mathcal{A}}\psi(M_P(a|s), M_Q(a|s))\right| \tag{48}$$

$$\leq \underbrace{\sum_{s\in\mathcal{A}}\sum_{a\in\mathcal{A}}\left|\psi(M_P(a|s), \hat{M}_Q(a|s)) - \psi(M_P(a|s), M_Q(a|s))\right|}_{=:\mathcal{T}_{2,1}}$$

$$+ \underbrace{\left|\sum_{s\in\mathcal{A}}\left(\hat{\pi}_Q(s) - \pi_Q(s)\right)\sum_{a\in\mathcal{A}}\psi(M_P(a|s), M_Q(a|s))\right|}_{=:\mathcal{T}_{2,2}} \tag{49}$$

By symmetry, bounding $\mathcal{T}_{2,1}$ proceeds identically to $\mathcal{T}_1$, and yields the same rate as $\mathcal{T}_1$. To upper bound $\mathcal{T}_{2,2}$, we consider

$$\sum_{a\in\mathcal{A}}\psi(M_P(a|s), M_Q(a|s)) = \sum_{a\in\mathcal{A}}M_Q(a|s)f_{JS}^w(M_P(a|s)/M_Q(a|s)) \leq H(w) \leq \log 2 \tag{50}$$

where $H(w) = -[w \log(w) + (1-w) \log(1-w)]$ with $w = \frac{\alpha}{1+\alpha} \in [0,1]$ is the binary entropy function of which the absolute maximum possible value is $\log 2$. To upper bound $\mathcal{T}_{2,2}$,

$$\mathcal{T}_{2,2} \leq \log 2 \cdot \mathbb{E}|\hat{\pi}_Q - \pi_Q| \tag{51}$$

We apply Lemma A3.8 to upper bound $\mathcal{T}_{2,2}$. Consider $\hat{\pi}_Q(s) = \frac{N_Q(s)}{N}$, for any $\delta > 0$, we have

$$\Pr\big[N_Q(s) \geq (1+\delta)\pi_Q(s)N\big] \leq c\,\|\varphi\|_{\pi_Q} \times \begin{cases} \exp\big(-\delta^2 \pi_Q(s)N/(72\,T)\big), & 0 \leq \delta \leq 1, \\ \exp\big(-\delta\,\pi_Q(s)N/(72\,T)\big), & \delta > 1, \end{cases}$$

and similarly for the lower tail with $0 < \delta < 1$. With $\epsilon = \delta \pi_Q(S)$, we have

$$\Pr\big[|\hat{\pi}_Q(s) - \pi_Q(s)| \geq \epsilon\big] \leq 2c\,\|\varphi\|_{\pi_Q} \times \begin{cases} \exp\big(-\epsilon^2 N/(72\,T\pi_Q(s))\big), & 0 \leq \epsilon \leq \pi_Q(s), \\ \exp\big(-\epsilon\,N/(72\,T)\big), & \epsilon > \pi_Q(s), \end{cases} \tag{52}$$

Using $\mathbb{E}|Z| = \int_0^\infty \Pr(|Z| \geq \epsilon)$ and splitting the integral at $\pi_Q(s)$,

$$\mathbb{E}\big[|\hat{\pi}_Q(s) - \pi_Q(s)|\big] \leq 2c\|\varphi\|_{\pi_Q} \Big( \int_0^{\pi_Q(s)} e^{-\frac{N\epsilon^2}{72T\pi_Q(s)}}\,d\epsilon + \int_{\pi_Q(s)}^\infty e^{-\frac{N\epsilon}{72T}}\,d\epsilon \Big) \tag{53}$$

$$\leq 2c\|\varphi\|_{\pi_Q} \Big( C\sqrt{\frac{T\pi_Q(s)}{N}} + \frac{72T}{N} \exp\big(-\frac{N\pi_Q(s)}{72T}\big) \Big) \tag{54}$$

$$= O\Big( \|\varphi\|_{\pi_Q} \sqrt{\frac{T\pi_Q(s)}{N}} \Big) \tag{55}$$

Thus we obtain

$$\mathbb{E}\big[|\hat{\pi}_Q - \pi_Q|\big] = \sum_{s \in \mathcal{A}} \mathbb{E}\big[|\hat{\pi}_Q(s) - \pi_Q(s)|\big] \tag{56}$$

$$\leq \sum_{s \in \mathcal{A}} C\|\varphi\|_{\pi_Q} \sqrt{\frac{T\pi_Q(s)}{N}} \tag{57}$$

$$= C\|\varphi\|_{\pi_Q} \sqrt{\frac{T}{N}} \sum_{s \in \mathcal{A}} \sqrt{\pi_Q(s)} \tag{58}$$

$$\leq C\|\varphi\|_{\pi_Q} \sqrt{\frac{Tk}{N}} \tag{59}$$

$$= O\Big(\frac{k}{\sqrt{N}}\Big) \tag{60}$$

where Eq. 60 holds since $\|\varphi\|_{\pi_Q} = \frac{1}{\sqrt{\pi_Q(s_0)}} \leq \frac{1}{\sqrt{\min_{s \in \mathcal{A}} \pi_Q(s)}} = O(\sqrt{k})$ for the first state $s_0$, and $T = O(1)$. To sum up, $\mathcal{T}_{2,2} = O\big(\frac{k}{\sqrt{N}}\big)$, and the rate for the total statistical error is

$$E_1 \leq \mathcal{T}_1 + \mathcal{T}_{2,1} + \mathcal{T}_{2,2} \tag{61}$$

$$= O\Big( \log N \cdot \sqrt{\frac{k^3 \log(kN)}{N}} + \frac{k^3}{N} \log\big(1 + \frac{N}{k}\big) \Big)$$

$$+ O\Big( \log N \cdot \sqrt{\frac{k^3 \log(kN)}{N}} + \frac{k^3}{N} \log\big(1 + \frac{N}{k}\big) \Big) + O\Big(\frac{k}{\sqrt{N}}\Big) \tag{62}$$

$$= O\Big( \log N \cdot \sqrt{\frac{k^3 \log(kN)}{N}} + \frac{k^3}{N} \log\big(1 + \frac{N}{k}\big) + \frac{k}{\sqrt{N}} \Big) \tag{63}$$

$\square$

### A3.2.4 PROOF OF PROPOSITION 4.1

*Proof of Proposition 4.1.* Let $\rho_P$ and $\rho_Q$ be the continuous stationary distributions of $\mathcal{S}_P$ and $\mathcal{S}_Q$ respectively. We expand $\mathcal{D}_f(\mathcal{S}_P, \mathcal{S}_Q)$ and $\mathcal{D}_f(M_P, M_Q)$,

$$\mathcal{D}_f(\mathcal{S}_P, \mathcal{S}_Q) = \int_{\mathbb{R}} \rho_Q(\mathrm{d}x) \mathcal{D}_f(\mathcal{S}_P(\cdot|x), (\mathcal{S}_Q(\cdot|x)) \tag{64}$$

$$\mathcal{D}_f(M_P, M_Q) = \sum_{i \in \mathcal{A}} \pi_Q(i) \mathcal{D}_f(M_P(\cdot|i), M_Q(\cdot|i)) \tag{65}$$

The quantizer $q_k : \mathbb{R} \to \mathcal{A} = [k]$ with boundaries $b_1 < \cdots < b_{k-1}$ partitions space into bins $B_i = [b_i, b_{i+1})$. Let $\rho_Q(B_i) = \int_{B_i} \mathrm{d}\rho_Q(x)$, then $\pi_Q(i) = \rho_Q(B_i)$.

Define two intermediate objects $U_P$ and $U_Q$ to be markov kernel such that each has a discrete state index $i \in \mathcal{A}$, within a given state $i$, the observable variable $x$ lives in a continuous space $\mathbb{R}$, The corresponding stationary distributions over states are $\pi_P$ for $P$ and $\pi_Q$ for $Q$. Thus

$$\mathcal{D}_f(U_P, U_Q) = \sum_{i \in \mathcal{A}} \pi_Q(i) \mathcal{D}_f(\mathcal{S}_P(\cdot|i), \mathcal{S}_Q(\cdot|i)) \tag{66}$$

where $\mathcal{S}_P(\cdot|i) = \mathbb{E}_{x \sim \rho_Q(B_i)}[\mathcal{S}_P(\cdot|x)]$ and similarly for $\mathcal{S}_Q(\cdot|i)$. We have

$$|\mathcal{D}_f(\mathcal{S}_P, \mathcal{S}_Q) - \mathcal{D}_f(M_P, M_Q)| \leq |\mathcal{D}_f(\mathcal{S}_P, \mathcal{S}_Q) - \mathcal{D}_f(U_P, U_Q)| + |\mathcal{D}_f(U_P, U_Q) - \mathcal{D}_f(M_P, M_Q)| \tag{67}$$

The second term is bounded as

$$|\mathcal{D}_f(U_P, U_Q) - \mathcal{D}_f(M_P, M_Q)| \tag{68}$$

$$= \left| \sum_{i \in \mathcal{A}} \pi_Q(i) \mathcal{D}_f(\mathcal{S}_P(\cdot|i), \mathcal{S}_Q(\cdot|i)) - \sum_{i \in \mathcal{A}} \pi_Q(i) \mathcal{D}_f(M_P(\cdot|i), M_Q(\cdot|i)) \right| \tag{69}$$

$$\leq \sum_{i \in \mathcal{A}} \pi_Q(i) \left| \mathcal{D}_f(\mathcal{S}_P(\cdot|i), \mathcal{S}_Q(\cdot|i)) - \mathcal{D}_f(M_P(\cdot|i), M_Q(\cdot|i)) \right| \tag{70}$$

$$= O\left(\frac{1}{k}\right) \tag{71}$$

Eq. 71 holds by applying Proposition A3.5 to each term in Eq. 70, yielding an $O(1/k)$ bound per term. Since the weighted sum of $O(1/k)$ terms remains $O(1/k)$, the overall bound follows. The first term is

$$\mathcal{D}_f(\mathcal{S}_P, \mathcal{S}_Q) - \mathcal{D}_f(U_P, U_Q) \tag{72}$$

$$= \int_{\mathbb{R}} \rho_Q(\mathrm{d}x) \mathcal{D}_f(\mathcal{S}_P(\cdot|x), (\mathcal{S}_Q(\cdot|x)) - \sum_{i \in \mathcal{A}} \pi_Q(i) \mathcal{D}_f(\mathcal{S}_P(\cdot|i), \mathcal{S}_Q(\cdot|i)) \tag{73}$$

$$= \sum_{i=1}^{k} \int_{B_i} \rho_Q(\mathrm{d}x) \mathcal{D}_f(\mathcal{S}_P(\cdot|x), (\mathcal{S}_Q(\cdot|x)) - \sum_{i \in \mathcal{A}} \pi_Q(i) \mathcal{D}_f(\mathcal{S}_P(\cdot|i), \mathcal{S}_Q(\cdot|i)) \tag{74}$$

$$= \sum_{i \in \mathcal{A}} \rho_Q(B_i) \mathbb{E}_{x \sim \rho_Q(B_i)}[\mathcal{D}_f(\mathcal{S}_P(\cdot|x), (\mathcal{S}_Q(\cdot|x))] - \sum_{i \in \mathcal{A}} \pi_Q(i) \mathcal{D}_f(\mathcal{S}_P(\cdot|i), \mathcal{S}_Q(\cdot|i)) \tag{75}$$

$$= \sum_{i \in \mathcal{A}} \pi_Q(i) \mathbb{E}_{x \sim \rho_Q(B_i)}[\mathcal{D}_f(\mathcal{S}_P(\cdot|x), (\mathcal{S}_Q(\cdot|x))] - \sum_{i \in \mathcal{A}} \pi_Q(i) \mathcal{D}_f(\mathcal{S}_P(\cdot|i), \mathcal{S}_Q(\cdot|i)) \tag{76}$$

$$= \sum_{i \in \mathcal{A}} \pi_Q(i) \left[ \mathbb{E}_{x \sim \rho_Q(B_i)}[\mathcal{D}_f(\mathcal{S}_P(\cdot|x), (\mathcal{S}_Q(\cdot|x))] - \mathcal{D}_f(\mathcal{S}_P(\cdot|i), \mathcal{S}_Q(\cdot|i)) \right] \tag{77}$$

$$=: \sum_{i \in \mathcal{A}} \pi_Q(i) J_i \tag{78}$$

Because $\mathcal{D}_f$ is jointly convex,

$$\mathcal{D}_f(\mathbb{E}_{x \sim \rho_Q(B_i)}[\mathcal{S}_P(\cdot|x)], \mathbb{E}_{x \sim \rho_Q(B_i)}[\mathcal{S}_Q(\cdot|x)]) \leq \mathbb{E}_{x \sim \rho_Q(B_i)}[\mathcal{D}_f(\mathcal{S}_P(\cdot|x), (\mathcal{S}_Q(\cdot|x))] \tag{79}$$

Therefore,

$$|\mathcal{D}_f(\mathcal{S}_P, \mathcal{S}_Q) - \mathcal{D}_f(U_P, U_Q)| = \sum_{i \in \mathcal{A}} \pi_Q(i) J_i \tag{80}$$

Lemma A3.3 implies a Lipschitz-type continuity bound in total variation distance, that is

$$|\mathcal{D}_f(P, Q) - \mathcal{D}_f(P', Q')| \leq 2L_f(\text{TV}(P, P') + \text{TV}(Q, Q')) \tag{81}$$

where $L_f$ depends on $C_1, C_1^*, C_2, C_2^*$ in Lemma A3.3. Applying Eq. 81 to $J_i$ yields

$$J_i = \mathbb{E}_{x \sim \rho_Q(B_i)}[\mathcal{D}_f(\mathcal{S}_P(\cdot|x), (\mathcal{S}_Q(\cdot|x))] - \mathcal{D}_f(\mathcal{S}_P(\cdot|i), \mathcal{S}_Q(\cdot|i)) \tag{82}$$

$$\leq \mathbb{E}_{x \sim \rho_Q(B_i)}[\mathcal{D}_f(\mathcal{S}_P(\cdot|x), (\mathcal{S}_Q(\cdot|x)) - \mathcal{D}_f(\mathcal{S}_P(\cdot|i), \mathcal{S}_Q(\cdot|i))] \tag{83}$$

$$\leq 2L_f \mathbb{E}_{x \sim \rho_Q(B_i)}[\text{TV}(\mathcal{S}_P(\cdot|x), \mathcal{S}_P(\cdot|i)) + \text{TV}(\mathcal{S}_Q(\cdot|x), \mathcal{S}_Q(\cdot|i))] \tag{84}$$

By Assumption A3.4,

$$\text{TV}(\mathcal{S}_P(\cdot|x), \mathcal{S}_P(\cdot|i)) + \text{TV}(\mathcal{S}_Q(\cdot|x), \mathcal{S}_Q(\cdot|i)) \leq (L_P + L_Q)\mathbb{E}_{x' \sim \rho_Q(B_i)}|x - x'| \tag{85}$$

Let $c_i$ be the centroid of $B_i$ and define the mean radius $r_i = \mathbb{E}_{x \sim \rho_Q(B_i)}|x - c_i|$. For any $x \in B_i$,

$$\mathbb{E}_{x' \sim \rho_Q(B_i)}|x - x'| \leq |x - c_i| + \mathbb{E}_{x' \sim \rho_Q(B_i)}|x' - c_i| = |x - c_i| + r_i \tag{86}$$

Then,

$$(L_P + L_Q)\mathbb{E}_{x' \sim \rho_Q(B_i)}|x - x'| \leq (L_P + L_Q)\mathbb{E}_{x' \sim \rho_Q(B_i)}|x - c_i| + r_i = 2(L_P + L_Q)r_i \tag{87}$$

Then,

$$J_i \leq 4L_f(L_P + L_Q)r_i \tag{88}$$

Summing over buckets with weight $\pi_P(i)$ gives:

$$|\mathcal{D}_f(\mathcal{S}_P, \mathcal{S}_Q) - \mathcal{D}_f(U_P, U_Q)| = \sum_{i \in \mathcal{A}} \pi_Q(i) J_i \tag{89}$$

$$\leq 4L_f(L_P + L_Q) \sum_{i \in \mathcal{A}} \pi_Q(i) r_i \tag{90}$$

$$= 4L_f(L_P + L_Q)\mathbb{E}_{x \sim \rho_Q}[x - q_k(x)] \tag{91}$$

$$= O(1/k) \tag{92}$$

By Eq. 71 and Eq. 92,

$$|\mathcal{D}_f(\mathcal{S}_P, \mathcal{S}_Q) - \mathcal{D}_f(M_P, M_Q)| \leq \frac{c}{k} \tag{93}$$

$\square$

### A3.2.5 BALANCING TWO ERRORS

A clear choice for k is found by balancing the dominant statistical error (Eq. 63) with the quantization error (Eq. 93) in rate form, ignoring logarithmic factors. The leading statistical term scales as $c_1 k^{\frac{3}{2}} N^{-\frac{1}{2}}$ and the quantization term as $\frac{c_2}{k}$. Minimizing their sum $f(k) = c_1 k^{\frac{3}{2}} N^{-\frac{1}{2}} + \frac{c_2}{k}$ by first-order condition $f'(k) = 0$ yields that

$$k^* = \left(\frac{4c_2}{3c_1}\right)^{\frac{2}{7}} N^{\frac{1}{5}} \tag{94}$$

Thus, up to constants and polylog factors, the optimal bin count is $k^* = \Theta(N^{\frac{1}{5}})$.

### A3.3 DECISION STATISTIC ANALYSIS

#### A3.3.1 AUXILIARY RESULTS FROM LITERATURE

**Lemma A3.11** (Second-Order Taylor Expansion of Generalized Jensen Shannon Divergence, Zhou et al. (2018)). *Let $P_1, P_2 \in \mathcal{P}(\mathcal{X})$ be two distinct probability distributions over a finite alphabet $\mathcal{X}$, representing a point of expansion. Let $\hat{P}_1, \hat{P}_2 \in \mathcal{P}(\mathcal{X})$ be two other probability distributions in a neighborhood of $(P_1, P_2)$. Let $\alpha$ be a fixed positive constant. The Generalized Jensen-Shannon (GJS) divergence, viewed as a function $GJS(\hat{P}_1, \hat{P}_2, \alpha)$, has the following second-order Taylor approximation around the point $(P_1, P_2)$.*

$$GJS(\hat{P}_1, \hat{P}_2, \alpha) = \underbrace{GJS(P_1, P_2, \alpha)}_{\text{Zeroth-Order Term}} + \underbrace{\sum_{x \in \mathcal{X}} (\hat{P}_1(x) - P_1(x))\alpha\iota_1(x) + \sum_{x \in \mathcal{X}} (\hat{P}_2(x) - P_2(x))\iota_2(x)}_{\text{First-Order Term}}$$

$$+ \underbrace{O\left(||\hat{P}_1 - P_1||^2 + ||\hat{P}_2 - P_2||^2\right)}_{\text{Remainder Term}} \tag{95}$$

*where the remainder term is of the order of the squared Euclidean distance between the points, $GJS(P_1, P_2, \alpha)$ is the zeroth-order term, the GJS function evaluated at the point of expansion $(P_1, P_2)$. The first-order term is a linear function of the differences $(\hat{P}_1 - P_1)$ and $(\hat{P}_2 - P_2)$. The summation is taken over all symbols $x$ in the alphabet $\mathcal{X}$. The partial derivatives of the GJS function, evaluated at $(P_1, P_2)$, are given by the information densities.*

$$\iota_1(x) := \iota_1(x|P_1, P_2, \alpha) = \log \frac{(1+\alpha)P_1(x)}{\alpha P_1(x) + P_2(x)} \tag{96}$$

$$\iota_2(x) := \iota_2(x|P_1, P_2, \alpha) = \log \frac{(1+\alpha)P_2(x)}{\alpha P_1(x) + P_2(x)} \tag{97}$$

**Lemma A3.12** (Central Limit Theorem for Additive Functionals, Holzmann (2005)). *Let $(X_1, \ldots, X_N)$ be a stationary, ergodic, discrete-time Markov chain with state space $\mathcal{S}$, transition operator $M$, and unique stationary distribution $\pi$. Let $f : \mathcal{S} \to \mathbb{R}$ be a real-valued function defined on the state space, and assume its expectation with respect to the stationary distribution is zero, i.e., $\mathbb{E}_\pi[f(x)] = 0$. Consider the additive functional $S_N(f) = \sum_{i=1}^{N} f(X_i)$. If a martingale approximation to $S_N(f)$ exits, then the Central Limit Theorem holds, i.e.:*

$$\frac{S_N(f)}{\sqrt{N}} \xrightarrow{d} N(0, \sigma^2(f)) \tag{98}$$

*The term $\sigma^2(f)$ is the asymptotic variance of the process.*

**Lemma A3.13** (Asymptotic Variance for Markov Chains, Holzmann (2005)). *Under the same conditions as Lemma A3.12, the asymptotic variance $\sigma^2(f)$ of the additive functional $S_N(f)$ is given by:*

$$\sigma^2(f) = 2 \lim_{\epsilon \to 0} \langle g_\epsilon, f \rangle - \|f\|^2 \tag{99}$$

*where $g_\epsilon$ is the solution to the following equation $((1+\epsilon)I - M)^{-1}$, which is a function defined on the state space $\mathcal{A}$. $\langle g_\epsilon, f \rangle$ is the inner product in the Hilbert space $L_2(\pi)$, calculated as $\langle g_\epsilon, f \rangle = \sum_{x \in \mathcal{A}} \pi(x)g_\epsilon(x)f(x)$. $\|f\|^2$ is the squared norm of the function $f$ in the space $L_2(\pi)$, which is its variance with respect to the stationary distribution.*

#### A3.3.2 PROOF OF PROPOSITION 4.3

*Proof of Proposition 4.3.* Let $\mathcal{F}_k$ be the family of stationary first-order Markov models on $\mathcal{A} := [k]$. Consider the following likelihood ratio,

$$\Lambda_{n,N} = \frac{1}{n} \log \frac{\sup\limits_{M,M' \in \mathcal{F}_k} M((a_{1:N}^P, a_{1:n}^T)) \, M'(a_{1:N}^Q)}{\sup\limits_{M,M' \in \mathcal{F}_k} M(a_{1:N}^P) \, M'((a_{1:N}^Q, a_{1:n}^T))} \tag{100}$$

$$= \frac{1}{n} \log \frac{\hat{M}_{\alpha 1}((a_{1:N}^P, a_{1:n}^T))\hat{M}_Q(a_{1:N}^Q)}{\hat{M}_P(a_{1:N}^P)\hat{M}_{\alpha 2}((a_{1:N}^Q, a_{1:n}^T))} \tag{101}$$

where $(a_{1:N}^P, a_{1:n}^T)$ denotes the concatenation of $a_{1:N}^P$ and $a_{1:n}^T$, $\hat{M}_{\alpha 1} = \frac{\alpha \hat{M}_P + \hat{M}_T}{1+\alpha}$, and $\hat{M}_{\alpha 2} = \frac{\alpha \hat{M}_Q + \hat{M}_T}{1+\alpha}$. By Eq. (4)-(6) in Gutman (1989), we have

$$\sup_{M \in \mathcal{F}_k} M\big((a_{1:N}^P, a_{1:n}^T)\big) = 2^{-(N+n)H((a_{1:N}^P, a_{1:n}^T))}, \quad \sup_{M' \in \mathcal{F}_k} M'\big(a_{1:N}^Q\big) = 2^{-N H(a_{1:N}^Q)}, \quad (102)$$

$$\sup_{M' \in \mathcal{F}_k} M'\big((a_{1:N}^Q, a_{1:n}^T)\big) = 2^{-(N+n)H((a_{1:N}^Q, a_{1:n}^T))}, \quad \sup_{M \in \mathcal{F}_k} M\big(a_{1:N}^P\big) = 2^{-N H(a_{1:N}^P)}, \quad (103)$$

where $H(\cdot)$ is the empirical conditional entropy per transition in the corresponding sequence. Plugging into the ratio gives

$$\Lambda_{n,N} = \frac{N+n}{n} H((a_{1:N}^P, a_{1:n}^T)) - \frac{N}{n} H(a_{1:N}^P) - \Big[ \frac{N+n}{n} H((a_{1:N}^Q, a_{1:n}^T)) - \frac{N}{n} H(a_{1:N}^Q) \Big] \tag{104}$$

With weight $\alpha = N/n$,

$$\Delta\mathrm{GJS}_n = \frac{N+n}{n} H((a_{1:N}^P, a_{1:n}^T)) - H(a_{1:n}^T) - \frac{N}{n} H(a_{1:n}^P)$$

$$- \Big[ \frac{N+n}{n} H((a_{1:N}^Q, a_{1:n}^T)) - H(a_{1:n}^T) - \frac{N}{n} H(a_{1:N}^Q) \Big] \tag{105}$$

The two terms $\pm H(a_{1:n}^T)$ cancel. Thus we obtain $\Delta\mathrm{GJS}_n = \Lambda_{n,N}$

$$\square$$

### A3.3.3  ASYMPTOTIC NORMALITY OF $\Delta\mathrm{GJS}_n$

**Theorem A3.14** (Asymptotic normality of $\Delta\mathrm{GJS}_n$ ). *Assume the setting of Section 4.2 with $\alpha = N/n$ and standard ergodicity, $\Delta\mathrm{GJS}_n$ is asymptotically normal. Under $H_0 : M_T = M_P$, $\mu_{H_0} = -\mathrm{GJS}(M_Q, M_P, \alpha) < 0$, and $\sigma_{H_0}^2 = \frac{\alpha^2}{N^2} \sigma_{1,0}^2 + \frac{1}{n^2} \sigma_{2,0}^2$, where $\sigma_{1,0}^2$ is the long-run variance of the P-reference-side information-density sum and $\sigma_{2,0}^2$ is the long-run variance of the test-side information-density sum (details in Appendix D). Under $H_1 : M_T = M_Q$, $\mu_{H_1} = +\mathrm{GJS}(M_P, M_Q, \alpha) > 0$, and $\sigma_{H_1}^2 = \frac{\alpha^2}{N^2} \sigma_{1,1}^2 + \frac{1}{n^2} \sigma_{2,1}^2$, where $\sigma_{1,1}^2$ is the Q-reference-side long-run variance, and $\sigma_{2,1}^2$ is the test-side long-run variance under $H_1$.*

*In both cases,*

$$\frac{\sqrt{n}(\Delta\mathrm{GJS}_n - \mu_{H_\bullet})}{\sqrt{\sigma_{H_\bullet}^2}} \overset{d}{\Rightarrow} \mathcal{N}(0, 1),$$

*where the bullet $\bullet \in \{0, 1\}$ denotes the active hypothesis.*

*Proof of Theorem A3.14.* We need to establish asymptotic normality of the test statistic $\Delta\mathrm{GJS}_n$ by performing a second-order Taylor Expansion of it and determining the asymptotic mean and asymptotic variance.

Since Lemma A3.11, adapted from Zhou et al. (2018), is a purely mathematical statement about the local properties of the GJS function itself, irrespective of how its input variables are generated, this lemma is equally applicable to Markov sources.

Thus, we can obtain Taylor Expansion of Generalized Jensen Shannon Divergence when it is applied to Markov source. Consider two distinct transition matrices of two Markov sources $M_1, M_2$. Let $\hat{M}_1$ and $\hat{M}_2$ be two other empirical transition matrices in a neighborhood of $(M_1, M_2)$. Let $\alpha$ be a fixed positive constant. The GJS divergence has the following second-order Taylor approximation around the point $(M_1, M_2)$.

$$\mathrm{GJS}(\hat{M}_1, \hat{M}_2, \alpha) = \mathrm{GJS}(M_1, M_2, \alpha)$$

$$+ \sum_{s \in \mathcal{A}} \pi_1(s) \sum_{a \in \mathcal{A}} (\hat{M}_1(a|s) - M_1(a|s))\alpha \iota_1(a|s) + \sum_{s \in \mathcal{A}} \pi_2(s) \sum_{a \in \mathcal{A}} (\hat{M}_2(a|s) - M_2(a|s))\iota_2(a|s)$$

$$+ O\left( ||\hat{M}_1 - M_1||^2 + ||\hat{M}_2 - M_2||^2 \right) \tag{106}$$

where $\pi_1$ and $\pi_2$ denote the stationary distributions of $M_1$ and $M_2$, respectively. And $\iota_1(a|s)$ and $\iota_2(a|s)$ are information densities:

$$\iota_1(a|s) := \iota_1\big((a|s)\big|M_1, M_2, \alpha\big) = \log \frac{(1+\alpha)M_1(a|s)}{\alpha M_1(a|s) + M_2(a|s)} \tag{107}$$

$$\iota_2(a|s) := \iota_2\big((a|s)\big|M_1, M_2, \alpha\big) = \log \frac{(1+\alpha)M_2(a|s)}{\alpha M_1(a|s) + M_2(a|s)} \tag{108}$$

Furthermore, because $\Delta\mathrm{GJS}_n = \mathrm{GJS}\left(\hat{M}_P, \hat{M}_t, \alpha\right) - \mathrm{GJS}\left(\hat{M}_Q, \hat{M}_t, \alpha\right)$ is constructed as the difference of two GJS functions, we can directly apply the Lemma A3.11 to derive the Taylor expansion $\Delta\mathrm{GJS}_n$ itself.

First, we define the following typical set, given any $M \in \mathcal{F}_k$.

$$\mathcal{C}_n(M) := \left\{ a_{1:n} \in \mathcal{A}^n : \max_{s\in\mathcal{A}, a\in\mathcal{A}} |\hat{M}_{a_{1:n}}(a|s) - M(a|s)| \leq \sqrt{\frac{\log n}{n}} \right\} \tag{109}$$

This is a direct generalization of the IID case discussed in Zhou et al. (2018), and can be justified in Lemma 3.1 of Wolfer (2023), which provides a precise asymptotic analysis of the confidence interval width for estimating the transition matrix. Next we establish an upper bound on the probability of atypical sequences. We need a two-step approach: first, ensure the number of visits $N_s$ in sequence $a_{1:n}$ to each state is sufficient, and then apply a concentration inequality under that condition.

$$\mathbb{P}\{a_{1:n} \notin \mathcal{C}_n(M)\} = \mathbb{P}\left\{ \max_{s\in\mathcal{A}, a\in\mathcal{A}} |\hat{M}_{a_{1:n}}(a|s) - M(a|s)| > \sqrt{\frac{\log n}{n}} \right\} \tag{110}$$

$$\leq \sum_{s\in\mathcal{A}} \mathbb{P}\left\{ \max_{a\in\mathcal{A}} |\hat{M}_{a_{1:n}}(a|s) - M(a|s)| > \sqrt{\frac{\log n}{n}} \right\} \tag{111}$$

$$\leq \sum_{s\in\mathcal{A}} \left[ \mathbb{P}\left\{ N_s < \frac{n\pi(s)}{2} \right\} + \mathbb{P}\left\{ \max_{a\in\mathcal{A}} |\hat{M}_{a_{1:n}}(a|s) - M(a|s)| > \sqrt{\frac{\log n}{n}} \,\Big|\, N_s \geq \frac{n\pi(s)}{2} \right\} \right] \tag{112}$$

$$\leq \sum_{s\in\mathcal{A}} \left[ c_1 \exp(-c_2 n\pi(s)) + 2k \exp\left(-2\frac{n\pi(s)}{2} \cdot \frac{\log n}{n}\right) \right] \tag{113}$$

$$= \sum_{s\in\mathcal{A}} \left[ c_1 \exp(-c_2 n\pi(s)) + 2k \cdot n^{-\pi(s)} \right] \tag{114}$$

$$\leq k \left[ c_1 \exp(-c_2 n\pi(s)) + 2k \cdot n^{-\pi(s)} \right] \tag{115}$$

$$:= \tau(n, M) \tag{116}$$

where $\pi(s)$ denotes the stationary probability of state $s$, the first term of Eq. 113 follows Chernoff-Hoeffding inequality for Markov Chains (Corollary 8.1 of Wolfer (2023)), and the second term of Eq. 113 follows McDiarmid's inequality, as its conditions of independence of variables and the bounded differences property are met. This is because the analysis is performed on the sub-problem of transitions from state $s$, conditional on the number of visits $N_s = k$ (where $k \geq \frac{n\pi(s)}{2}$), which ensures the subsequent $k$ transitions can be treated as IID samples. A similar application of this technique is detailed in Wolfer (2023). Moreover, the constant $c_1$ depends on the initial state of the chain, measuring its deviation from the steady state, while $c_2$ depends on the mixing speed of the chain, measuring how quickly it converges to its steady state. Thus,

$$\mathbb{P}\left\{ a_{1:N}^P \notin \mathcal{C}_N(M_P) \quad \text{or} \quad a_{1:n}^T \notin \mathcal{C}_n(M_P) \quad \text{or} \quad a_{1:N}^Q \notin \mathcal{C}_N(M_Q) \right\} \tag{117}$$

$$\leq \mathbb{P}\{a_{1:N}^P \notin \mathcal{C}_N(M_P)\} + P\{a_{1:n}^T \notin \mathcal{C}_n(M_P)\} + \mathbb{P}\{a_{1:N}^Q \notin \mathcal{C}_N(M_Q)\} \tag{118}$$

$$= \tau(\alpha n, M_P) + \tau(n, M_P) + \tau(\alpha n, M_Q) \tag{119}$$

This means as long as the observed Markov chain sequences are sufficiently long, the probability of sequences being atypical can be made arbitrarily small.

Then, under $H_0$, we derive the Taylor expansion of $\Delta\text{GJS}_n = \text{GJS}\left(\hat{M}_P, \hat{M}_T, \alpha\right) - \text{GJS}\left(\hat{M}_Q, \hat{M}_T, \alpha\right)$ around the true transition matrices $(M_P, M_Q)$. The first term is expanded as

$$\text{GJS}\left(\hat{M}_P, \hat{M}_T, \alpha\right) = \text{GJS}(M_P, M_P, \alpha)$$

$$+ \sum_{s \in \mathcal{A}} \pi_P(s) \sum_{a \in \mathcal{A}} (\hat{M}_P(a|s) - M_P(a|s))\alpha\iota_1(a|s) + \sum_{s \in \mathcal{A}} \pi_P(s) \sum_{a \in \mathcal{A}} (\hat{M}_T(a|s) - M_P(a|s))\iota_2(a|s)$$

$$+ O\left(||\hat{M}_P - M_P||^2 + ||\hat{M}_T - M_P||^2\right) \tag{120}$$

where $\text{GJS}(M_P, M_P, \alpha) = 0$, and for a given symbol $a$ and state $s$,

$$\iota_1(a|s) := \iota_1((a|s)\big| M_P, M_P, \alpha) = \log \frac{(1+\alpha)M_P(a|s)}{\alpha M_P(a|s) + M_P(a|s)} = 0 \tag{121}$$

$$\iota_2(a|s) := \iota_2((a|s)\big| M_P, M_P, \alpha) = \log \frac{(1+\alpha)M_P(a|s)}{\alpha M_P(a|s) + M_P(a|s)} = 0 \tag{122}$$

Thus $\text{GJS}\left(\hat{M}_P, \hat{M}_T, \alpha\right) = O\left(||\hat{M}_P - M_P||^2 + ||\hat{M}_T - M_P||^2\right)$. Then, the second term of $\Delta\text{GJS}_n$ is expanded as

$$\text{GJS}\left(\hat{M}_Q, \hat{M}_T, \alpha\right) = \text{GJS}(M_Q, M_P, \alpha)$$

$$+ \sum_{s \in \mathcal{A}} \pi_Q(s) \sum_{a \in \mathcal{A}} (\hat{M}_Q(a|s) - M_Q(a|s))\alpha\iota_1(a|s) + \sum_{s \in \mathcal{A}} \pi_P(s) \sum_{a \in \mathcal{A}} (\hat{M}_T(a|s) - M_P(a|s))\iota_2(a|s)$$

$$+ O\left(||\hat{M}_Q - M_Q||^2 + ||\hat{M}_T - M_P||^2\right) \tag{123}$$

where

$$\iota_1(a|s) := \iota_1((a|s)\big| M_Q, M_P, \alpha) = \log \frac{(1+\alpha)M_Q(a|s)}{\alpha M_Q(a|s) + M_P(a|s)} \tag{124}$$

$$\iota_2(a|s) := \iota_2((a|s)\big| M_Q, M_P, \alpha) = \log \frac{(1+\alpha)M_P(a|s)}{\alpha M_Q(a|s) + M_P(a|s)} \tag{125}$$

Therefore, we obtain the expansion for $\Delta\text{GJS}_n$ and

$$\Delta\text{GJS}_n = -\text{GJS}(M_Q, M_P, \alpha)$$

$$- \sum_{s \in \mathcal{A}} \pi_Q(s) \sum_{a \in \mathcal{A}} (\hat{M}_Q(a|s) - M_Q(a|s))\alpha\iota_1(a|s) - \sum_{s \in \mathcal{A}} \pi_P(s) \sum_{a \in \mathcal{A}} (\hat{M}_t(a|s) - M_P(a|s))\iota_2(a|s)$$

$$+ O\left(\frac{\log n}{n}\right) \tag{126}$$

Here we connect GJS to information densities,

$$\text{GJS}(M_Q, M_P, \alpha) = \alpha\text{D}_{KL}(M_Q, \frac{\alpha M_Q + M_P}{1+\alpha}) + \text{D}_{KL}(M_P, \frac{\alpha M_Q + M_P}{1+\alpha}) \tag{127}$$

$$= \alpha \sum_{s \in \mathcal{S}} \pi_Q(s) \sum_{a \in \mathcal{A}} M_Q(a|s) \log \frac{M_Q(a|s)}{\frac{\alpha M_Q(a|s) + M_P(a|s)}{1+\alpha}} + \sum_{s \in \mathcal{S}} \pi_P(s) \sum_{a \in \mathcal{A}} M_P(a|s) \frac{M_Q(a|s)}{\frac{\alpha M_Q(a|s) + M_P(a|s)}{1+\alpha}} \tag{128}$$

$$= \alpha \sum_{s \in \mathcal{S}} \pi_Q(s) \sum_{a \in \mathcal{A}} M_Q(a|s) \log \frac{(1+\alpha)M_Q(a|s)}{\alpha M_Q(a|s) + M_P(a|s)} + \sum_{s \in \mathcal{S}} \pi_P(s) \sum_{a \in \mathcal{A}} M_P(a|s) \frac{(1+\alpha)M_Q(a|s)}{\alpha M_Q(a|s) + M_P(a|s)} \tag{129}$$

$$= \alpha \sum_{s \in \mathcal{S}} \pi_Q(s) \sum_{a \in \mathcal{A}} M_Q(a|s)\iota_1(a|s) + \sum_{s \in \mathcal{S}} \pi_P(s) \sum_{a \in \mathcal{A}} M_P(a|s)\iota_2(a|s) \tag{130}$$

where $\iota_1(a|s)$ and $\iota_2(a|s)$ are defined in Eq. 124 and Eq. 125. We subsititute Eq. 130 into Eq. 126 and obtain

$$\Delta\text{GJS}_n = -\alpha \sum_{s\in\mathcal{A}} \pi_Q(s) \sum_{a\in\mathcal{A}} \hat{M}_Q(a|s)\iota_1(a|s) - \sum_{s\in\mathcal{A}} \pi_P(s) \sum_{a\in\mathcal{A}} \hat{M}_T(a|s)\iota_2(a|s) + O\left(\frac{\log n}{n}\right) \tag{131}$$

Recall that $\hat{M}_Q(a|s) = \frac{N_Q(s,a)}{N_Q(s)}$, where $N_Q(s)$ is the number of occurences of state $s$ in $a_{1:N}^Q$, and $N_Q(s,a)$ the number of times $s$ is followed by $a$ in $a_{1:N}^Q$. According to Ergodic Theorem (Strong Law of Large Numbers, e.g. Levin & Peres (2017), Theorem C.1), we consider a long Markov chain to be time-homogeneous, that is for a state $s$, we have $N_Q(s) \approx N \cdot \pi_Q(s)$. Based on this, we simplify the first term of Eq.131.

$$\sum_{s\in\mathcal{A}} \pi_Q(s)\alpha \sum_{a\in\mathcal{A}} \hat{M}_Q(a|s)\iota_1(a|s) = \alpha \sum_{s\in\mathcal{A}} \pi_Q(s) \sum_{a\in\mathcal{A}} \frac{N_Q(s,a)}{N_Q(s)}\iota_1(a|s) \tag{132}$$

$$= \frac{\alpha}{N} \sum_{s\in\mathcal{A}} \sum_{a\in\mathcal{A}} N_Q(s,a)\iota_1(a|s) \tag{133}$$

$$= \frac{\alpha}{N} \sum_{i=2}^{N} \iota_1(a_i^Q|a_{i-1}^Q) \tag{134}$$

Similarly, the second term of Eq.131 is simplified as:

$$\sum_{s\in\mathcal{A}} \pi_P(s) \sum_{a\in\mathcal{A}} \hat{M}_T(a|s)\iota_2(a|s) = \frac{1}{n} \sum_{i=2}^{n} \iota_2(a_i^T|a_{i-1}^T) \tag{135}$$

Combining Eq.134 and Eq.135, we get

$$\Delta\text{GJS}_n = -\frac{\alpha}{N} \sum_{i=2}^{N} \iota_1(a_i^Q|a_{i-1}^Q) - \frac{1}{n} \sum_{i=2}^{n} \iota_2(a_i^T|a_{i-1}^T) + O\left(\frac{\log n}{n}\right) \tag{136}$$

Then we compute the asymptotic mean and asymptotic variance of Eq. 136. By comparing Eq. 130 and Eq. 131, we obtain the asymptotic mean.

$$\mathbb{E}[\Delta\text{GJS}_n] = -\text{GJS}(M_Q, M_P, \alpha) \tag{137}$$

Eq. 136 shows that the random behavior of $\Delta\text{GJS}_n$ is primarily determined by two additive functionals on Markov chains. Since the two reference sequences, $a_{1:N}^Q$ and $a_{1:n}^T$ are mutually independent, the total variance is the sum of their individual variances.

$$\text{Var}(\Delta\text{GJS}_n) = \text{Var}(-\frac{\alpha}{N} \sum_{i=2}^{N} \iota_1(a_i^Q|a_{i-1}^Q)) + \text{Var}(-\frac{1}{n} \sum_{i=2}^{n} \iota_2(a_i^T|a_{i-1}^T)) \tag{138}$$

$$= \frac{\alpha^2}{N^2}\text{Var}(\sum_{i=2}^{N} \iota_1(a_i^Q|a_{i-1}^Q)) + \frac{1}{n^2}\text{Var}(\sum_{i=2}^{n} \iota_2(a_i^T|a_{i-1}^T)) \tag{139}$$

Here we use Lemma A3.12 and A3.13 to compute the asymptotic variance for $\Delta\text{GJS}_n$. We begin by defining a new Markov chain whose state at time $i$ is given by $b_i := (a_{i-1}^Q, a_i^Q)$. Then we can define a function $f_1$ that acts on the state $b_i$, $f_1(b_i) =:= \iota_1(a_i^Q|a_{i-1}^Q)$. With these definitions, we have successfully converted the original sum over transitions into a sum over the states of the new chain, which perfectly fits the framework of Lemma A3.12 and A3.13.

$$\sum_{i=2}^{N} \iota_1(a_i^Q|a_{i-1}^Q) \Leftrightarrow \sum_{i=2}^{N} f_1(b_i) \tag{140}$$

According to Lemma A3.13, the asymptotic variance $\sigma_1^2$ of the additive functional $\sum_{i=2}^N f_1(b_i)$ is given by

$$\sigma_{1,0}^2 = 2 \lim_{\epsilon \to 0} \langle g_{1,\epsilon}, f_1 \rangle - \|f_1\|^2 \tag{141}$$

Now we need to calculate the two main components of this formula. The stationary distribution $\pi'$ of the new chain is determined by $\pi' = \pi_Q(s) \cdot M_Q(a|s)$. By Eq. 130, we get

$$\mu_1 = \mathbb{E}_{\pi'}[f_1(b)] = \sum_{(s,a) \in \mathcal{A} \times \mathcal{A}} \pi'(s,a) f_1(s,a) \tag{142}$$

$$= \sum_{s \in \mathcal{A}} \pi_Q(s) \sum_{a \in \mathcal{A}} M_Q(a|s) \iota_1(a|s) \tag{143}$$

$$= \mathrm{D}_{KL}(M_Q, \frac{\alpha M_Q + M_P}{1 + \alpha}) \tag{144}$$

We obtain the centered function

$$\tilde{f}_1(s,a) = f_1(s,a) - \mu_1 = \iota_1(a|s) - \mu_1 \tag{145}$$

Then according to Lemma A3.13, we calculate the squared norm $\|\tilde{f}_1\|^2$, which is the variance of $\tilde{f}_1$ under the stationary distribution $\pi'$.

$$\|\tilde{f}_1\|^2 = \mathrm{Var}_{\pi'}(f_1) = \mathbb{E}_{\pi'}[(\tilde{f}_1(b))^2] = \sum_{(s,a) \in \mathcal{A} \times \mathcal{A}} \pi'(s,a)(\iota_1(a|s) - \mu_1)^2 \tag{146}$$

Calculating the inner product $\langle g_{1,\epsilon}, \tilde{f}_1 \rangle$ requires first finding $g_{1,\epsilon}$ by solving the resolvent equation:

$$g_{1,\epsilon} = ((1 + \epsilon)I - M_b)^{-1} \tilde{f}_1 \tag{147}$$

where $M_b$ is the transition operator of the new chain and can be constructed from $M_Q$. Each element of the $M_b$ matrix, $M_b((s,a),(s',a'))$, represents the probability of the new chain transitioning from state $(s,a)$ to state $(s',a')$.

$$M_b((s,a),(s',a')) = \begin{cases} M_Q(a'|s') & \text{If } s' = shift(s,a) \\ 0 & \text{otherwise} \end{cases} \tag{148}$$

where $shift(s,a)$ denotes an operation that removes the first element of the sequences $s$ and appends $a$ to the end. After solving $g_{1,\epsilon}$, we compute the inner product:

$$\langle g_{1,\epsilon}, f_1 \rangle = \sum_{(s,a) \in \mathcal{A} \times \mathcal{A}} \pi'(s,a) g_{1,\epsilon}(s,a) \tilde{f}_1(s,a) \tag{149}$$

We take the limit $\lim_{\epsilon \to 0} \langle g_{1,\epsilon}, f_1 \rangle$, then substitute the limit and the value of Eq. 146 into Eq. 141 get the final asymptotic variance $\sigma_{1,0}^2$. Similarly, we use the same method to calculate the asymptotic variance $\sigma_{2,0}^2 = \mathrm{Var}(\sum_{i=2}^n \iota_2(a_i^T|a_{i-1}^T))$. While the asymptotic variance does not generally admit a closed-form expression, Lemma A3.12 and A3.13 provide us with constructive representations. They can be used to compute or approximate the asymptotic variance in practice.

Now we have proved that under $H_0$, the asymptotic normality of $\Delta\mathrm{GJS}_n$, that is

$$\frac{\sqrt{n}(\Delta\mathrm{GJS}_n - \mu)}{\sigma_{H_0}} \xrightarrow{d} \mathcal{N}(0,1) \tag{150}$$

where $\mu_{H_0} = \mathbb{E}[\Delta\mathrm{GJS}_n] = -\mathrm{GJS}(M_Q, M_P, \alpha)$ and variance $\sigma_{H_0}^2 = \frac{\alpha^2}{N^2}\sigma_{1,0}^2 + \frac{1}{n^2}\sigma_{2,0}^2$.

Analogously, under $H_1$, we can prove the asymptotic normality of $\Delta\mathrm{GJS}_n$ with $\mu_{H_1} = \mathrm{GJS}(M_P, M_Q, \alpha)$ and variance $\sigma_{H_1}^2 = \frac{\alpha^2}{N^2}\sigma_{1,1}^2 + \frac{1}{n^2}\sigma_{2,1}^2$, where $\sigma_{1,1}^2 = \mathrm{Var}(\sum_{i=2}^N \iota_1(a_i^P|a_{i-1}^P))$ and $\sigma_{2,1}^2 = \mathrm{Var}(\sum_{i=2}^n \iota_2(a_i^T|a_{i-1}^T))$. As discussed in the variance framework above, they can be represented by the resolvent formulation as in Eq. 141 and Eq. 147.

$\square$

## A4 EXPERIMENTS: CONFIGURATIONS AND MORE RESULTS

### A4.1 IMPLEMENTATION AND CONFIGURATIONS

Our implementation is adapted from MAUVE ((Pillutla et al., 2023)) and Lastde ((Xu et al., 2025)). All detection experiments were conducted on one RTX 4090, while data generation ran on an A40 GPU. We use 9 open-source models and 3 closed-source models for generating text. Open-source models include GPT-XL (Radford et al., 2019), GPT-J-6B (Wang & Komatsuzaki, 2021), GPT-Neo-2.7B (EleutherAI, 2021), GPT-NeoX-20B (Black et al., 2022), OPT-2.7B (Zhang et al., 2022), Llama-2-13B (Touvron et al., 2023), Llama-3-8B (Llama Team, 2024), Llama-3.2-3B (Meta AI, 2024), and Gemma-7B (Gemma Team, Google DeepMind, 2024). Closed-source models include Gemini-1.5-Flash (Gemini Team, Google, 2024), GPT-4.1-mini (OpenAI, 2025a), and GPT-5-Chat (OpenAI, 2025b).

**Generation Pipeline**  In our generation pipeline, for each dataset, we filtered out samples with text length less than 150 words and always condition only on the first 30 tokens of the human text. Each machine passage is generated between 100 and 200 tokens. After generation, we pair each human passage with its corresponding machine passage and truncate both to the shorter side (measured in words). Thus every human-machine pair used for detection has the same length and there is no systematic length advantage for either class.

**Default Decoding Strategy**  In our experiments, unless otherwise specified, for each model family we use a fixed default decoding configuration. Concretely, for open-source models on HuggingFace we use the standard decoding configuration temperature = 1.0, top-p = 1.0, top-k = 50. For GPT-4.1-mini and GPT-5-chat (OpenAI API), we follow the default settings temperature = 1.0, top-p = 1.0 (no top-k parameter). For Gemini, we use the default settings of the Gemini API, temperature = 1.0, top-p = 0.95, top-k = 64.

### A4.2 MORE RESULTS

#### A4.2.1 EXPANSION OF TABLE 1 AND TABLE 2

Table 9,10,11, 12,13, and 14 show the detection results on XSum, WritingPrompts,and SQuAD datasets. The performance is the average over three detections, where each detection is conducted on a randomly sampled test set.

| | Gemini-1.5-Flash | GPT-4.1-mini | GPT-5-Chat | Avg |
|---|---|---|---|---|
| Likelihood | 53.2 $\pm$1.31 | 55.54 $\pm$1.09 | 43.03 $\pm$2.69 | 50.59 |
| LogRank | 52.01 $\pm$2.53 | 57.96 $\pm$2.81 | 45.86 $\pm$3.88 | 51.94 |
| Entropy | 63.19 $\pm$1.78 | 51.7 $\pm$1.02 | 56.8 $\pm$2.02 | 57.23 |
| DetectLRR | 49.85 $\pm$2.54 | 62.26 $\pm$0.91 | 54.14 $\pm$3.6 | 55.42 |
| Lastde | 59.26 $\pm$3.39 | 55.97$\pm$2.18 | 45.3 $\pm$1.34 | 53.51 |
| Lastde++ | **76.9** $\pm$1.62 | 69.29 $\pm$2.00 | 48.14 $\pm$3.28 | 64.78 |
| DNA-GPT | 60.85 $\pm$1.41 | 55.7 $\pm$0.46 | 45.4 $\pm$0.77 | 53.98 |
| Fast-DetectGPT | 75.52 $\pm$1.58 | 66.7 $\pm$1.45 | 48.51 $\pm$2.01 | 63.58 |
| DetectGPT | 62.58 $\pm$1.31 | 61.25 $\pm$3.08 | 50.17 $\pm$0.29 | 58 |
| DetectNPR | 58.77 $\pm$2.47 | 62.17 $\pm$1.50 | 53.32 $\pm$0.97 | 58.09 |
| R-Detect | 63.68 $\pm$0.77 | 63.43 $\pm$2.31 | 58.74 $\pm$1.62 | 61.95 |
| Binoculars | 74.84 $\pm$2.12 | 61.12 $\pm$1.47 | 45.94 $\pm$0.67 | 60.63 |
| FourierGPT | 52.06 $\pm$0.39 | 55.53 $\pm$2.31 | 61.1 $\pm$1.1 | 56.23 |
| SurpMark$_{k=6}$ | 70.24 $\pm$0.77 | 84.07 $\pm$2.21 | 84.16 $\pm$1.01 | 79.49 |
| SurpMark$_{k=7}$ | 71.22 $\pm$0.32 | 82.52 $\pm$1.11 | **87.02** $\pm$1.4 | 80.25 |
| SurpMark$_{k=8}$ | 69.03 $\pm$1.74 | **85.78** $\pm$0.76 | 86.38 $\pm$0.94 | **80.40** |

Table 9: Detection results on XSum for text generated by 3 closed-source models under the black-box setting.

|  | Gemini-1.5-Flash | GPT-4.1-mini | GPT-5-Chat | Avg |
|---|---|---|---|---|
| Likelihood | 80.53 ±1.29 | 82.95 ±1.23 | 62.00 ±2.95 | 75.16 |
| LogRank | 74.73 ±2.64 | 80.66 ±2.81 | 58.01 ±4.04 | 71.13 |
| Entropy | 46.34 ±3.11 | 19.00 ±6.43 | 25.23 ±4.08 | 30.19 |
| DetectLRR | 48.22 ±2.7 | 68.50 ±1.06 | 43.92 ±2.48 | 53.55 |
| Lastde | 41.09 ±2.88 | 55.72 ±2.62 | 30.64 ±1.59 | 42.48 |
| Lastde++ | 76.90 ±1.05 | 68.49 ±2 | 30.64 ±3.23 | 58.68 |
| DNA-GPT | 78.19 ±0.87 | 63.70 ±1.73 | 45.60 ±3.2 | 62.50 |
| Fast-DetectGPT | 91.96 ±0.31 | 70.23 ±1.91 | 30.01 ±4.07 | 64.07 |
| DetectGPT | 87.12 ±0.49 | 78.04 ±0.9 | 58.72 ±2.01 | 74.63 |
| DetectNPR | 80.47 ±1.23 | 75.80 ±0.97 | 55.97 ±2.31 | 70.75 |
| R-Detect | 83.31 ±0.89 | 78.79 ±1.92 | 77.06 ±0.48 | 79.72 |
| Binoculars | 95.35 ±0.1 | 80.55±0.34 | 42.26±0.67 | 72.72 |
| FourierGPT | 77.8 ±0.36 | 77.96 ±1.05 | 74.45 ±1.72 | 76.74 |
| SurpMark$_{k=6}$ | 86.64 ±2.33 | 85.80 ±0.57 | 82.25 ±1.03 | 84.90 |
| SurpMark$_{k=7}$ | 86.68 ±1.4 | 83.64 ±0.33 | 83.73 ±0.52 | 84.68 |
| SurpMark$_{k=8}$ | 89.43 ±0.35 | **87.27** ±0.14 | **83.56** ±0.67 | **86.75** |

Table 10: Detection results on WritingPrompts for text generated by 3 closed-source models under the black-box setting.

|  | Gemini-1.5-Flash | GPT-4.1-mini | GPT-5-Chat | Avg |
|---|---|---|---|---|
| Likelihood | 35.74 ±3.46 | 61.82 ±3.21 | 43.83 ±2.01 | 47.13 |
| LogRank | 34.86 ±2.61 | 61.78 ±3.52 | 45.62 ±3.66 | 47.42 |
| Entropy | 65.55 ±1.08 | 45.46 ±1.43 | 58.94 ±0.65 | 56.65 |
| DetectLRR | 35.46 ±1.84 | 59.10 ±2.11 | 51.42 ±2.50 | 48.66 |
| Lastde | 44.03 ±1.55 | 60.15 ±2.92 | 49.95 ±3.65 | 51.38 |
| Lastde++ | 52.47 ±1.86 | 66.90 ±2.18 | 51.76 ±3.02 | 57.04 |
| DNA-GPT | 47.15 ±0.93 | 50.74 ±2.88 | 58.45 ±1.18 | 52.11 |
| Fast-DetectGPT | 49.98 ±1.33 | 68.04 ±1.19 | 51.64 ±1.98 | 56.55 |
| DetectGPT | 57.87 ±2.65 | 70.95 ±0.82 | 54.90±0.83 | 61.24 |
| DetectNPR | 55.63 ±2.91 | **74.53** ±1.29 | 55.67 ±2.13 | 61.94 |
| R-Detect | 60.86 ±1.33 | 72.69 ±1.41 | 67.45 ±2.37 | 67 |
| Binoculars | 53.34 ±2.53 | 73.69±0.55 | 60.76 ±0.67 | 62.6 |
| FourierGPT | 53.89 ±2.57 | 55.66 ±2.25 | 58.92 ±2.24 | 56.16 |
| SurpMark$_{k=6}$ | 66.84 ±1.11 | 70.87 ±0.86 | 68.57 ±1.48 | 68.76 |
| SurpMark$_{k=7}$ | **67.51** ±1.3 | 69.27 ±1.83 | 73.23 ±0.87 | **70.00** |
| SurpMark$_{k=8}$ | 59.53 ±1.49 | 72.27 ±1.32 | **74.81** ±1.02 | 68.87 |

Table 11: Detection results on SQuAD for text generated by 3 closed-source models under the black-box setting.

|  | GPT2-XL | GPT-J-6B | GPT-Neo-2.7B | GPT-NeoX-20B | OPT-2.7B | Llama-2-13B | Llama-3-8B | Llama-3.2-3B | Gemma-7B | Avg |
|---|---|---|---|---|---|---|---|---|---|---|
| Likelihood | 76.5 ±0.63 | 62.74 ±1.07 | 58.36 ±1.62 | 60.58 ±1.8 | 68.51 ±1.37 | 92.22 ±0.48 | 93.41 ±0.82 | 51.61 ±0.62 | 55.13 ±1.18 | 68.78 |
| LogRank | 80.16 ±0.89 | 67.83 ±1.13 | 64.54 ±0.98 | 63.58 ±1.25 | 72.33 ±1.56 | 94.56 ±0.32 | 95.05 ±0.17 | 59.35 ±0.08 | 59.13 ±0.68 | 76.89 |
| Entropy | 59.65 ±1.52 | 56.37 ±0.66 | 63.76 ±1.43 | 55.32 ±1.11 | 52.88 ±0.68 | 42.33 ±2.58 | 29.31 ±3.19 | 55 ±2.89 | 53.2 ±1.48 | 50.40 |
| DetectLRR | 83.2 ±0.83 | 76.5 ±0.88 | 76.94 ±1.09 | 68.4 ±1.35 | 77.49 ±0.54 | 95.74 ±0.23 | 94.85 ±0.08 | 75.05 ±0.31 | 66.42 ±1.42 | 81.42 |
| Lastde | 91.97 ±0.44 | 77.99 ±0.89 | 82.49 ±0.85 | 72.12 ±1.63 | 77.85 ±0.68 | 92.01 ±0.89 | 94.29 ±0.38 | 59.52 ±0.05 | 61.09 ±1.27 | 82.57 |
| Lastde++ | 98.99 ±0.21 | 85.38 ±0.63 | 87.5 ±0.11 | 80.3 ±0.92 | 87.93 ±0.54 | 92.52 ±0.43 | 95.9 ±0.14 | 59.9 ±0.08 | 65.68 ±0.97 | 87.51 |
| DNA-GPT | 71.43 ±1.33 | 55.47 ±2.85 | 54.43 ±3.2 | 56.31 ±1.86 | 58.2 ±1.72 | 93.69 ±0.36 | 96.54 ±0.12 | 50.37 ±0.07 | 55.29 ±1.04 | 70.70 |
| Fast-DetectGPT | 95.54 ±0.34 | 78.6 ±0.56 | 81.84±0.88 | 83.76 ±1.28 | 90.55 ±0.77 | 97.77±0.05 | 96.78 ±0.21 | 61.86 ±1.42 | 63.2 ±1.18 | 84.71 |
| DetectGPT | 92.88 ±1.3 | 71.86 ±1.79 | 76.67 ±2.01 | 78.06 ±0.87 | 82.88 ±1.23 | 82.79 ±0.62 | 83.61 ±1.25 | 56.06±2.65 | 61.6 ±2.94 | 77.18 |
| DetectNPR | 91.87 ±1.13 | 72.36 ±1.46 | 78.83 ±0.66 | 76.76 ±1.48 | 84.06 ±1.21 | 94.29 ±0.86 | 92.31 ±0.3 | 59.62 ±1.77 | 60.52 ±1.78 | 80.05 |
| R-Detect | 72.87 ±1.49 | 59.86 ±1.11 | 67.59 ±0.48 | 63.45 ±2.45 | 69.75 ±0.71 | 72.11 ±0.93 | 81.06 ±0.84 | 62.43 ±0.82 | 46.75 ±0.73 | 66.21 |
| Binoculars | 98.87 ±0.13 | 74.66 ±0.48 | 78.05±1.27 | 76.18±1.22 | 79.89 ±0.79 | 96.78 ±0.21 | 96.19 ±0.16 | 48.22 ±0.71 | 63.71 ±0.72 | 79.17 |
| FourierGPT | 51.8 ±1.39 | 52.52 ±2.02 | 50.44 ±2.96 | 59.17 ±0.28 | 48.16 ±3.01 | 63.38 ±2.42 | 59.74 ±3.4 | 51.98 ±1.73 | 53.62 ±0.78 | 54.53 |
| SurpMark$_{k=6}$ | 96.95 ±0.43 | 88.35 ±1.02 | 92.26 ±0.65 | 81.58 ±0.72 | 90.88 ±0.1 | 96.87 ±0.26 | 97.77 ±0.35 | 73.96 ±0.86 | 73.01 ±0.98 | 87.96 |
| SurpMark$_{k=7}$ | 97 ±0.8 | 89.26 ±0.48 | 92.92 ±0.06 | 82.45 ±1.03 | 91.16 ±1.08 | 97.09 ±0.45 | 97.48 ±0.31 | 73.07 ±0.6 | 72.97 ±0.85 | 88.16 |
| SurpMark$_{k=8}$ | 95.55 ±0.21 | 85.49 ±0.63 | 88.33 ±0.83 | 82.35 ±0.49 | 90.19 ±0.41 | 96.83 ±0.16 | 97.24±0.08 | 72.92 ±1.02 | 70.11 ±0.98 | 86.56 |

Table 12: Detection results on XSum for text generated by 9 open-source models under the black-box setting.

| | GPT2-XL | GPT-J-6B | GPT-Neo-2.7B | GPT-NeoX-20B | OPT-2.7B | Llama-2-13B | Llama-3-8B | Llama-3.2-3B | Gemma-7B | Avg |
|---|---|---|---|---|---|---|---|---|---|---|
| Likelihood | 94.55 ±0.63 | 88.73 ±1.11 | 89.67 ±0.84 | 87.12 ±1.13 | 85.15 ±2.55 | 99.48 ±0.2 | 99.61 ±0.08 | 85.95 ±0.35 | 83.16 ±1.45 | 90.38 |
| LogRank | 96.04±0.43 | 91.78 ±1.18 | 92.20 ±1.22 | 89.68 ±0.57 | 89.96 ±0.62 | 99.59 ±0.01 | 99.81 ±0.11 | 89.09 ±1.05 | 86.00 ±0.86 | 92.68 |
| Entropy | 34.72 ±2.75 | 33.64 ±2.81 | 32.82 ±2.13 | 32.63 ±1.74 | 40.88 ±2.17 | 5.83 ±3.74 | 8.42 ±4.86 | 53.00 ±2.55 | 37.16 ±2.4 | 31.01 |
| DetectLRR | 96.96 ±0.31 | 95.31 ±0.42 | 94.85 ±0.16 | 92.03 ±0.32 | 95.68 ±0.64 | 98.57 ±0.12 | 99.81 ±0.03 | 92.44 ±0.17 | 89.19 ±0.03 | 94.98 |
| Lastde | 98.50 ±0.2 | 93.94 ±0.12 | 95.97 ±0.33 | 90.36 ±0.82 | 96.05 ±0.18 | 97.97 ±0.48 | 98.69 ±0.23 | 92.04 ±0.1 | 84.96 ±0.56 | 94.28 |
| Lastde++ | 99.68 ±0.11 | 95.96 ±0.51 | 98.86 ±0.1 | 92.68 ±0.74 | 98.39 ±0.12 | 99.14 ±0.08 | 99.56 ±0.06 | 95.04 ±0.3 | 92.59 ±0.65 | 96.88 |
| DNA-GPT | 90.53 ±1.62 | 85.34 ±1.13 | 85.72 ±0.7 | 83.01 ±1.41 | 85.05 ±1.29 | 98.88 ±0.12 | 99.65 ±0.03 | 84.47 ±0.65 | 80.60 ±0.81 | 88.14 |
| Fast-DetectGPT | 99.67 ±0.02 | 93.80 ±0.6 | 96.62 ±0.31 | 92.22 ±0.27 | 94.99 ±0.52 | 99.56 ±0.01 | 99.84 ±0.04 | 93.55 ±0.53 | 89.36 ±1.03 | 95.51 |
| DetectGPT | 95.88 ±0.2 | 85.83 ±1.15 | 91.12 ±1.52 | 85.17 ±1.84 | 90.13 ±1.21 | 92.67 ±0.63 | 93.10 ±0.61 | 80.08 ±1.07 | 83.10 ±2.3 | 88.56 |
| DetectNPR | 98.29 ±0.2 | 89.77 ±0.33 | 93.02 ±0.92 | 87.96 ±0.55 | 92.36 ±1.43 | 98.20 ±0.51 | 98.52 ±0.18 | 85.22 ±0.5 | 86.71 ±1.03 | 92.23 |
| R-Detect | 86.68 ±1.35 | 75.93±1.06 | 75.23 ±0.59 | 73.83 ±1.1 | 51.03±2.57 | 79.69 ±0.88 | 82.79 ±0.93 | 71.2 ±2.36 | 72.62 ±0.89 | 74.33 |
| Binoculars | 99.6 ±0.03 | 93.7 ±0.51 | 94.96 ±0.21 | 93.22 ±0.21 | 91.33 ±0.86 | 98.9 ±0.16 | 99 ±0.06 | 93.4 ±0.27 | 89.22 ±0.82 | 94.81 |
| FourierGPT | 60.23 ±4.8 | 59.81 ±1.62 | 68.08 ±1.46 | 60.6 ±0.29 | 56.95±3.04 | 91.4 ±0.74 | 91.61 ±1.14 | 58.68 ±1.28 | 61.52 ±0.72 | 67.65 |
| SurpMark$_{k=6}$ | 99.44 ±0.06 | 97.60 ±0.22 | 98.32 ±0.57 | 94.38 ±0.16 | 97.22 ±0.16 | 99.47 ±0.07 | 99.65 ±0.1 | 92.71 ±1.45 | 89.28 ±1.69 | 96.45 |
| SurpMark$_{k=7}$ | 99.27 ±0.12 | 97.29 ±0.61 | 97.63 ±0.17 | 94.31 ±0.12 | 96.79 ±0.52 | 99.53 ±0.06 | 99.86 ±0.02 | 93.61 ±0.41 | 89.42 ±0.95 | 96.41 |
| SurpMark$_{k=8}$ | 99.9 ±0.01 | 96.85 ±1.06 | 97.61 ±0.38 | 93.93 ±0.24 | 96.48 ±0.4 | 99.59 ±0.03 | 99.87 ±0.03 | 91.65±0.37 | 90.37 ±1.43 | 96.25 |

Table 13: Detection results on WritingPrompts for text generated by 9 open-source models under the black-box setting.

| | GPT2-XL | GPT-J-6B | GPT-Neo-2.7B | GPT-NeoX-20B | OPT-2.7B | Llama-2-13B | Llama-3-8B | Llama-3.2-3B | Gemma-7B | Avg |
|---|---|---|---|---|---|---|---|---|---|---|
| Likelihood | 84.00 ±2.33 | 73.00 ±3.12 | 71.93 ±2.95 | 68.40 ±1.32 | 78.01 ±1.25 | 91.47 ±1.43 | 88.77 ±1.01 | 58.11 ±1.86 | 59.10 ±1.58 | 74.75 |
| LogRank | 88.39 ±2.06 | 78.14 ±0.96 | 78.13 ±2.26 | 72.85 ±1.45 | 83.68 ±1.2 | 93.55±0.59 | 90.48 ±1.3 | 64.69±0.64 | 62.41 ±1.72 | 79.15 |
| Entropy | 58.93 ±3.11 | 51.43 ±2.6 | 56.24 ±2.91 | 49.86 ±1.68 | 52.88 ±3.1 | 38.92 ±2.37 | 38.72 ±2.71 | 51.00 ±2.26 | 50.18 ±1.82 | 49.80 |
| DetectLRR | 93.05 ±0.11 | 85.61 ±1.24 | 89.56 ±1.01 | 80.38 ±1.19 | 92.28 ±1.05 | 94.98 ±0.35 | 91.47 ±1.45 | 77.14 ±1.09 | 70.89 ±2.31 | 86.15 |
| Lastde | 97.45 ±0.37 | 85.71 ±1.45 | 88.82 ±0.44 | 78.01 ±1.87 | 92.78 ±1.18 | 89.88 ±1.03 | 90.89 ±0.72 | 67.41 ±2.9 | 62.40 ±2.55 | 83.71 |
| Lastde++ | 99.72 ±0.05 | 93.27 ±0.42 | 96.51 ±0.05 | 82.42 ±0.3 | 96.13 ±0.21 | 94.85 ±0.14 | 94.72 ±0.02 | 77.47 ±0.32 | 72.43 ±0.24 | 89.72 |
| DNA-GPT | 83.97 ±2.21 | 71.23 ±2.17 | 78.21±1.45 | 71.93 ±1.86 | 78.33 ±1.43 | 95.15±0.49 | 95.00 ±0.32 | 59.52 ±1.61 | 60.06 ±1.67 | 77.04 |
| Fast-DetectGPT | 98.60 ±0.05 | 88.09 ±1.05 | 89.00 ±1.18 | 81.79 ±1.58 | 92.89 ±0.6 | 97.32 ±0.28 | 97.32 ±0.05 | 67.56 ±2.47 | 69.29 ±0.61 | 86.87 |
| DetectGPT | 94.59 ±0.43 | 80.95 ±2.04 | 86.34 ±1.21 | 69.04 ±2.6 | 80.45 ±2.84 | 84.08 ±1.65 | 82.13 ±1.72 | 56.56 ±3.7 | 62.44 ±1.54 | 77.40 |
| DetectNPR | 94.64 ±0.26 | 83.59 ±1.24 | 87.34 ±1.29 | 75.01±2.13 | 83.07 ±1.78 | 93.09 ±0.69 | 90.18 ±1.05 | 63.52 ±2.43 | 67.25 ±1.7 | 81.97 |
| R-Detect | 63.58 ±0.97 | 55.04 ±0.64 | 60.28 ±1.67 | 52.77 ±2.64 | 51.03 ±0.72 | 88.15 ±0.69 | 81.06 ±0.87 | 53.03 ±2.77 | 47.02 ±3.4 | 61.33 |
| Binoculars | 99.09 ±0.04 | 88.91 ±1.03 | 89.49 ±0.46 | 76.66 ±1.21 | 89.49 ±0.27 | 95.1 ±0.02 | 94.04 ±0.3 | 63.46 ±0.46 | 67.77 ±2.58 | 84.89 |
| FourierGPT | 52.12±3.12 | 50.5 ±2.56 | 56.5 ±2.79 | 49.76 ±1.82 | 52.3 ±2.61 | 62.49 ±0.86 | 64.82 ±1.22 | 53.83 ±0.72 | 52.38 ±2.49 | 54.97 |
| SurpMark$_{k=6}$ | 97.88 ±0.55 | 92.93 ±0.82 | 94.99 ±0.3 | 84.39±0.18 | 95.37 ±0.6 | 95.89 ±0.49 | 93.76 ±0.35 | 78.54±1.97 | 69.92 ±0.54 | 89.30 |
| SurpMark$_{k=7}$ | 98.77 ±0.72 | 92.74 ±0.45 | 95.72 ±0.38 | 82.45 ±1.03 | 96.68 ±0.65 | 96.13 ±0.3 | 94.17 ±0.57 | 75.55 ±1.21 | 68.27 ±0.95 | 88.94 |
| SurpMark$_{k=8}$ | 98.76 ±0.66 | 90.78 ±0.23 | 94.56±0.1 | 79.36 ±1.67 | 97.26 ±0.21 | 94.81 ±0.41 | 93.32 ±0.16 | 76.55 ±1.2 | 67.47 ±0.83 | 88.10 |

Table 14: Detection results on SQuAD for text generated by 9 open-source models under the black-box setting.

### A4.2.2 EMPIRICAL CALIBRATION OF THE BIN-COUNT SCALING CONSTANT

Our intention in Section 4.2 is justify the scaling law $k = \Theta(N^{1/5})$. In practice, for each dataset we treat the theorem as providing the functional form $k = CN^{1/5}$ and then select $k$ by a small grid search. To handle constant $C$, we examined the ratio $\frac{k}{N^{1/5}}$ across several reference size and found it to be consistently around 0.8. This suggests that in our regime the implicit constant is approximately $C \approx 0.8$, and that the empirically chosen $k$ is well aligned with the theoretical scaling law.

| Number of ref samples | $N$ (approx. total transitions) | Empirical best $k$ | $N^{1/5}$ | $\frac{k}{N^{1/5}}$ |
|---|---|---|---|---|
| 100 | 15,000 | 6 | 6.84 | 0.88 |
| 300 | 45,000 | 7 | 8.52 | 0.82 |
| 400 | 60,000 | 7 | 9.03 | 0.78 |
| 600 | 90,000 | 7 | 9.80 | 0.71 |
| 900 | 135,000 | 9 | 10.62 | 0.85 |

Table 15: Scaling of the empirically optimal number of bins $k$ with the total number of transitions $N$.

### A4.2.3 EMPIRICAL VALIDATION OF THE ASYMPTOTIC NORMAL APPROXIMATION

While the asymptotic variance in Theorem 4.4 does not provide a simple closed-form expression, Appendix A3.3.3 along with Lemma A3.13 give an explicit numerical procedure to solve it. To quantitatively compare this theoretical variance with empirical fluctuations, we proceed as follows. We first compute the theoretical variance using the estimated Markov kernels from reference data. Then we estimate the empirical variance of $\Delta\text{GJS}_n$ in detection procedure. Table 16 reports theoretical variance and empirical variance with test length 250. Overall, the theoretical variance captures the right order of magnitude $\Delta\text{GJS}_n$ fluctuations, so we interpret it as a conservative asymptotic scale parameter rather than a precise finite-sample variance estimator.

Finally, to assess the distributional shape, we ran Shapiro-Wilk tests on the obtained $\Delta\text{GJS}_n$ score, as shown in Table 17, the Shapiro-Wilk statistics are close to 1 and the p-values are not small

| Model | Emp $\sigma^2$ (Human) | Emp $\sigma^2$ (LM) | Th $\sigma^2$ (Human) | Th $\sigma^2$ (LM) | Ratio Th/Emp (Human) | Ratio Th/Emp (LM) |
|---|---|---|---|---|---|---|
| Llama3-8B | $1.15 \times 10^{-5}$ | $1.06 \times 10^{-5}$ | $2.48 \times 10^{-5}$ | $7.50 \times 10^{-5}$ | 2.16 | 7.08 |
| Llama3.2-3B | $1.12 \times 10^{-5}$ | $9.93 \times 10^{-6}$ | $2.73 \times 10^{-5}$ | $9.11 \times 10^{-5}$ | 2.44 | 9.17 |
| Gemma-7B | $1.50 \times 10^{-6}$ | $5.96 \times 10^{-7}$ | $3.56 \times 10^{-6}$ | $2.43 \times 10^{-6}$ | 2.37 | 4.08 |

Table 16: Comparison between empirical and theoretical variances of $\Delta$GJS under human and LM text.

| Setting | SQuAD@GPT-5-chat | WritingPrompts@Llama3-8B | XSum@Qwen3-8B |
|---|---|---|---|
| $H_1$ (LM text) stat | 0.9952 | 0.9856 | 0.9974 |
| $H_1$ (LM text) p-value | 0.9078 | 0.1203 | 0.9969 |
| $H_0$ (human text) stat | 0.9876 | 0.9854 | 0.9929 |
| $H_0$ (human text) p-value | 0.2032 | 0.1143 | 0.6632 |

Table 17: Shapiro-Wilk test statistics and p-values for $\Delta\text{GJS}_n$ under LM-generated ($H_1$) and human ($H_0$) text.

(larger than 0.05). This indicates no evidence against normality and empirically supports the central-limit-theorem-based approximation in Theorem A3.3.3, consistent with the variance comparison above.

### A4.2.4 SCORE DISTRIBUTION

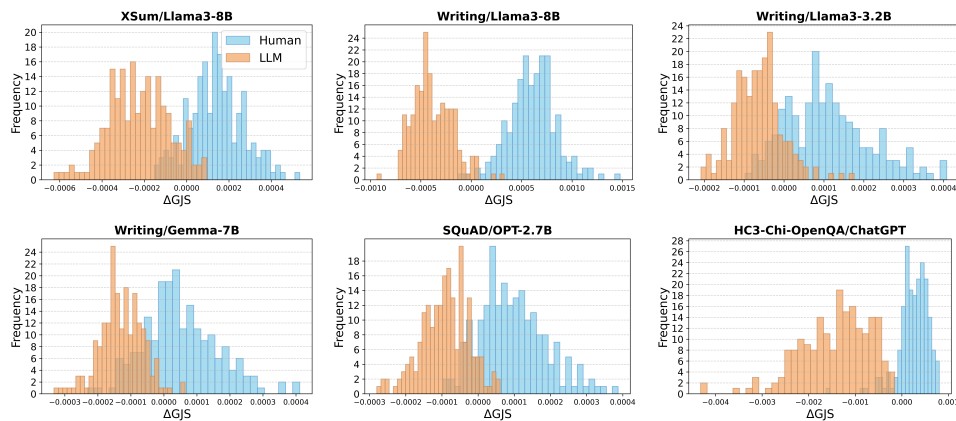

Figure 7: SurpMark's score distribution.

### A4.2.5 EFFECT OF TEST LENGTH

### A4.2.6 TPR

In Table 18, we include TPR@FPR=1% and 5% for SurpMark and two strong baselines (Lastde++ and Fast-DetectGPT) across evaluation settings. Overall, these results indicate that SurpMark is particularly effective in the low-false-positive regime.

### A4.2.7 CROSS-DOMAIN GENERALIZATION

### A4.2.8 DECODING STRATEGIES

In Table 20, to evaluate the effect of decoding stratigy, we use standard decoding strategies as described in Appendix A4.1, varying one hyperparameter at a time while keeping the others at their default values. For open-source models on HuggingFace and Gemini, we (i) set top-p = 0.96, (ii) set top-k = 40, (iii) set temperature = 0.7. For GPT-5-chat, we vary one parameter at a time: top-p = 0.96 or temperature = 0.7 (no top-k parameter is exposed). Across all three models and decoding strategies,

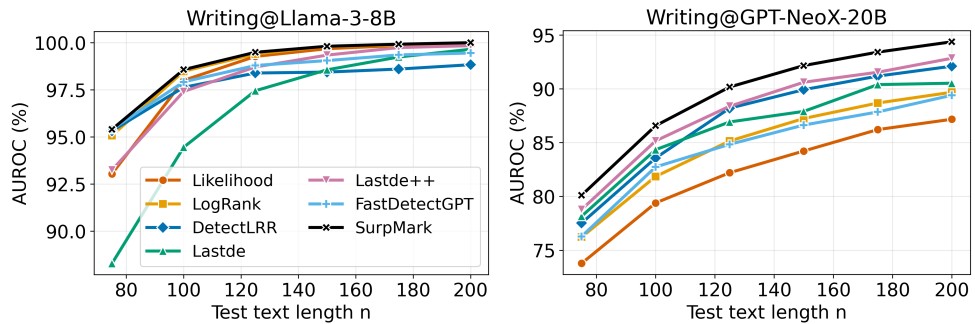

Figure 8: AUROC vs test length.

| Method | | XSum@ GPT-5-Chat | WritingPrompts@ GPT-4.1-mini | XSum@ Llama2-13B | SQuAD@ GPT-Neo-2.7B | WritingPrompts@ Llama3-8B | HC3-Chi-Psy |
|---|---|---|---|---|---|---|---|
| Lastde++ | TPR@FPR=1% | 4.00 | 6.00 | 76.67 | **53.33** | 97.33 | 22.00 |
| | TPR@FPR=5% | 12.67 | 18.67 | 82.67 | 81.33 | 99.33 | 31.33 |
| Fast-DetectGPT | TPR@FPR=1% | 2.00 | 3.30 | **80.67** | 47.33 | 94.67 | 26.00 |
| | TPR@FPR=5% | 4.00 | 22.00 | 86.67 | 77.33 | 98.00 | 40.00 |
| SurpMark | TPR@FPR=1% | **31.33** | **31.33** | 75.33 | 41.33 | **100.00** | **90.00** |
| | TPR@FPR=5% | **37.33** | **50.00** | **90.00** | **90.00** | **100.00** | **97.33** |

Table 18: TPR at fixed FPR levels (1% and 5%) for different detectors and datasets.

SurpMark either matches or exceeds the best baseline, and is especially strong under top-p/top-k sampling.

### A4.2.9 PARAPHRASING ATTACK

Here we examine the robustness of detection methods to the paraphrasing attack. For SurpMark, we consider three paraphrase scenarios. Ref-P applies paraphrasing only to the offline references. Test-P paraphrases only the incoming text, which is the most realistic case in practice. Both-P paraphrases both sides. We follow the setup of Lastde++ and Fast-DetectGPT, and use T5-Paraphraser to perform paraphrasing attacks on texts. Under the practically most relevant Test-P case, the losses are minimal. Under Ref-P, the changes are modest. Under Both-P the drop is larger but still competitive. It shows that SurpMark's surprisal-dynamics features are largely invariant to semantics-preserving rewrites.

### A4.2.10 PROMPT-ENGINEERED ADVERSARIAL ATTACKS

In this section, we run experiments with simple prompt-engineered attacks beyond plain paraphrasing. Specifically, for the XSum and WritingPrompts datasets, we design two types of attacks: (**attack 1**) prompts that ask the model to mimic human writing style, using instructions such as "Messy casual summary of the news article." or "Short story in a quick, slightly messy human style."; and (**attack 2**) prompts that explicitly instruct the model to evade detection, such as "Write a summary of the article that is designed to evade AI-text detectors." or "Continue the story in a way that is hard for AI-text detectors." See Table 22 for comparison. "SurpMark ref-attack" applies the adversarial prompts only when generating the reference machine texts, "SurpMark test-attack" applies them only to the test texts, and "SurpMark both-attack" applies the same adversarial prompts to both the reference and test texts. Across both datasets, SurpMark variants (especially the test-attack and both-attack settings) experience much smaller accuracy drops under all three attacks, showing the strongest overall robustness.

### A4.2.11 ABLATION ON NECESSITY OF FIRST-ORDER MARKOV CHAIN

In Table 23, we evaluate the necessity of the use of first-order markov chain by comparing against the 1-gram distribution of surprisal states. Across the datasets, the first-order Markov features outperform the 1-gram distribution, with especially large gains on GPT-5-chat. This shows that modeling surprisal transitions, rather than only the stationary distribution, is particularly important for harder-to-detect models.

| Test | self-ref | WritingPrompts-as-ref | XSum-as-ref | SQuAD-as-ref |
|---|---|---|---|---|
| XSum@Gemma_7b | 72.97 | 68.32 | – | 73.81 |
| WritingPrompts@Gemma_7b | 89.42 | – | 86.60 | 90.00 |
| SQuAD@Gemma_7b | 68.27 | 72.92 | 70.44 | – |
| XSum@GPT-Neox-20b | 82.45 | 82.52 | – | 81.80 |
| WritingPrompts@GPT-Neox-20b | 94.31 | – | 93.45 | 92.48 |
| SQuAD@GPT-Neox-20b | 82.45 | 81.43 | 83.02 | – |

Table 19: AUROC of SurpMark under different reference choices across datasets and models.

| Method / Data@Model | XSum@OPT-2.7B | | | XSum@Gemma-7B | | | WritingPrompts@GPT-5-chat | |
|---|---|---|---|---|---|---|---|---|
| | top-p | top-k | temperature | top-p | top-k | temperature | top-p | temperature |
| Likelihood | 79.24 | 67.95 | 93.53 | 66.73 | 55.56 | 87.56 | 57.29 | 66.99 |
| LogRank | 82.01 | 72.11 | 94.72 | 68.28 | 59.25 | 88.69 | 55.03 | 65.60 |
| Entropy | 46.87 | 56.53 | 45.27 | 49.16 | 52.12 | 45.06 | 36.24 | 33.68 |
| LRR | 83.43 | 77.88 | 93.48 | 68.66 | 67.86 | 86.22 | 47.28 | 59.38 |
| Lastde | 86.09 | 81.26 | 94.19 | 68.34 | 58.24 | 85.03 | 39.38 | 47.74 |
| Lastde++ | 92.64 | 87.38 | 97.24 | 81.43 | 69.42 | 93.15 | 45.15 | 57.26 |
| Fast-DetectGPT | 90.64 | 85.03 | **98.28** | 80.41 | 68.70 | **95.99** | 41.61 | 57.54 |
| SurpMark $k = 6$ | 92.41 | **87.81** | 96.65 | **82.13** | 72.38 | 93.79 | 75.80 | **77.08** |
| SurpMark $k = 7$ | **93.90** | 87.20 | 95.96 | 80.90 | **77.88** | 93.57 | **77.32** | **77.08** |

Table 20: AUROC of different detectors across decoding parameters, datasets, and models.

### A4.2.12 ANALYSIS OF PERFORMANCE DISPARITY: MARGINAL VS. TRANSITION SURPRISAL

We investigate the performance disparity observed between closed-source (e.g., GPT-5-chat) and open-source models. Our analysis suggests that the distinguishing factor lies in the divergence between the generator and human text at the marginal surprisal level versus the transitional level.

For many open-source models, the marginal surprisal gap—the difference in the stationary distribution of token surprisals—is sufficiently large. Consequently, detectors relying on marginal statistics (e.g., Likelihood, LogRank, Entropy) perform well, and the relative gain from SurpMark is moderate. Conversely, for advanced closed-source models, this marginal gap is nearly negligible, rendering unigram-based methods ineffective. However, a significant transition gap persists in the surprisal dynamics. SurpMark captures these temporal dependencies, explaining its substantial performance advantage on proprietary models.

To quantify this, we compute the Jensen-Shannon (JS) divergence for both marginal surprisal distributions ($JS_{\text{marginal}}$) and first-order transition distributions ($JS_{\text{transition}}$) between human and machine text. As shown in Table 24, for GPT-5-chat, the ratio of transition divergence to marginal divergence is approximately 30, indicating that the signal primarily resides in the dynamics. In contrast, for GPT-J-6B, this ratio is close to 1, suggesting that marginal statistics alone are nearly as informative as transition statistics.

### A4.2.13 THRESHOLD SELECTION

The natural decision rule is simply the sign test by setting $\tau = 0$. Our detector is built around the difference between two GJS divergences. Intuitively, $\Delta GJS$ is positive when the test sequence is closer to the machine reference than to the human reference, and negative in the opposite case. Also, $\Delta GJS$ can be viewed as a log-likelihood ratio $\Lambda_{n,N}$. In the classical Neyman-Pearson framework, the optimal likelihood-ratio test with equal class priors and symmetric costs is precisely $\Lambda_{n,N} \gtrless 0$. We additionally perform a threshold sensitivity study in Table 25. For each dataset and generator, we sweep $\tau$ over the full score range on the test set, compute precision/recall, and identify an optimal threshold $\tau^*$ that maximizes F1. We then compare F1 at our fixed choice $\tau = 0$. Across all generators and datasets, F1 at $\tau = 0$ is typically about 95-97% of the oracle F1. This shows that in practice, our parameter-free sign-based rule already operates very close to the best threshold.

In Lastde, the authors propose a fixed threshold of 2 for Lastde++ regardless of the source model, motivated by plotting score distributions and empirical performance across their experiments. In Table 26, we therefore compare F1 of two methods at their respective threshold. Across three of the four settings, SurpMark achieves higher AUROC, and in all four settings it attains a higher F1. On

| | Xsum@Llama-3-8B | | WritingPrompts@GPT-NeoX-20B | | SQuAD@Llama-2-13B | |
|---|---|---|---|---|---|---|
| | Original | Paraphrased | Original | Paraphrased | Original | Paraphrased |
| Fast-DetectGPT | 96.78 | 95.3 ($\downarrow$1.48) | 92.22 | 89.51 ($\downarrow$2.71) | 94.85 | 92.78 ($\downarrow$2.07) |
| Lastde++ | 93.42 | 91.3 ($\downarrow$2.12) | 92.68 | 91.94 ($\downarrow$0.74) | 97.32 | 92.12 ($\downarrow$5.2) |
| SurpMark Ref-P | 97.77 | 97.06 ($\downarrow$0.61) | 94.31 | 93.12 ($\downarrow$1.19) | 96.13 | 94.89 ($\downarrow$1.24) |
| SurpMark Test-P | 97.77 | 97.33 ($\downarrow$0.44) | 94.31 | 94.05 ($\downarrow$0.26) | 96.13 | 95.46 ($\downarrow$0.67) |
| SurpMark Both-P | 97.77 | 97.17 ($\downarrow$0.6) | 94.31 | 92.22 ($\downarrow$2.09) | 96.13 | 93.98 ($\downarrow$2.15) |

Table 21: Robustness to paraphrase attacks. AUROC on three settings—XSum@Llama-3-8B, WritingPrompts@GPT-NeoX-20B, and SQuAD@Llama-2-13B. For SurpMark, Ref-P/Test-P/Both-P denote paraphrasing the reference set, the test text, or both.

| | WritingPrompts@GPT-J-6B | | | XSum@GPT-J-6B | | |
|---|---|---|---|---|---|---|
| | Original | Attack 1 | Attack 2 | Original | Attack 1 | Attack 2 |
| Lastde++ | 96.96 | 84.24 ($\downarrow$ 12.72) | 85.42 ($\downarrow$ 11.52) | 85.38 | 69.79 ($\downarrow$ 15.59) | 73.55 ($\downarrow$ 11.83) |
| Fast-DetectGPT | 93.80 | 85.95 ($\downarrow$ 7.85) | 79.26 ($\downarrow$ 14.54) | 78.60 | 75.44 ($\downarrow$ 3.16) | 74.09 ($\downarrow$ 4.51) |
| SurpMark ref-attack | 97.60 | 95.06 ($\downarrow$ 2.54) | 94.67 ($\downarrow$ 2.93) | 88.35 | 83.84 ($\downarrow$ 4.51) | 83.85 ($\downarrow$ 4.50) |
| SurpMark test-attack | 97.60 | 95.62 ($\downarrow$ 1.98) | 92.59 ($\downarrow$ 5.01) | 88.35 | 86.37 ($\downarrow$ 1.98) | 84.86 ($\downarrow$ 3.49) |
| SurpMark both-attack | 97.60 | 94.30 ($\downarrow$ 3.30) | 92.74 ($\downarrow$ 4.86) | 88.35 | 84.44 ($\downarrow$ 3.91) | 85.23 ($\downarrow$ 3.12) |

Table 22: AUROC under adversarial attacks for different detectors on GPT-J-6B.

SQuAD@Llama-3-8B, Lastde++ has slightly higher AUROC, but at their fixed thresholds SurpMark still achieves higher F1, indicating SurpMark's sign-based decision rule is better calibrated and less sensitive to threshold choice.

| Metric / Dataset | GPT-J-6B | | | GPT-5-chat | | |
| --- | --- | --- | --- | --- | --- | --- |
| | XSum | WritingPrompts | SQuAD | XSum | WritingPrompts | SQuAD |
| 1-gram distribution | 86.07 | 96.60 | 91.62 | 55.89 | 78.43 | 54.58 |
| First-order Markov chain | 88.35 | 97.60 | 92.93 | 84.16 | 82.25 | 68.57 |

Table 23: AUROC of unigram vs. first-order Markov detectors across models and datasets.

| **Generator** | **Dataset** | **JS-marginal** | **JS-transition** | **Ratio (Transition / Marginal)** |
| --- | --- | --- | --- | --- |
| GPT-J-6B | XSum | 0.00180 | 0.00228 | $\approx 1.27$ |
| | SQuAD | 0.00358 | 0.00392 | $\approx 1.09$ |
| GPT-5-chat | XSum | 0.00006 | 0.00170 | $\approx \mathbf{29.97}$ |
| | SQuAD | 0.00024 | 0.00100 | $\approx 4.17$ |
| GPT-4.1-mini | XSum | 0.00030 | 0.00160 | $\approx 5.33$ |
| | SQuAD | 0.00052 | 0.00150 | $\approx 2.88$ |

Table 24: Comparison of Jensen-Shannon (JS) divergence on marginal surprisal distributions versus first-order transition distributions. The high ratio for closed-source models (e.g., GPT-5-chat) indicates that detection signals are dominated by transition dynamics rather than marginal statistics.

| Setting | AUROC | $\tau^*$ | F1@ $\tau^*$ | F1@ $\tau = 0$ |
| --- | --- | --- | --- | --- |
| XSum@GPT-J-6B | 89.12 | $2.92 \times 10^{-5}$ | 83.56 | 80.36 |
| WritingPrompts@Llama-2-13B | 99.75 | $-9.29 \times 10^{-6}$ | 98.66 | 98.66 |
| SQuAD@Llama-3-8B | 93.56 | $-4.49 \times 10^{-5}$ | 87.58 | 82.69 |
| WritingPrompts@GPT-5-chat | 80.63 | $-1.34 \times 10^{-5}$ | 76.13 | 75.07 |

Table 25: AUROC and F1 scores at the optimal threshold $\tau^*$ and at $\tau = 0$ across different settings.

| Metric | Method | XSum@GPT-J-6B | WritingPrompts@Llama-2-13B | SQuAD@Llama-3-8B | WritingPrompts@GPT-5-chat |
| --- | --- | --- | --- | --- | --- |
| AUROC | Lastde++ | 85.38 | 99.14 | 94.72 | 30.64 |
| | SurpMark $k = 6$ | 88.35 | 99.47 | 93.76 | 82.25 |
| F1 at respective | Lastde++ | 63.44 | 95.56 | 80.93 | 0.00 |
| fixed threshold | SurpMark $k = 6$ | 80.36 | 98.66 | 82.69 | 75.07 |

Table 26: Comparison of AUROC and F1 at fixed thresholds for Lastde++ and SurpMark ($k = 6$) across different settings.

