# OpenReview forum: "Black-box Detection of LLM-generated Text Using Generalized Jensen-Shannon Divergence"
_ICLR.cc/2026/Conference — Submitted to ICLR 2026_

### Official Review · Reviewer_rzH5 · 2025-10-22

**Soundness:** 3
**Presentation:** 4
**Contribution:** 2
**Rating:** 2
**Confidence:** 4

**Summary:**

This paper is well written and interesting. It seeks to improve the common approach of using model log-likelihood of a text as a statistic for discriminating between human and machine generated text. The proposed improvement is do as follows: 1) discretize log-likelihood by putting values of log-likelihood into bins, 2) use human and machine generated reference texts to compute approximate Markov transition probabilities from bin to bin for human text and machine text.3) Use generalized Jensen-Shannon divergence to classify whether a sequence looks more like it was generated from the human-reference Markov process or machine-reference Markov process. Use this as a classifier for human/machine generated text. This is an interesting and novel approach.

Since I will be fairly direct in my criticism of the paper below, let me reiterate here that I think the essential idea of the paper is nice, and congratulate the authors on that.

**Strengths:**

The core idea of the paper is (I think) novel and interesting. It is a fairly simple approach and seems to improve upon the baselines it studies. It is a simple idea which is worthy of study. I am interested in the idea of using k-means clustering, as opposed to other methods, to group the set of next token probabilities.

Some nice mathematics is done around the question of detecting which of two Markov processes generated a sequence.

**Weaknesses:**

1. A lot of interesting mathematics is done, but I don't think this addresses the crucial questions of the effectiveness of the detector. For example, I think the mathematics all concerns the white box case, where machine texts are generated from the exact Markov transition probabilities used in the detector. Similarly, theorem 4.4 (asymptotic normality) is a nice piece of mathematics, but I think it risks giving false confidence in the test, e.g. because it ignores the fact that different language models have different statistics, and that different humans generate text in different ways (non-native speaker bias etc.), that neither humans nor machines generate language according to a one-step Markov process. I think there's a risk that the casual reader takes the very substantial mathematical content of this article as a mathematical proof of the effectiveness of the algorithm, which it isn't.

2. A second point on the mathematics, which I raise for the authors but do not use in calibrating my score since I'm unable to fully defend it. Most of your mathematics concerns the question, 'Given two Markov processes P and Q over the same state space, and a sequence w generated either by P or Q, how can we determine the probability of w being generated by P vs generated by Q'. Given the ubiquity of Markov processes in science, I would be astonished if this question is not extremely well treated in the statistics literature already. There is no need to respond to this comment since I haven't justified it with evidence, it does not affect my score.

3. I know (and dislike that) referees always request more baselines, but I don't think that your choice of baseline detectors is representative of the state of the art. Specifically, what your detector does nicely is use patterns in the structure of log-likelihood, rather than just average over the sequence of log-likelihood. But there are other detectors which do this, for example [Detecting Subtle Differences between Human and Model Languages Using Spectrum of Relative Likelihood](https://aclanthology.org/2024.emnlp-main.564/) (Xu et al., EMNLP 2024)]. Binoculars makes an (apparently strong) claim to outperform Fast-detectGPT, so I think would be useful to include. There are probably many more of which I'm unaware. This is a really crucial point for me, your idea of spotting patterns in log-likelihood rather than averages of log-likelihood is not entirely new, and so I really need to see that you have compared your approach to all of the other methods which try to do this.

4. There are plenty of ways that you could have discretized the space of token probabilities. Do you know whether your k-means clustering represents a sensible chocie? Does it outperform binning, for example, into k-bins which occur with roughly equal frequency in the reference texts?

**Questions:**

Minor comment: I don't think it's fair to characterize Fast-Detect GPT as requiring substantial compute. The novelty of their algorithm is that, rather than computing likelihood of hundreds of perturbations of the original text w_1...w_n, they use only (at time n) the likelihoods of different choices of token n, given the true values of w_1...w_{n-1}. This is information that you get almost for free when computing p(w_n|w_1..w_{n-1}), and so the compute time is comparable with that of just computing the likelihood of the sequence w_1...w_n.

2. Your criticism of classifiers is that they must be retrained for different domains. Do you retrain your baseline Markov transition probabilities for different domains [Edit: I think I get an answer in section 5, that you do retrain for each, using 300 texts. I think you are saying that GPT2-Large is used to compute the machine transition-probabilities in each case, could I double check that this is true. In particular, can I make 100% sure that you do not use the generator model in computing transition-probabilities]?

Do you have a robust claim that retraining these probabilities is easier than for the classifiers? Do you know how susceptible to domain shift your detector is (in circumstances where there is not a good corpus of domain texts to generate your transition probabilities]?

3. (related to 2) On page 3 you mention that you discretize log-likelihoods using k-means clustering. This is the absolutely crucial bit of your algorithm, I would like to know at this point what data you are using to do this clustering. Similarly for the modelling as a Markov chain. In the following section you mention that 'reference texts' t_P and t_Q are used, are the reference texts specific to the language model? Specific to the domain of text generation? This is crucial information about what your algorithm does that shouldn't be buried in the appendix (and I couldn't even find it in the appendix).

4. When we discuss algorithms to detect machine generated text, it is crucial to know how the text was generated. (e.g. Temperature/top-k/top-p?) These generation strategies have an enormous effect on the effectiveness of detectors (as noted in the work of Ippolito et al that you cite). I can't find any reference in your document to how text was generated, it is missing from section A3.1.

5. Please include information on the length of human and machine texts in your experiments. Do you use the full length of human texts from the passages, and cap the machine text at a certain length? I suspect that FastDetect-GPT handles passages of different lengths very poorly, due to the way they normalise their statistic. Does your comparison with FastDetect-GPT hold up if, for example, you ensure that every machine and human text in your test set is exactly 150 tokens in length?

6. If I understand correctly, your main experiments are run on collections of 150 texts. I'm not sure that you should report AUROC to 2dp when you have so few samples.

Minor Comments:
7. On the first page you distinguish between 'Global statistics' and 'distributional statistics'. I'm not sure I understand the distinction or the terminology. The likelihood statistic is computed by summing log likelihood along a sequence. The fast-detect statistic is computed by summing log-likelihood minus expected log likelihood along an orbit. In what sense is one 'global' and one 'distributional'.
8. You claim (p3) to have built a 'likelihood free' hypothesis test. What do you mean by this? You are still using the sequence of (black box model) likelihoods as the primary input. Do you use it only to say that you don't have access to the underlying true human or model likelihoods? If so, this is true of every black box detector. Which is fine, but I misunderstood your wording to mean something stronger.

---

> ### Author Response · Authors · 2025-11-20
> **Response to Question 8 and Weakness 1**
>
> Thank you very much for your detailed reviews. Here are our responses to your concerns:
>
> **Q8: Clarification for Likelihood-free Hypothesis Testing** In classical statistics, “likelihood-free” usually refers to procedures that **do not require evaluating the likelihood of the true data-generating distribution, but instead rely only on observed samples and summary statistics**. In our setting, the observed samples are not raw texts but the discretized surprisal (negative log-likelihood) sequences of the test and reference passages under a fixed proxy LM. Our test never evaluates the true likelihoods of the human and generator distributions. In this sense, the test is “likelihood-free” with respect to the true hypotheses.
>
> **W1: Clarification on the Theoretical Contributions and the Purpose of the Analysis**
>
>
> **White-box misunderstanding** We would like to clarify that in our theoretical and experimental setting, the basic question is: given reference sequences $w^P$ and $w^Q$, produced respectively by a “human” source $P$ and a “machine” source $Q$ whose likelihood functions we cannot evaluate, and a test sequence $w^T$, how can we decide whether $w^T$ was generated by $P$ or by $Q$ using only $w^P$ and $w^Q$(and a proxy language model)? This is a likelihood-free hypothesis testing problem because we never have access to the true likelihoods $p_P(w^T)$ or $p_Q(w^T)$. Instead, we build summary statistics (quantized surprisal Markov transitions) from the proxy LM on the two reference sequences and on $w^T$, and perform a divergence-based test on these summaries. It is black-box because $P$ and $Q$ are treated as generators whose internal parameters and exact likelihoods are not accessible. We only see their reference and never assume any parametric form for their underlying models.
>
> **Task definition** Our task definition follows the **same standard LM-vs-human setup as DetectGPT and DNA-GPT**: we fix a given language model Q and a human text distribution P, and the goal is to decide, for a given text, whether it was generated by Q or drawn from P (DNA-GPT, Sec. 3 on page 3).  Because the LM and the human distribution are fixed in this setup, DNA-GPT's likelihood-gap hypothesis, as well as the theoretical results that build on it, such as the principled choice of the number of regenerations, are also stated for this particular LM-human pair, **rather than for all possible language models and all human writers**. Analogously, our theory analyzes ΔGJS for a fixed LM-human setting, but are not intended as a universal proof of effectiveness across all possible LMs and human populations.
>
>
> **Clarify goal and scope of the theory**
> As discussed at line 53, black-box text detection can be naturally framed as a likelihood-free hypothesis testing. To make LFHT applicable to text detection, we make three adaptations: (**1**) we use token-level surprisals from a fixed proxy LM as observable features (inspired by prior text detection work); (**2**) we apply k-means quantization to obtain discrete, statistically tractable state sequences, (**3**) we propose a **new test statistic**, $\Delta GJS$, **not only new to text detection, but also new to LFHT**, which compares two reference samples rather than relying on a single-sided reference distribution.
>
> These design choices, as well as the empirical findings that a simple first-order Markov model already give good performance (Fig. 2(b)), motivate the theoretical analysis in Section 4: under an idealized first-order Markov model fitted to the discretized surprisal states, we analyze the discretization effect on the estimation of GJS (f-divergences) and the properties of our new statistic $\Delta GJS$.
>
> We do not intend Section 4 to be read as a full mathematical proof that the algorithm will succeed on all realistic generators. In the revision, we will make this scope explicit and reposition the theoretical results as providing (i) **a principled justification for our proposed ΔGJS as a test statistic** under a simple first-order Markov model fitted to the discretized surprisal states, Proposition 4.3 shows that ΔGJS is exactly the log-likelihood ratio comparing the hypothesis that the test state sequence follows the kernel estimated from the human reference to the hypothesis that it follows the kernel estimated from the LM reference. (2) **design guidance and sanity checks for scaling of** $k$, rather than guarantees of performance. We will also add a short statement at the beginning of Section 4 clarifying that the mathematical results are **not a universal proof of effectiveness**, but rather **an analysis of ΔGJS under a simplified model that helps interpret the statistic**.

---

> ### Author Response · Authors · 2025-11-20
> **Response to Weakness 3, 4 and Question 1**
>
> **W3: More Baselines**
> In response to the reviewer's concern, we have added two baselines: Binoculars and Spectrum of Relative Likelihood (FourierGPT). For a fair comparison, we configure Binoculars with GPT-2-Large as the observer model and GPT-2-XL as the performer model, and we use GPT-2-Large as the estimator in FourierGPT to compute the token-level negative log-likelihood sequences. SurpMark achieves the highest AUROC on 14/15 generator-dataset combinations. Please note that we do not claim that using local patterns in log-likelihood is new. In fact, we explicitly state at line 73 that token dynamics expose rich local patterns in likelihood and cite Lastde [1] as an example of a detector that exploits such patterns. In our experiments, we also include Lastde and Lastde++ as strong baselines. We will run comparison with these two baselines for all generator and datasets and merge the results into main tables.
> | Method        |    |    GPT-Neox-20b            |       |        |       OPT-2.7b         |       |       |            Gemma-7b     |       |    |GPT-4.1-mini                 |       |    | GPT-5-chat                  |       |
> |--------------|-------------------|---------------|-------|------------------|---------------|-------|-----------------|---------------|-------|------------------|---------------|-------|------------------|---------------|-------|
> |              | XSum              | WritingPrompts| SQuAD | XSum             | WritingPrompts| SQuAD | XSum            | WritingPrompts| SQuAD | XSum             | WritingPrompts| SQuAD | XSum             | WritingPrompts| SQuAD |
> | Binoculars   | 76.18             | 93.22         | 76.66 | 79.89            | 91.33         | 89.49 | 63.71           | 89.22         | 67.77 | 62.12            | 80.55         | **73.69** | 45.94            | 42.26         | 60.76 |
> | FourierGPT   | 59.17             | 60.60         | 49.76 | 48.16            | 56.95         | 52.30 | 53.62           | 61.52         | 52.38 | 55.53            | 77.96         | 55.66 | 61.10            | 74.45         | 58.92 |
> | SurpMark k=6 | 81.58             | **94.38**         | **84.39** | **90.88**            | **97.22**         | 95.37 | **73.01**           | 89.28         | **69.92** | 84.07            | **85.80**         | 70.87 | **84.07**            | 82.25         | 68.57 |
> | SurpMark k=7 | **82.45**             | 94.31         | 79.36 | 90.19            | 96.79         | **96.68** | 72.97           | **89.42**         | 68.27 | **87.02**            | 83.64         | 69.29 | 82.52            | **83.73**         | **73.23** |
>
> [1] Training-free LLM-generated Text Detection by Mining Token Probability Sequences. Xu et al., ICLR25
>
> **W4: Ablation study on k-means** We thank the reviewer for prompting this analysis. To address this, we ran additional ablations comparing k-means, equal-mass (equal-frequency) binning, and equal-width binning. Across all datasets and k values, k-means is the most robust quantization scheme: it consistently reaches or matches the best AUROC, while equal-mass can degrade sharply on XSum@GPT-4.1-mini and equal-width is unstable and often much worse.
> | Dataset@Source model          | Method              | k = 6 | k = 7 | k = 8 | k = 9 | best|
> |------------------|---------------------|-------|-------|-------|-------|-----|
> | XSum@GPT4.1 mini      | k-means             | 77.43 | 80.42 | 79.32 | 81.79 | 81.79 |
> | XSum@GPT4.1 mini     | equal-width binning | 80.28 | 80.41 | 76.2  | 78.16 | 80.41|
> | XSum@GPT4.1 mini     | equal-mass binning  | 71.69 | 71.4  | 74.22 | 75.2  |75.2 |
> | WritingPrompts@GPT5-chat| k-means             | 80.63 | 82.05 | 84.86 | 83.88 | 84.86 |
> | WritingPrompts@GPT5-chat| equal-width binning | 73.91 | 62.9  | 72.06 | 71.08 | 73.91 |
> | WritingPrompts@GPT5-chat| equal-mass binning  | 82.96 | 84.42 | 83.59 | 84.52 |84.52 |
> | SQuAD@Gemini     | k-means             | 72.64 | 77.74 | 72.56 | 75.01 |77.74|
> | SQuAD@Gemini     | equal-width binning | 60.36 | 58.21 | 54.87 | 56.85 |60.36|
> | SQuAD@Gemini     | equal-mass binning  | 72.93 | 75.08 | 73.65 | 73.41 |75.08|
>
> **Q1: Fast-DetectGPT** We agree our original wording was unfair to Fast-DetectGPT. Our intention was to contrast regeneration-based methods that require hundreds of sampled perturbations (e.g., DetectGPT, NPR) with detectors that only rely on simple proxy-LM statistics. In the revision, we will (i) remove the reference to Fast-DetectGPT in the sentence around Line 45 that discusses “substantial compute”, and (ii) discuss in the Related Work about computational profiles of different detectors to more accurately reflect their relative costs.

---

> ### Author Response · Authors · 2025-11-20
> **Response to Question 2 and 3**
>
> **Q2: Domain Retraining Cost and Proxy Model Usage for References**
>
> **Do we “retrain” per domain?** We do **rebuild** the human and machine reference transition matrices for each dataset/domain, but this is not training a parametric classifier. For each dataset we sample 300 human texts and generate 300 model outputs from the source model, forming the human and machine reference corpora.
>
> **Which model is used to compute transition probabilities?** In all experiments, unless otherwise specified (as in Fig. 5(c)), GPT-2 Large is the proxy LM used to compute token surprisals (negative log probability) for every reference and test passage, regardless of which source model produced the text. The Markov transition probabilities are always estimated from these surprisal sequences via Algorithm 1-2. The source/generator model is used only once offline to produce the raw machine texts for machine corpus. it is never queried for probabilities and never enters the transition estimation.
>
> **No use of the generator inside the detector.** We do not use the source/generator model when computing transition probabilities. All surprisal scores and hence all Markov transitions are derived from the fixed proxy LM (GPT2-Large), consistent with our black-box detection setting.
>
> **Estimating transition probabilities is easier than training classifier.** In practice, building the reference statistics in SurpMark is very lightweight. On our hardware (RTX 4090), constructing the reference (computing surprisal and estimating transition counts for 300 human and 300 LM paragraphs) takes about 12 seconds wall-clock. After surveying recent text-detection papers, we found that most works do not report training or adaptation time for their classifier-based detectors. An earlier benchmark, TURINGBENCH [1], is one exception and explicitly reports per-task run-times for several detectors: the GROVER detector takes about 25-30 minutes, the GPT-2 detector 5-10 minutes, GLTR about 4-5 hours, a BERT-based detector 25-40 minutes, and a RoBERTa-based detector 45 minutes to 1 hour (Table 9 in Appendix A.4). Compared to these ranges, SurpMark's reference-building cost of roughly 12 seconds for 300 human and 300 LM paragraphs is negligible.
>
> [1] TURINGBENCH: A Benchmark Environment for Turing Test in the Age of Neural Text Generation
>
> **Domain shift susceptibility** We measured this directly by forcing out-of-domain (OOD) reference corpora when computing transition probabilities. We compared self-ref (in-domain reference) to OOD reference (associate with the corresponding generator). Even when we deliberately use out-of-domain reference corpora to estimate transition probabilities, the impact is small: across all 24 configurations, AUROC changes by only 1.16 pp on average, with a worst-case deviation of 4.65 pp. Changes remain moderate and can even be positive sometimes.
>
> | Test | self-ref | WritingPrompts-as-ref | XSum-as-ref | SQuAD-as-ref |
> |:--|:--:|:--:|:--:|:--:|
> | Xsum@Llama2-13b | 97.09 | 97.22 | - | 96.45 |
> | WritingPrompts@Llama2-13b | 99.53 | - | 99.71 | 99.33 |
> | SQuAD@Llama2-13b | 96.13 | 94.36 | 95.67 | - |
> | XSum@Llama3-8b | 97.09 | 97.25 | - | 95.13 |
> | WritingPrompts@Llama3-8b | 99.86 | - | 99.95 | 99.88 |
> | SQuAD@Llama3-8b | 94.17 | 93.77 | 93.9 | - |
> | XSum@Gemma_7b | 72.97 | 68.32 | - | 73.81 |
> | WritingPrompts@Gemma_7b | 89.42| - | 86.6 | 90 |
> | SQuAD@Gemma_7b | 68.27 |  72.92 | 70.44 | - |
> | XSum@GPT-Neox-20b | 82.45 | 82.52 | - | 81.8 |
> | WritingPrompts@GPT-Neox-20b | 94.31| - | 93.45 | 92.48 |
> | SQuAD@GPT-Neox-20b | 82.45 | 81.43 | 83.02 | - |
>
> **Q3:  Clarifying k-Means Discretization Data and the Role of Reference Texts**
>
> In our experiments, for each dataset, we first build two reference corpora: human texts $D_Q$ from that domain (e.g., XSum, SQuAD, WritingPrompts) and machine texts $D_P$ generated on the same dataset by the source model. We then use a fixed proxy LM (GPT-2 Large) to compute token surprisals on all texts in $D_Q$ and $D_P$ and run 1-d k-means on this pooled set; the resulting discretizer $q_k$ and Markov transitions are estimated purely from these reference surprisals. Thus, the reference text $t_Q$, $t_P$ are domain-specific, the machine reference text $t_P$ are generator-specific.

---

> ### Author Response · Authors · 2025-11-20
> **Response to Question 4, 5, and 6**
>
> **Q4: Ablation Study on top-p/top-k/temperature**
>
> In our experiments, for each model family we use a fixed default decoding configuration. Concretely, for open-source models on HuggingFace we use the standard decoding configuration temperature = 1.0, top-p = 1.0, top-k = 50. For GPT-4.1-mini and GPT-5-chat (OpenAI API), we follow the default settings temperature = 1.0, top-p = 1.0 (no top-k parameter). For Gemini, we use the default settings of the Gemini API, temperature = 1.0, top-p = 0.95, top-k = 64. We will add these decoding configurations to Appendix in the revision.
>
> To evaluate the effect of decoding stratigy, we use standard decoding strategies, varying one hyperparameter at a time while keeping the others at their default values. For open-source models on HuggingFace and Gemini, we (i) set top-p = 0.96, (ii) set top-k = 40, (iii) set temperature = 0.7. For GPT-5-chat, we vary one parameter at a time: top-p = 0.96 or temperature = 0.7 (no top-k parameter is exposed). Across all three models and decoding strategies, SurpMark either matches or exceeds the best baseline, and is especially strong under top-p/top-k sampling.
>
> | Method / Data@Model | XSum@OPT-2.7B |        |             | XSum@Gemma-7B |        |             | WritingPrompts@GPT-5-chat |             |
> |---------------------|:-------------:|:------:|:-----------:|:-------------:|:------:|:-----------:|:-------------------------:|:-----------:|
> | Method / Metric     |    top-p      | top-k  | temperature |    top-p      | top-k  | temperature |           top-p           | temperature |
> | Likelihood          |     79.24     | 67.95  |    93.53    |     66.73     | 55.56  |    87.56    |           57.29           |    66.99    |
> | LogRank             |     82.01     | 72.11  |    94.72    |     68.28     | 59.25  |    88.69    |           55.03           |    65.60    |
> | Entropy             |     46.87     | 56.53  |    45.27    |     49.16     | 52.12  |    45.06    |           36.24           |    33.68    |
> | LRR                 |     83.43     | 77.88  |    93.48    |     68.66     | 67.86  |    86.22    |           47.28           |    59.38    |
> | Lastde              |     86.09     | 81.26  |    94.19    |     68.34     | 58.24  |    85.03    |           39.38           |    47.74    |
> | Lastde++            |     92.64     | 87.38  |    97.24    |     81.43     | 69.42  |    93.15    |           45.15           |    57.26    |
> | Fast-DetectGPT      |     90.64     | 85.03  |  **98.28**  |     80.41     | 68.70  |  **95.99**  |           41.61           |    57.54    |
> | SurpMark k=6        |     92.41     | **87.81** |  96.65   |   **82.13**   | 72.38  |    93.79    |           75.80           |  **77.08**  |
> | SurpMark k=7        |   **93.90**   | 87.20  |    95.96    |     80.90     | **77.88** | 93.57    |         **77.32**         |  **77.08**  |
>
>
> **Q5: Text Length Control and Comparison with FastDetect-GPT**
> In our generation pipeline, for each dataset, we filtered out samples with text length less than 150 words and always condition only on the first 30 tokens of the human text. Unleass otherwise specified (Fig. 4(b,c), Fig. 5(a,b)), each machine passage is generated between 100 and 200 tokens. After generation, we pair each human passage with its corresponding machine passage and **truncate both to the shorter side** (measured in words). Thus every human-machine pair used for detection has the same length and there is no systematic length advantage for either class. This processing follows the evaluation setup of Lastde [1]. We will add the description in the revision.
>
> To address reviewer's concern about Fast-DetectGPT's sensitity to passage length, we re-ran the comparison under strict length-control. We ensure all passages are generated more than 150 tokens and then truncate to exactly 150 tokens. The results show that SurpMark remains consistently stronger than Fast-DetectGPT.
>
> |     WritingPrompts      | GPT-NeoX-20B | Llama3-8B | Gemma-7B |
> |-----------------|--------------|-----------|----------|
> | Fast-DetectGPT  | 86.63        | 99.05     | 86.56    |
> | SurpMark        | 92.16        | 99.81     | 90.76    |
>
> **Q6: AUROC Precision, Small Sample Size** Our evaluation setup with 150 human and 150 machine test texts per split follows prior work Lastde[1]. Likewise, we report AUROC to two decimal places simply to stay consistent with this prior work and enable direct comparison.
>
> [1] Training-free LLM-generated Text Detection by Mining Token Probability Sequences. Xu et al. ICLR25

---

> ### Author Response · Authors · 2025-11-20
> **Response to Question 7**
>
> **Q7: Distinction Between Global and Distributional Statistics** By global statistics, we meant detectors that evaluate each candidate sequence with a single forward pass of the proxy LM and produce a score by aggregating its token-level outputs. In this sense, the plain log-likelihood test is global: it requires **only one run** of the proxy model on the input and sums the resulting log-likelihoods. By “distributional statistics”, we meant detectors that, for each candidate sequence, generate a local distribution of contrastive samples and base their decision on that induced distribution. In our terminology, Fast-DetectGPT is “distributional” because, for each input sequence, it explicitly constructs a local distribution of conditional log-likelihoods around that sequence and bases its statistic on the mean and variance of this distribution.

---

> ### Comment · Reviewer_rzH5 · 2025-11-23
> **Thanks for responses**
>
> Dear Authors,
>
> Many thanks for your very comprehensive responses to my questions. I'm particularly happy to see the success of k-means clustering (it wasn't obvious to me that using this method of grouping transition probabilities would be sensible, so I'm happy to see a performance uplift, I think this is a nice idea that is unique to your paper).
>
> I think I have only one substantial comment remaining (based on your response).
>
> You write that the Task Definition is just to distinguish between one particular language model and human text. In particular, your transition probabilities for machine text are particular to the language model used for generation. As such, I think this makes your approach less widely applicable than say the approach of Fast-DetectGPT and most alternatives. Most approaches have one fundamental statistic (e.g. some notion of perplexity appropriately modified to take into account perplexity of perturbations of the text). They cheat a little bit by computing AUROC separately for different language model generators, which means they don't have to fix one threshold which works irrespective of the language model being used to generate text. But at least the fundamental method for computing a score for each text does not change with the language model.
>
> By contrast, your method of assigning a score to a text is dependent on knowing which language model was used to generate it. Which, a priori, looks like a weakness to me.
>
> Do you know how effective your method is if you don't recompute transition probabilities for each language model? As in, what is the best result you can get for your method without knowing the language model used for generation? I would like to see some evidence that your method still outperforms the baselines when you don't use knowledge of the generator model to compute your scores.
>
> Thanks again for your very detailed responses.

---

> > ### Comment · Reviewer_rzH5 · 2025-11-24
> > **'Black Box'**
> >
> > Could I just make completely sure I understand what is happening in your black box case.
> >
> > You wish to distinguish between texts generated by one fixed language model and human text, in some given domain. You first look at training data, use GPT2-large (not the generator model) to generate next-token probabilities in the two cases, do your k-means clustering to group probabilities and compute transition probabilities on your training set. You then go to your test samples and run your classifier.
> >
> > Is this a correct understanding? So you don't use internal model probabilities from the generator model, but you do use knowledge of which model is the generator in order to compute your Markov transition probabilities?
> >
> > Black box is used differently in different settings, so I just wanted to check that I understand exactly what you do. Thanks.

---

> > > ### Author Response · Authors · 2025-11-26
> > > **On Black-box**
> > >
> > > Thank you for the careful reading and for checking the details of the black-box setup.
> > >
> > > Broadly speaking, your understanding is correct.
> > >
> > > On the reference samples, we (**1**) use GPT2-large (the proxy, not the generator) to compute next-token probabilities and corresponding surprisal values for both human and generator texts. (**2**) apply $k$-means to the surprisal values to obtain discretized surprisal states. (**3**) estimate two first-order Markov chains over these states, one from the human training texts and one from the generator training texts, by counting state transitions.
> > >
> > > At test time, we again pass each sample (human-written or generator-generated) through GPT2-large, map its surprisal sequence to the same clusters, and then evaluate our GJS-based statistic using the two learned Markov chains. The final detector is a likelihood-ratio–style classifier derived from these two Markov models; there is no additional neural classifier trained on top.
> > >
> > > So, to answer your specific question: we do not use internal probabilities from the generator model at any point. In setup for main experiments, we only use knowledge of which training texts came from that generator to decide which Markov chain (human vs. that generator) their transition counts contribute to.
> > >
> > > Our use of “black box” is precisely in this sense: we only require sample access to the generator, not access to its parameters or token-level probabilities. We will clarify this terminology more explicitly in the revision.

---

> ### Author Response · Authors · 2025-11-26
> **Round 2 Response (1/2)**
>
> To address your concern about relying on generator-specific transition probabilities, we explicitly evaluated SurpMark in a setting where we do not recompute transitions separately for each language model using **two methods**.
>
> (1) **Family-ref SurpMark**
>  On the WritingPrompts dataset, we consider variants of SurpMark. In the original self-ref setting, SurpMark is trained with a separate human/LM reference for each generator. In the more restrictive cross-ref setting, which is closer to your unknown-generator scenario, we instead build a single shared reference per dataset: we sample about 100 human-LM pairs from each of the three generators, pool all of these pairs together, and fit one Markov model and one scoring rule that is used unchanged for all generators. In our experiments we generally observe that AUROC is well maintained when we move from the per-generator “self-ref” setting to the shared “cross-ref” setting.
>
>
> | Method             | Gemini 1.5-Flash | GPT-4.1-mini | GPT-5-chat |
> |:-------------------|-----------------:|-------------:|-----------:|
> | Lastde++           | 76.90            | 68.49        | 30.64      |
> | Fast-DetectGPT     | **91.96**            | 70.23        | 30.01      |
> | SurpMark self-ref  | 89.43            | 87.27        | 83.73      |
> | SurpMark cross-ref | 90.41            | **91.42**        | **81.12**      |
>
> We see a similar pattern on XSum with a different trio of models. However, we also find that the outcome depends on how the reference is constructed: if we pool all six generators together into a single mixed reference, performance degrades noticeably. We attribute this degradation to the heterogeneity between these two families. GPT-J-6B, GPT-neo-2.7B and GPT-neox-20B are older GPT-series language models, while Gemini 1.5-Flash, GPT-4.1-mini and GPT-5-chat are more recent, instruction-tuned commercial systems with different stylistic and distributional characteristics.
>
> | Method             | GPT-J-6B | GPT-neo-2.7B | GPT-neox-20B |
> |:-------------------|---------:|-------------:|------------:|
> | Lastde++           |   85.38  |       87.50  |      80.30  |
> | Fast-DetectGPT     |   78.60  |       81.84  |      83.76  |
> | SurpMark self-ref  |   89.26  |       92.92  |      82.45  |
> | SurpMark cross-ref |   **85.57**  |       **90.18**  |      **79.60**  |

---

> ### Author Response · Authors · 2025-11-26
> **Round 2 Response (2/2)**
>
> (2) **GJS as fingerprint**
>
> We also consider a multi-class “GJS fingerprint” variant that uses a single shared quantizer and a small number of Markov chains. Concretely, we first pool all surprisal sequences from the human reference texts and from several LM reference texts and run one k-means over this pooled set to obtain a shared set of discretization bins. Using this common quantizer, we then estimate one first-order Markov chain for the human class and one machine Markov chain for each generator, i.e., in total (#models + 1) Markov chains. Given a test text
> $t$, we map its surprisal sequence through the same k-means bins, estimate its empirical first-order transition kernel, and compute the GJS divergence between this kernel and each of the (#models + 1) reference chains. If the GJS between $t$ and the human chain is no larger than its GJS to any machine chain, we classify $t$ as human; otherwise, we classify $t$ as coming from the generator whose machine chain yields the smallest GJS to $t$. This decision rule simply treats GJS as a distance and assigns a test text $t$ to whichever source (human or a specific LM) has the Markov chain whose transition pattern is closest to $t$ in that GJS distance.
>
> We test multi-class classification (SurpMark-MC) between GPT-J-6B, GPT-4.1-mini, and LLaMA2-13B (plus human), and this GJS-based nearest-source rule achieves 82.3% overall accuracy, with 78.7% accuracy for human texts, 78.0% for GPT-J-6B, 84.0% for GPT-4.1-mini, and 96.0% for LLaMA2-13B. If we collapse these classes into a binary task (LM vs. human) and use the continuous detection score $ΔGJS=GJS(t,human)-\min_j GJS(t,LM_j)$, we obtain AUROC for each test set in table below. We further evaluate SurpMark-MC on passages not generated by these three models, including LLaMA3-8B, GPT-5-chat, and GPT-neo-20B, and still obtain AUROC scores that are comparable to or better than those of Lastde++ and Fast-DetectGPT.
>
> | WritingPrompts         | LLaMA2-13B | GPT-4.1-mini | GPT-J-6B | LLaMA3-8B | GPT-5-chat | GPT-neo-20B |
> |:---------------|-----------:|-------------:|---------:|----------:|-----------:|------------:|
> | Lastde++       |     99.14  |        68.49 |    **95.96** |     99.56 |      30.64 |       92.68 |
> | Fast-DetectGPT |     99.56  |        70.23 |    93.80 |     99.84 |      30.01 |       92.22 |
> | SurpMark self-ref |  99.59  |        87.27 |    96.85 |     99.87 |      83.56 |       93.93 |
> | SurpMark-MC    |     **99.72**  |        **90.01** |    95.61 |     **99.77** |      **64.27** |       **92.84** |
>
> Overall, these experiments show that SurpMark also performs well in unknown-generator scenarios and exhibits non-trivial generalization to generators that were not explicitly targeted, We believe this robustness largely stems from the fact that the whole framework is grounded in a solid statistical foundation—discretized surprisal Markov models and GJS-based comparison—rather than relying on generator-specific heuristics.

---

### Official Review · Reviewer_EQL7 · 2025-10-29

**Soundness:** 3
**Presentation:** 3
**Contribution:** 3
**Rating:** 6
**Confidence:** 4

**Summary:**

This paper proposes SurpMark, a black-box, reference-based detector for identifying LLM-generated text by measuring its statistical proximity to human and machine reference corpora using the Generalized Jensen-Shannon (GJS) divergence. The method first uses a proxy language model to compute the token surprisal sequence for both the reference texts (from human and machine sources) and the test text. A shared quantizer, derived from the reference surprisals via k-means clustering, is then used to discretize these continuous surprisal values into a finite number of interpretable states. The core of the method summarizes each text—references and test—by estimating its first-order Markov transition matrix over these discrete surprisal states, capturing local token dynamics. Finally, the test text is assigned a GJS score that measures the divergence between its transition pattern and those of the human and machine references, with the decision rule based on this differential score, effectively framing detection as a likelihood-free hypothesis test. The numerical experiment shows the efficiency of the proposed method.

**Strengths:**

1.	The authors propose SurpMark, a novel reference-based detection framework that formulates the identification of LLM-generated text as a likelihood-free hypothesis test using Generalized Jensen-Shannon (GJS) divergence, eliminating the need for per-instance regeneration or classifier training.
2.	A key strength of the method is its principled integration of token surprisal dynamics via discretized first-order Markov transition matrices, which captures local structural patterns in text while ensuring robustness to calibration mismatches between the proxy and source models.
3.	The paper provides a rigorous theoretical analysis justifying the choice of discretization bins through a bias-variance trade-off, proving the GJS statistic's equivalence to a normalized log-likelihood ratio, and establishing its asymptotic normality.
4.	Extensive experiments across multiple datasets, LLMs, and attack scenarios—including paraphrasing—demonstrate strong and consistent performance, with ablation studies validating design choices and confirming theoretical predictions.

**Weaknesses:**

1.	The experimental validation of the first-order Markov assumption in Figure 2(b) is potentially biased, as it compares models of different orders without clarifying whether higher-order models are provided with proportionally larger reference corpora to mitigate their severe data sparsity. Since the number of parameters grows exponentially with the order, higher-order models inherently suffer from greater statistical estimation error, including row-wise transition noise and missing transitions error, nder fixed reference size, making their lower performance likely an artifact of insufficient data rather than evidence of inferior modeling capability; the authors should provide details on the reference data size used for each model order to ensure a reasonable and unbiased comparison.
2.	The definition of the symbol $ N $ in the paper appears to be inconsistent. In the theoretical analysis (e.g., Theorem 4.2), $ N $ should denote the total number of reference transitions, as the statistical error bounds—such as row-wise transition noise and missing transition error—depend on this aggregate quantity. However, on line 245, $ N $ is defined as the length of a single reference library, and in Algorithm 1 (line 5), it is described as the "total number of transitions," which may lead to ambiguity. Furthermore, in the experimental results (e.g., Figures 4 and 5), the "reference length" is varied at small scales, suggesting a per-sequence interpretation. The authors should explicitly clarify whether $ N $ refers to the length per sequence or the total aggregated number of transitions across all sequences to avoid confusion in both theoretical and empirical contexts.
3.	The paper positions kernel-based reference methods like R-Detect[A] and MMD-MP[B] as requiring costly kernel training and permutation testing, yet fails to include a direct and comprehensive comparison in the main results, relegating it to an appendix without full experimental transparency. Moreover, the emphasis on the need for training kernel parameters in these methods is misleading, as their offline training costs can be amortized over large-scale deployment. The authors should provide a head-to-head comparison with R-Detect under identical reference corpus sizes and test conditions in the main tables, along with empirical runtime measurements to substantiate claims about computational efficiency.
4.	The paper does not sufficiently evaluate the robustness of SurpMark in scenarios where test samples exhibit significant distributional shifts relative to the fixed reference corpora, which may limit its reliability in real-world deployments with heterogeneous input streams. Given that the method relies on discretized Markov dynamics different from R-Detect[A] and MMD-MP[B], the authors should provide additional experiments assessing performance under domain mismatch or dynamic data drift to better validate its practical generalizability.
5.	The paper's figures, while visually informative, are not provided in vector format, resulting in noticeable blurriness—particularly in dense plots such as Figure 4 and Figure 5. Several figures suffer from suboptimal formatting choices: Figure 7 uses an overly small x-axis font, and the legend in Figure 5(c) is too small to be easily readable. Moreover, the tables in the appendix lack typographic emphasis.
6.	For practical online detection scenarios, the latency for detecting a single text instance is a key concern. The paper claims that avoiding expensive per-instance regeneration is one of its main motivations, which is reasonable in principle. However, Figure 6 shows that SurpMark only begins to amortize its cost advantage after processing around 300 samples in batch mode. This implies that for single-instance detection, SurpMark is likely slower than Fast-DetectGPT. Therefore, the paper’s central motivation—achieving superior computational efficiency over regeneration-based methods—is not convincingly substantiated by the current experimental design.
7.	In Section 3, the paper cites Khandelwal et al. (2018)[C] to justify that “LM predictions rely mainly on short range context,” thereby motivating the use of a first-order Markov model. This claim is no longer persuasive in the era of large-scale LLMs such as LLaMA 3 and GPT-5, whose predictions depend heavily on extended context. The authors should consider weakening or removing this outdated reference and instead emphasize the empirical evidence presented in Figure 2(b) as the primary justification for adopting a first-order approximation.

**Questions:**

1.	The paper does not discuss how to set the tunable threshold τ, which is crucial for practical deployment. Could the authors clarify how τ should be chosen (e.g., via validation or theoretical guidance) and provide empirical analysis on its impact on detection performance?
2.	The empirical validation of the asymptotic normality of the GJS statistic, while visually supported by histograms in Figure 2(c) and Figure 7, would be strengthened with formal statistical tests for normality. Could the authors provide quantitative assessments such as Shapiro-Wilk test results or other normality metrics to more rigorously confirm the distributional assumptions?
3.	The paper does not specify the length distribution of the human and machine reference texts used to build the transition matrices, nor does it clarify whether short or low-quality references were filtered out during preprocessing. Could the authors provide details on how reference text lengths were controlled and whether a minimum length threshold was applied to ensure reliable estimation of the Markov transition statistics?
4.	Table 1 shows that the proposed method exhibits a considerable advantage on closed-source models, while Table 2 indicates much smaller gains on open-source ones. This is a noteworthy phenomenon. Could there be deeper underlying factors—such as tokenization mismatches, proxy-model bias, or domain differences—that explain this disparity?
5.	Section 5.2, labeled as “Ablation Studies” is more accurately a hyperparameter sensitivity analysis. A proper ablation study should explicitly answer the following questions:
(1)	Necessity of the Markov chain: If the transition matrix is removed and only the stationary distribution of surprisal states (i.e., the 1-gram distribution, a $k$-dimensional vector) is used as a feature, how much does performance degrade?
(2)	Necessity of the GJS divergence: If the Generalized Jensen–Shannon (GJS) divergence is replaced by a simpler distance metric such as the $L_1$ or $L_2$ norm, what is the performance impact?
(3)	Necessity of k-means quantization: If k-means clustering is replaced by simpler schemes such as equal-width or equal-mass binning, how does performance change?

[A] Deep Kernel Relative Test for Machine-generated Text Detection. ICLR 2025.

[B] Detecting machine-generated texts by multi-population aware optimization for maximum mean discrepancy. ICLR 2024.

[C] Sharp Nearby, Fuzzy Far Away: How Neural Language Models Use Context.

---

> ### Author Response · Authors · 2025-11-20
> **Response to Weakness 1 and 2**
>
> We would like to thank the reviewer for the time and constructive comments.
>
> **W1: Potential Bias in Figure 2b**
> We apologize for not providing enough background on how Figure 2(b) was constructed. All AUROC values in Figure 2(b) are computed using the same amount of reference and test data. As in our main experiments, we use 300 human paragraphs and 300 machine-generated paragraphs, each with length about 100-200 tokens as reference. We use 150 human paragraphs and 150 machine paragraphs as test data. Intuitively, increasing the Markov order makes the state space explode while the amount of reference is fixed, so transition estimates become very sparse and noisy.
>
> The degradation with larger order is a variance / sparsity effect that arises both on the reference side and on the test side: (i) **State-space explosion**: A first-order chain with $k$ bins has $k^2$ transitions; an order-$l$ chain effectively has $k^{l+1}$ transitions. With only 300 human + 300 machine paragraphs of 100-200 tokens, many high-order contexts in the reference data are observed only a few times or not at all, so the estimated transitions become extremely noisy. (ii) **Short test sequences**: Each test paragraph is itself only 100-200 tokens long. Even if the reference transitions were perfectly estimated, an order-$l$ model on a 100-200-token sequence can observe only a very small number of distinct $l$-length contexts. The higher-order model is severely under-sampled on each individual test example.
>
> To isolate the effect of reference sparsity, in the XSum@GPT-J-6B dataset, we fixed $k=6$ and the test set (150 human + 150 machine paragraphs, 100-200 tokens each), and increased the reference size from 300 to 1200 paragraphs per side. AUROC for higher-order models improves only slightly and remains clearly below the first-order model.
>
> | Order \ Reference size | 300   | 600   | 900   | 1200  |
> |------------------------|-------|-------|-------|-------|
> | 1                      | 86.51 | 86.17 | 86.69 | 86.89 |
> | 2                      | 81.49 | 82.08 | 82.77 | 82.80 |
> | 3                      | 74.98 | 74.74 | 75.84 | 75.85 |
> | 4                      | 64.72 | 69.51 | 71.20 | 73.00 |
>
> To further evaluate the effect of test sparsity, in another WritingPrompts@Genmma-7B dataset, we vary the test passage length from 150 to 300 tokens while keeping the reference size fixed (300 passages per side, each with fixed 300 tokens). AUROC consistently increases for all orders, but the first-order model remains clearly best, and higher orders still lag behind by several points.
>
> | Order \ Test Length | 150   | 200   | 250   | 300   |
> |----------------|-------|-------|-------|-------|
> | 1              | 90.59 | 92.66 | 94.60 | 94.58 |
> | 2              | 89.42 | 90.67 | 92.21 | 92.60 |
> | 3              | 88.67 | 90.57 | 91.40 | 91.40 |
> | 4              | 62.29 | 65.59 | 68.46 | 68.33 |
>
> Taken together, these ablations reflect practical text-detection settings with limited reference data and short passages. In this regime, the first-order model offers the best bias-variance tradeoff, so we believe it is the most reasonable default choice.
>
> **W2: Inconsistent Definition and Use of the Symbol "N"**
> We thank the reviewer for pointing out this notational ambiguity. In the paper, $N$ is intended to denote the total number of reference transitions. We will introduce separate symbols, $m$ for the number of reference sequences, $\{l_j\}$ for their length, and $N=\sum_{j=1}^{m}(l_j-1)$. In experiments, both the number of references and the per-reference length contribute to the total transitions, so we separately study their effects. We will revise the paper to make this notation fully consistent.

---

> ### Author Response · Authors · 2025-11-20
> **Response to Weakness 3 to 7**
>
> **W3: Comparison with R-Detect**
> We have run a head-to-head comparison of R-Detect and SurpMark under exactly the same setup as the main table (same reference sizes, test sets); see the table below. We report six generators across datasets, where SurpMark outperforms R-Detect in 17 out of 18 settings, often by a substantial margin. In the revision, we will merge all the results into the main table.
>
> Regarding computational cost, under the same main-table setup the empirical throughput of R-Detect is 1.04 items/s, whereas SurpMark achieves 12.45 items/s. Importantly, for R-Detect we did not include any kernel-training time, since we directly used the checkpoint provided in its official repository. Even under this favorable assumption for R-Detect, SurpMark remains dramatically more efficient.
>
> | Method        | | GPT-5-chat      |  |  | GPT-4.1-mini           | | | Gemini-1.5-Flash              |  |
> |:-------------|:----------:|:---------------:|:----------:|:------------:|:----------------------:|:------------:|:----------------:|:-----------------------------:|:----------------:|
> |              | XSum       | WritingPrompts  | SQuAD      | XSum         | WritingPrompts         | SQuAD        | XSum             | WritingPrompts                | SQuAD            |
> | R-Detect     | 58.74      | 77.06           | 67.45      | 63.43        | 78.79                  | 72.69        | 63.58            | 83.31                        | 60.86            |
> | SurpMark k=7 | 87.02      | 83.73           | 73.23      | 82.52        | 83.64                  | 72.27        | 71.22            | 86.68                        | 67.51            |
>
>
> | Method        | | Gemma-7b       | | | Llama2-13b      | |  | Llama3-8b       |  |
> |:-------------|:--------:|:--------------:|:--------:|:----------:|:---------------:|:----------:|:---------:|:---------------:|:---------:|
> |              | XSum     | WritingPrompts | SQuAD    | XSum       | WritingPrompts  | SQuAD      | XSum      | WritingPrompts  | SQuAD     |
> | R-Detect     | 46.75    | 72.62          | 47.02    | 72.11      | 79.69           | 88.15      | 81.06     | 82.79           | 81.06     |
> | SurpMark k=7 | 92.97    | 90.37          | 68.27    | 97.09      | 99.53           | 96.13      | 97.48     | 97.48           | 94.17     |
>
>
>
>
>
> **W4: Cross-Domain Generalization**
>
> We have evaluated this scenario in table below. We compared self-ref (in-domain reference) to Out-Of-Domain (OOD) reference (associate with the corresponding generator). Even when we deliberately use out-of-domain reference corpora to estimate transition probabilities, the impact is small: across all 24 configurations, AUROC changes by only 1.16 pp on average, with a worst-case deviation of 4.65 pp. Changes remain moderate and can even be positive sometimes.
>
> | Test | self-ref | WritingPrompts-as-ref | XSum-as-ref | SQuAD-as-ref |
> |:--|:--:|:--:|:--:|:--:|
> | Xsum@Llama2-13b | 97.09 | 97.22 | - | 96.45 |
> | WritingPrompts@Llama2-13b | 99.53 | - | 99.71 | 99.33 |
> | SQuAD@Llama2-13b | 96.13 | 94.36 | 95.67 | - |
> | XSum@Llama3-8b | 97.09 | 97.25 | - | 95.13 |
> | WritingPrompts@Llama3-8b | 99.86 | - | 99.95 | 99.88 |
> | SQuAD@Llama3-8b | 94.17 | 93.77 | 93.9 | - |
> | XSum@Gemma_7b | 72.97 | 68.32 | - | 73.81 |
> | WritingPrompts@Gemma_7b | 89.42| - | 86.6 | 90 |
> | SQuAD@Gemma_7b | 68.27 |  72.92 | 70.44 | - |
> | XSum@GPT-Neox-20b | 82.45 | 82.52 | - | 81.8 |
> | WritingPrompts@GPT-Neox-20b | 94.31| - | 93.45 | 92.48 |
> | SQuAD@GPT-Neox-20b | 82.45 | 81.43 | 83.02 | - |
>
>
> **W5: Issues with Figure and Table Formatting** We will replace all plots with cleaner vector graphics and improve the appendix table formatting in the revision.
>
> **W6: Latency Analysis**
>
> We would like to clarify that the setup in Figure 6 is actually unfavorable to SurpMark. In that plot we deliberately include the one-time reference construction cost in the runtime, even though in any realistic deployment the reference corpus is fixed and can be built offline and reused. By contrast, regeneration-based detectors incur an online per-instance cost that cannot be moved offline. Our claim about real-time behaviour is specifically about the online phase once reference construction is ready. Under this setting, SurpMark operates at 31.26 items/s, compared to 12.61 item/s for Fast-DetectGPT.
>
>
> **W7: Outdated Reference**
>
> In the revision, we will remove this citation and make Figure 2(b) and our ablation results (in response to W1) the main justification for a first-order Markov approximation.

---

> ### Author Response · Authors · 2025-11-20
> **Response to Question 1 and 2**
>
> **Q1: Threshold** We thank the reviewer for raising this point. The natural decision rule is simply the sign test by setting $\tau=0$. Our detector is built around the difference between two GJS divergences. Intuitively, $\Delta GJS$ is positive when the test sequence is closer to the machine reference than to the human reference, and negative in the opposite case. Also, $\Delta GJS$ can be viewed as a log-likelihood ratio $\Lambda_{n,N}$. In the classical Neyman-Pearson framework, the optimal likelihood-ratio test with equal class priors and symmetric costs is precisely $\Lambda_{n,N} \gtrless 0$. To address the reviewer's request for empirical analysis, we additionally perform a threshold sensitivity study. For each dataset and generator, we sweep $\tau$ over the full score range on the test set, compute precision/recall, and identify an optimal threshold $\tau^*$ that maximizes F1. We then compare F1 at our fixed choice $\tau=0$. Across all generators and datasets, F1 at $\tau=0$ is typically about 95-97% of the oracle F1. This shows that in practice, our parameter-free sign-based rule already operates very close to the best threshold.
>
> | Setting                     | AUROC | τ*        | F1@τ* | F1@τ=0 |
> |-----------------------------|:-----:|:---------:|:-----:|:------:|
> | XSum@GPT-J-6B              | 89.12 | 2.92E-05  | 83.56 | 80.36  |
> | WritingPrompts@Llama-2-13B | 99.75 | -9.29E-06 | 98.66 | 98.66  |
> | SQuAD@Llama-3-8B           | 93.56 | -4.49E-05 | 87.58 | 82.69  |
> | WritingPrompts@GPT-5-chat  | 80.63 | -1.34E-05 | 76.13 | 75.07  |
>
> In Lastde[1], the authors propose a fixed threshold of 2 for Lastde++ regardless of the source model, motivated by plotting score distributions and empirical performance across their experiments. We therefore compare F1 of two methods at their respective threshold. Across three of the four settings, SurpMark achieves higher AUROC, and in all four settings it attains a higher F1. On SQuAD@Llama-3-8B, Lastde++ has slightly higher AUROC, but at their fixed thresholds SurpMark still achieves higher F1, indicating SurpMark's sign-based decision rule is better calibrated and less sensitive to threshold choice.
> | Metric                 | Method        | XSum@GPT-J-6B | WritingPrompts@Llama-2-13B | SQuAD@Llama-3-8B | WritingPrompts@GPT-5-chat |
> |------------------------|--------------|---------------|-----------------------------|------------------|---------------------------|
> | AUROC                  | Lastde++     | 85.38         | 99.14                       | 94.72            | 30.64                     |
> |                        | SurpMark k=6 | 88.35         | 99.47                       | 93.76            | 82.25                     |
> | F1 at respective fixed threshold  | Lastde++     | 63.44         | 95.56                       | 80.93            | 0                         |
> |                        | SurpMark k=6 | 80.36         | 98.66                       | 82.69            | 75.07                     |
>
> [1] Training-free LLM-generated Text Detection by Mining Token Probability Sequences. Xu et al. ICLR25
>
> **Q2: Statistical Tests to Validate Asymptotic Normality**
>
> We have added quantitative normality diagnostics for $\Delta GJS$. For representative settings (SQuAD@GPT-5-chat, WritingPrompts@Llama3-8B, XSum@Qwen3-8B), under each hypothesis, we run detection and get 150 scores, and apply Shapiro-Wilk test.  The results are strongly consistent with normality, so we fail to reject normality at the 5% level. These empirical results are consistent with the asymptotic normality predicted by Theorem 4.4 in the finite-sample regime we study.
>
> | Setting          | SQuAD@GPT-5-chat | WritingPrompts@Llama3-8B | XSum@Qwen3-8B |
> |------------------|-----------------:|--------------------------:|--------------:|
> | **H1 (LM text)** stat    | 0.9952            | 0.9856                   | 0.9974       |
> | **H1 (LM text)** p-value | 0.9078            | 0.1203                   | 0.9969       |
> | **H0 (human text)** stat    | 0.9876            | 0.9854                   | 0.9929       |
> | **H0 (human text)** p-value | 0.2032            | 0.1143                   | 0.6632       |

---

> ### Author Response · Authors · 2025-11-20
> **Response to Question 3 and 4**
>
> **Q3: Reference Text Length Control**
>
> Our generation pipeline are the same for both reference and test data. In the generation pipeline, for each dataset, we filtered out samples with text length less than 150 words and always condition only on the first 30 tokens of the human text. Unleass otherwise specified (Fig. 4(b,c), Fig. 5(a,b)), each machine passage is generated between 100 and 200 tokens. After generation, we pair each human passage with its corresponding machine passage and **truncate both to the shorter side** (measured in words). Thus every human-machine pair used for detection has the same length and there is no systematic length advantage for either class. This processing follows the evaluation setup of Lastde [1]. We will add the description in the revision.
>
> To further determine a minimum length threshold, we extended the results in Fig. 4(b). We  control both reference and test lengths via longer generations and truncation. For each (ref length,test length), we varied the number of bins k and reported the resulting AUROC range. We observe a large gain when increasing the minimum reference length from 50 to 75-100 tokens, while further increasing it beyond 100 yields only modest improvements with largely overlapping AUROC ranges. Thus, if resources are very limited, a minimum reference length of 75 tokens already performs close to longer references, and a more conservative choice of 100 tokens provides stable estimation of the Markov transition statistics.
>
> | Ref length ↓ / Test length → | 100            | 150             | 200             | 250             |
> |--------------------------------------|----------------|-----------------|-----------------|-----------------|
> | 50                                   | [74.55, 80.52] | [84.00, 85.27]  | [85.97, 87.92]  | [86.87, 89.03]  |
> | 75                                   | [80.98, 83.74] | [86.09, 89.48]  | [87.86, 91.07]  | [89.18, 92.66]  |
> | 100                                  | [78.43, 85.12] | [85.39, 91.44]  | [88.31, 92.44]  | [90.16, 93.73]  |
> | 200                                  | [77.47, 83.89] | [84.89, 90.76]  | [88.03, 92.49]  | [90.18, 94.45]  |
>
>
> [1] Training-free LLM-generated Text Detection by Mining Token Probability Sequences. Xu et al. ICLR25
>
> **Q4: Performance Disparity on Closed-Source vs. Open-Source Models**
> Our analysis suggests that the key difference is how distinguishable each generator is from human text at the marginal surprisal level. For some models (mostly the open-source ones we study), the marginal surprisal gap is already large, so marginal-surprisal-based detectors (Likelihood, LogRank, DetectGPT, Fast-DetectGPT, etc.) naturally perform very well and SurpMark offers only modest additional gains. When the marginal gap is small (e.g., GPT-5-chat), a simple unigram model is insufficient and first-order modeling of surprisal dynamics provides a much larger improvement, which explains the stronger relative advantage of our method on these closed-source models. We support this interpretation with Jensen Shannon divergence summaries for marginal surprisal and first-order transitions, reported in the table below.
>
> | Generator    | Dataset | JS-marginal      | JS-transition | Ratio(Transition / Marginal) |
> |-------------|---------|-------------------|----------------|------------------------|
> | GPT-J-6B    | XSum    | 0.0018           | 0.00228        | ≈1.27                   |
> | GPT-J-6B    | SQuAD   | 0.00358           | 0.00392        | ≈1.09                   |
> | GPT-5-chat  | XSum    | 0.0000567         | 0.00170        | ≈ 29.97                |
> | GPT-5-chat  | SQuAD   | 0.00024           | 0.00100        | ≈ 4.17                 |
> | GPT-4.1-mini| XSum    | 0.00030           | 0.00160        | ≈ 5.33                 |
> | GPT-4.1-mini| SQuAD   | 0.00052           | 0.00150        | ≈ 2.88                 |

---

> ### Author Response · Authors · 2025-11-20
> **Response to Question 5**
>
> **Q5: Key Ablation Studies**
> We thank the reviewer for this helpful clarification. We will rename Sec. 5.2 to “**Ablation and Sensitivity Analysis**“. Then, we have conducted the three requested experiments:
>
> (1) **Necessity of the Markov chain**
>
> Across all datasets, the first-order Markov features outperform the 1-gram distribution, with especially large gains on GPT-5-chat. This shows that modeling surprisal transitions, rather than only the stationary distribution, is particularly important for harder-to-detect models.
> |                          | GPT-J-6B                      |                          |                          | GPT-5-chat                    |                          |                          |
> |--------------------------|------------------------------:|-------------------------:|-------------------------:|------------------------------:|-------------------------:|-------------------------:|
> | Metric / Dataset         | XSum        | WritingPrompts | SQuAD    | XSum        | WritingPrompts | SQuAD    |
> | 1-gram distribution      |       86.07 |          96.60 |    91.62 |       55.89 |          78.43 |    54.58 |
> | First-order Markov chain |       88.35 |          97.60 |    92.93 |       84.16 |          82.25 |    68.57 |
>
>
> (2) **Necessity of GJS**
>
> GJS achieves the best AUROC on most dataset and source model. This suggests that GJS is a more robust similarity measure than L1/L2.
> |                    | GPT-5-chat                          |                           |                           | GPT-4.1 mini                      |                           |                           |
> |--------------------|-------------------------------------|---------------------------|---------------------------|-----------------------------------|---------------------------|---------------------------|
> | Metric / Dataset   | XSum                                | WritingPrompts            | SQuAD                     | XSum                              | WritingPrompts            | SQuAD                     |
> | GJS                | 87.02                          | 83.73               | 73.23               | 82.52                       | 83.64              | 69.27               |
> | L1                 | 77.18                        | 84.79               | 60.08               | 73.51                      | 82.17               | 62.28               |
> | L2                 | 77.76                        | 83.96               | 59.92               | 73.58                       | 83.04              | 59.14               |
>
>
> (3) **Necessity of k-means**
>
> Across all datasets and k values, k-means is the most robust quantization scheme: it consistently reaches or matches the best AUROC, while equal-mass can degrade sharply on XSum@GPT-4.1-mini and equal-width is unstable and often much worse.
>
> | Dataset@Source model          | Method              | k = 6 | k = 7 | k = 8 | k = 9 | best|
> |------------------|---------------------|-------|-------|-------|-------|-----|
> | XSum@GPT4.1-mini      | k-means             | 77.43 | 80.42 | 79.32 | 81.79 | 81.79 |
> | XSum@GPT4.1-mini     | equal-width binning | 80.28 | 80.41 | 76.2  | 78.16 | 80.41|
> | XSum@GPT4.1-mini     | equal-mass binning  | 71.69 | 71.4  | 74.22 | 75.2  |75.2 |
> | WritingPrompts@GPT5-chat| k-means             | 80.63 | 82.05 | 84.86 | 83.88 | 84.86 |
> | WritingPrompts@GPT5-chat| equal-width binning | 73.91 | 62.9  | 72.06 | 71.08 | 73.91 |
> | WritingPrompts@GPT5-chat| equal-mass binning  | 82.96 | 84.42 | 83.59 | 84.52 |84.52 |
> | SQuAD@Gemini     | k-means             | 72.64 | 77.74 | 72.56 | 75.01 |77.74|
> | SQuAD@Gemini     | equal-width binning | 60.36 | 58.21 | 54.87 | 56.85 |60.36|
> | SQuAD@Gemini     | equal-mass binning  | 72.93 | 75.08 | 73.65 | 73.41 |75.08|

---

### Official Review · Reviewer_ex8a · 2025-10-29

**Soundness:** 2
**Presentation:** 2
**Contribution:** 3
**Rating:** 4
**Confidence:** 3

**Summary:**

The paper proposes SurpMark, a novel, reference-based method for detecting machine-generated text. Instead of relying on brittle, absolute surprisal values, it models the dynamics of surprisal. This paper suggests quantizing surprisals into $k$ states and modeling the text as a first-order Markov chain of these states.

**Strengths:**

The authors propose a novel framework to learn state transitions of token surprisal as a first-order Markov chain. It cleverly abstracts the detection signal away from the specific tokens to state transitions internally within LLMs.

The propositions defined in the paper are also very extensive and highly appreciated.

**Weaknesses:**

- The theoretical justification for the core framework (i.e. L178, which cites Khandelwal et al.), which relies on a first-order Markov chain assumption, is outdated. The paper cited only discusses older architectures such as LSTMs, and it does not align with the assumption stated here with LLMs.
-  In Theorem 4.2, the assumption $\pi_{\min} \gtrsim 1/k$ appears questionable. The surprisal distribution is rarely uniform in practice, making this assumption unrealistic and invalidating the proof in the Appendix.
- This indirectly casts doubt on the validity of the subsequent theorem regarding the optimal $k$. The authors mention that they compute this theoretical optimum and then fine-tune around it; however, it would be helpful to report whether the experimentally observed optimum diverges significantly from the theoretical value. (Some later experiments in Section 5.2 do show some divergence.)
- The choice of K-means clustering requires an ablation study. K-means is sensitive to outliers, which is particularly problematic in the stated settings. Additionally, are the interpretive labels such as "Very predictable" validated or grounded in any sort of analysis or were these labels assigned by the authors?
- The method relies on a Markov model to enhance detection - which is very prone to adversarial attacks (apart from just paraphrasing). It would be valuable to evaluate the robustness of this framework under simple prompt-engineered adversarial attacks.

**Questions:**

- Although mathematically sound, can the authors intuitively explain the GJS divergence equation (above Equation 2)?

All the questions and suggestions have been listed in the weaknesses. I am willing to increase my score if the authors address these concerns.

---

> ### Author Response · Authors · 2025-11-20
> **Response to Weakness 1 to 3**
>
> Thank you for reviewing our paper and providing valuable feedback.  We hope the following answers have cleared up any doubts.
>
> **W1: On the First-Order Markov Approximation for Surprisal States**
> We agree that our citation to Khandelwal et al. (2018) is not the most appropriate reference for modern transformer-based LLMs. We will remove the sentence that cites Khandelwal et al.
>
> Instead, we will clarify that the first-order Markov chain appears only as a simple abstraction of the discretized surprisal state sequence in our theoretical analysis, and that its adequacy is supported by our own empirical results (Fig. 2(b)), and a new ablation showing that higher-order chains suffer from data sparsity and degraded AUROC under limited reference data (see response to Q1 of Reviewer gbAS for detailed discussion).
>
>
> **W2: Assumption in Theorem 4.2** In our notation, $\pi_{\min} \gtrsim 1/k$ means  $\pi_{\min} \ge c/k$ for some absolute constant $c>0$. $\pi_{\min}$ is the minimum stationary mass over the discretized surprisal states. This is a standard non-degeneracy condition that rules out vanishingly rare states; it does not assume that the surprisal (or the discretized states) are uniformly distributed. The distribution can still be quite skewed.
>
> Technically, at line 1174, $\pi_{\min}$ appears in the denominator when we express the number of visits to a state $s$ in terms of its stationary mass, leading to constants of the form $1/\sqrt{\pi_{\min}}$. Thus, we indeed require $\pi_{\min}$ to be bounded away from 0 to keep the constants finite and meaningful.
>
> The numbers in the table are computed from the empirical 1-gram distribution of quantized surprisal states on the reference corpus, as a practical approximation of stationary distribution. So $\pi_{\min}$ is indeed on the order of 1/k rather than being extremely small, suggesting the assumption on $\pi_{\min}$ is reasonable.
>
> | Empirical $\pi_{\min}^{human}$ or  $\pi_{\min}^{LM}$      | k = 6  | k = 7  |
> |--------------------------|--------|--------|
> | WritingPrompts@human     | 0.098  | 0.065  |
> | WritingPrompts@Qwen3-8B  | 0.029  | 0.019  |
> | XSum@human               | 0.088  | 0.064  |
> | XSum@GPT-neox-20B        | 0.061  | 0.043  |
>
> **W3: Theoretical vs. Experimental Optimal k** From table below, the empirically selected $k$ tracks the theoretical recommendation $k^* ∝ N^{1/5}$ very closely. The ratio stays close to a constant 0.8, and the empirical optimum differs from the rounded theoretical value by 1-2 bins. Moreover, we do observe a sweet spot region of k in Fig.3. In the sweet spot region, the AUROC-vs.-$k$ curve is relatively flat, so these small shifts have negligible impact on detection performance.
>
>
> | Number of ref samples | $N$ (approximate total transitions) | Empirical best $k$ | $N^{1/5}$ | $\frac{k}{N^{1/5}}$    |
> |-------------------|------------------------------------|------------------|---------------------|------|
> | 100               | 15,000                             | 6                | 6.84                | 0.88 |
> | 300               | 45,000                             | 7                | 8.52                | 0.82 |
> | 400               | 60,000                             | 7                | 9.03                | 0.78 |
> | 600               | 90,000                             | 7                | 9.80                | 0.71 |
> | 900               | 135,000                            | 9                | 10.62               | 0.85 |

---

> ### Author Response · Authors · 2025-11-20
> **Response to Weakness 4**
>
> **W4: K-means Ablation Study and Validation of Interpretive Labels**
> We have now added an explicit ablation comparing k-means against equal-width and equal-mass binning. Across all datasets and k values, k-means is the most robust quantization scheme: it consistently reaches or matches the best AUROC, while equal-mass can degrade sharply on XSum@GPT-4.1-mini and equal-width is unstable and often much worse.
>
> Regarding the interpretive labels, these are only used for visualization and intuition, not by the detector itself. Concretely, surprisal is defined as negative log likelihood, so low surprisal corresponds to high model likelihood (tokens that the proxy finds very predictable), and high surprisal corresponds to low likelihood (tokens that are surprising). Our labels simply reflect this  relationship. We will clarify in the revision that these names are descriptive labels grounded in the proxy model's log-likelihood and do not affect the detector's computation in either the reference calibration, scoring, or the theoretical analysis.
>
> | Dataset@Source model          | Method              | k = 6 | k = 7 | k = 8 | k = 9 | best|
> |------------------|---------------------|-------|-------|-------|-------|-----|
> | XSum@GPT4.1-mini      | k-means             | 77.43 | 80.42 | 79.32 | 81.79 | **81.79** |
> | XSum@GPT4.1-mini     | equal-width binning | 80.28 | 80.41 | 76.2  | 78.16 | 80.41|
> | XSum@GPT4.1-mini     | equal-mass binning  | 71.69 | 71.4  | 74.22 | 75.2  |75.2 |
> | WritingPrompts@GPT5-chat| k-means             | 80.63 | 82.05 | 84.86 | 83.88 | **84.86** |
> | WritingPrompts@GPT5-chat| equal-width binning | 73.91 | 62.9  | 72.06 | 71.08 | 73.91 |
> | WritingPrompts@GPT5-chat| equal-mass binning  | 82.96 | 84.42 | 83.59 | 84.52 |84.52 |
> | SQuAD@Gemini     | k-means             | 72.64 | 77.74 | 72.56 | 75.01 |**77.74**|
> | SQuAD@Gemini     | equal-width binning | 60.36 | 58.21 | 54.87 | 56.85 |60.36|
> | SQuAD@Gemini     | equal-mass binning  | 72.93 | 75.08 | 73.65 | 73.41 |75.08|

---

> ### Author Response · Authors · 2025-11-20
> **Response to Weakness 5 and Question 1**
>
> **W5: Robustness Against Prompt-Engineered Adversarial Attacks** We have added explicit experiments with simple prompt-engineered attacks beyond plain paraphrasing. Specifically, for the XSum and WritingPrompts datasets, we design two types of attacks: (**attack 1**) prompts that ask the model to **mimic human writing style**, using instructions such as “Messy casual summary of the news article.” or “Short story in a quick, slightly messy human style.”; and (**attack 2**) prompts that explicitly **instruct the model to evade detection**, such as “Write a summary of the article that is designed to evade AI-text detectors.” or “Continue the story in a way that is hard for AI-text detectors.” See table below for comparison. “SurpMark ref-attack” applies the adversarial prompts only when generating the reference machine texts, “SurpMark test-attack” applies them only to the test texts, and “SurpMark both-attack” applies the same adversarial prompts to both the reference and test texts. Across both datasets, SurpMark variants (especially the test-attack and both-attack settings) experience much smaller accuracy drops under all three attacks, showing the strongest overall robustness.
>
> | Method                | WritingPrompts@GPT-J-6B Original |  Attack 1                       |  Attack 2                        | XSum@GPT-J-6B Original |  Attack 1                               |  Attack 2                                |
> |-----------------------|----------------------------------|-----------------------------------------------|-----------------------------------------------|------------------------|---------------------------------------------|-----------------------------------------------|
> | Fast-DetectGPT        | 93.8                             | 85.95 ($\downarrow$ 7.85)                     | 79.26 ($\downarrow$ 14.54)                    | 78.6                   | 75.44 ($\downarrow$ 3.16)                   | 74.09 ($\downarrow$ 4.51)                    |
> | Lastde++              | 96.96                            | 84.24 ($\downarrow$ 12.72)                    | 85.42 ($\downarrow$ 11.52)                    | 85.38                  | 69.79 ($\downarrow$ 15.59)                  | 73.55 ($\downarrow$ 11.83)                   |
> | SurpMark ref-attack   | 97.6                             | 95.06 ($\downarrow$ 2.54)                     | 94.67 ($\downarrow$ 2.93)                     | 88.35                  | 83.84 ($\downarrow$ 4.51)                   | 83.85 ($\downarrow$ 4.5)                     |
> | SurpMark test-attack  | 97.6                             | 95.62 ($\downarrow$ 1.98)                     | 92.59 ($\downarrow$ 5.01)                     | 88.35                  | 86.37 ($\downarrow$ 1.98)                   | 84.86 ($\downarrow$ 3.49)                    |
> | SurpMark both-attack  | 97.6                             | 94.3 ($\downarrow$ 3.3)                       | 92.74 ($\downarrow$ 4.86)                     | 88.35                  | 84.44 ($\downarrow$ 3.91)                   | 85.23 ($\downarrow$ 3.12)                    |
>
>
> **Q1: Intuitive Explanation of GJS Divergence**
>
> In our binary hypothesis testing setting, $GJS(\hat{M}_P, \hat{M}_T, \alpha)$ measures how much the empirical Markov estimators fitted to the reference and test sequences prefer the hypothesis that they share a single source over the hypothesis that the reference and test sequences come from two different Markov sources. A larger value means that it is much easier to explain the two sequences by fitting separate models to them, so they are easier to distinguish.
>
> More formally, quantities such as $GJS(\hat{M}_P, \hat{M}_T, \alpha)$ have a direct **log-likelihood-ratio interpretation**. Using the definitions in Proposition 4.3, we can write
>
> $GJS(\hat{M}_P, \hat{M}_T, \alpha)$
>
> = $\frac{1}{n} \log \frac{\sup_{M_1 \in F_k} M_1((a^P_{1:N},a^T_{1:n}))}{\sup_{M_2,M_3 \in F_k}M_2(a^P_{1:N})M_3(a^T_{1:N})}$
>
> where the numerator is the maximum likelihood of the concatenated reference and test sequences under a single source, and the denominator is the maximum joint likelihood when the two sequences are generated by two independent sources. The detailed proof is very similar to the proof of Proposition 4.3. We rewrite the statistic as a log-likelihood ratio in terms of the empirical Markov transition frequencies of the reference and test sequences and then analyze its large-sample limit using a standard tool from information theory (the “method of types”, which studies likelihoods through empirical distributions).

---

> > ### Comment · Reviewer_ex8a · 2025-11-22
> > **Response to Authors**
> >
> > I highly appreciate the author's responses and the rigorous experiments provided. I still have a few concerns.
> >
> > - I'm not very convinced behind the reasoning of why a first-order Markov chain is used. This is similar to Reviewer `gbAS`'s concern and I will continue to follow this discussion. If the authors can provide a suitable justification (apart from empirical evidence), it would be appreciated.
> > - One final experiment and metric I would request the authors to add into the paper is the inclusion of TPR@FPR=5% and 1% with some baselines.
> >
> > A minor question out of curiosity: I wonder how $\Delta$GJS has evolved from older to newer models and whether text generation has fundamentally become harder.
> >
> > ---
> >
> > I appreciate the experiments provided to the other reviewers as well, since they resolve other minor concerns I had.

---

> > > ### Author Response · Authors · 2025-11-26
> > > **Round 2 Response (2/2)**
> > >
> > > **TPR**
> > > In the revision, we will add TPR@FPR=1% and 5% for SurpMark and two strong baselines (Lastde++ and Fast-DetectGPT) across all seven evaluation settings. Overall, these results indicate that SurpMark is particularly effective in the low-false-positive regime.
> > > | Method          | FPR | XSum@GPT-5-Chat | WritingPrompts@GPT-4.1-mini | XSum@Llama2-13B | SQuAD@GPT-Neo-2.7B | WritingPrompts@Llama3-8B | HC3-Chi-Psy |HC3-Chi-QA | Avg
> > > |----------------|-----|-----------------|------------------------------|-----------------|--------------------|---------------------------|------------|------------|-----------|
> > > | Lastde++       | TPR@FPR=1%  | 4.00            | 6.00                         | 76.67           | **53.33**              | 97.33                     | 22.00      | 38.00     | 42.48|
> > > | Lastde++       | TPR@FPR=5%  | 12.67           | 18.67                        | 82.67           | 81.33              | 99.33                     | 31.33      | 65.33     |55.9|
> > > | Fast-DetectGPT | TPR@FPR=1%  | 2.00            | 3.30                         | **80.67**           | 47.33              | 94.67                     | 26.00      | 44.00     |42.57|
> > > | Fast-DetectGPT | TPR@FPR=5%  | 4.00            | 22.00                        | 86.67           | 77.33              | 98.00                     | 40.00      | 60.00     |55.43|
> > > | SurpMark       | TPR@FPR=1%  | **31.33**           | **31.33**                        | 75.33           | 41.33              | **100.00**                    | **90.00**      | **74.67**     |**63.43**
> > > | SurpMark       | TPR@FPR=5%  | **37.33**           | **50.00**                        | **90.00**           | **90.00**              | **100.00**                    | **97.33**      | **97.33**     | **80.28**|
> > >
> > >
> > > **Whether text detection become harder** We interpret your question as asking whether text detection has become fundamentally harder on newer models. As shown in the table, older open-source generators (GPT-J-6B and Gemma-7B) have slightly negative mean values, whereas GPT-5-chat shows a slightly positive mean. The magnitude remains in the same order ($10^{-5}$), but the sign change indicates that the statistic becomes less polarized between human and machine samples for GPT-5-chat. In other words, the gap that $\Delta \mathrm{GJS}$ exploits shrinks for newer models, so distinguishing GPT-5-chat from human text is statistically harder, even though the signal does not disappear.
> > >
> > > | Model      | mean ΔGJS ($\times$1e-5) |
> > > | ---------- | ----------------- |
> > > | GPT-J-6B   | -3.34             |
> > > | Gemma-7B   | -2.10             |
> > > | GPT-5-chat | +2.22             |
> > >
> > > We also provide a related discussion of whether detection becomes harder on newer closed-source models in our response to reviewer EQL7 under Q4 (“Performance Disparity on Closed-Source vs. Open-Source Models”). In short, we argue there that the main driver of this disparity is the size of the marginal surprisal gap: for (mostly) open-source models this gap is already large so likelihood-based detectors perform well and SurpMark brings only modest gains, whereas for newer closed-source models such as GPT-5-chat the marginal gap is small and first-order surprisal dynamics yield a substantial advantage.

---

> ### Author Response · Authors · 2025-11-26
> **Round 2 Response (1/2)**
>
> **On First-order Markov chain** We clarify our reasoning by (i) starting from Gray's Markov approximation theory, (ii) explaining how the gain from order
> $K$ to $K+1$ is governed by conditional mutual information, (iii) mapping this theory to our discretized surprisal process, and (iv) presenting empirical measurements showing that the additional benefit of a second-order approximation over a first-order one is very small.
>
> (1) Best finite-order Markov approximation in Gray's theory. Following Gray's Entropy and Information Theory [1, Sec. 6.4, Cor. 6.4.1–6.4.2; Sec. 7.4, Cor. 7.4.2–7.4.3], consider a stationary discrete-time source $\{X _ n\}$. Gray constructs, for each order $K$, a canonical $K$-th order Markov chain $M _ K$ whose conditional distributions match those of the source given the last $K$ symbols. He shows that $M_K$ is optimal in the sense that it uniquely minimizes the relative entropy rate between the true source and any $K$-th order Markov chain on the same alphabet. In other words, the family of finite-order Markov chains $\{M_K\}$ provides a sequence of best approximations to the stationary process in the relative-entropy-rate sense. Formally,
>
> \begin{align}
> H _ {p\|p^K} (\{X _ n\}) \\
> & = \inf _ {M _ K \in \mathcal{M} _ K} H _ {p\|M _ K}(\{X _ n\})  \\
> & = I(X _ 0;X _ {-\infty}^{-K-1}|X _ {-K}^{-1})
> \end{align}
>
> where $p$ is the true stationary source,
>
> $p^K$ is the canonical $K$-th order Markov approximation to $p$,
>
> $\mathcal{M}_K$ is the class of stationary $K$-th order Markov sources on the same alphabet,
>
> $H_{p\|q} (\{X_n\})$ is the relative entropy rate of $p$ with respect to $q$,
>
> $X_{-\infty}^{-K-1}=(X_{-∞}⋯,X_{-K-1})$ is the infinite past,
>
> $X_{-K}^{-1}=(X_{-K},⋯,X_{-1})$ is the block of the last $K$ symbols,
>
> and $I(⋅;⋅|⋅)$ conditional mutual information. Applying the above indentity with $K+1$ instead of $K$, we get
>
> \begin{align}
> H_{p\|p^{K+1}}(\{X_n\}) = I(X_0;X_{-∞}^{-K-2}|X_{-K-1}^{-1})
> \end{align}
>
> We are interested in the gain from going from order $K$ to $K+1$, so
> \begin{align}
> \Delta _ K&= H _ {p\|p^K} (\{X _ n\})-H _ {p\|p^{K+1}}(\{X _ n\}) \\
> & = I(X _ 0;X _ {-\infty}^{-K-1}|X _ {-K}^{-1}) - I(X _ 0;X _ {-∞}^{-K-2}|X _ {-K-1}^{-1}) \\
> & = I(X _ 0;X _ {-K-1}|X _ {-K}^{-1})
> \end{align}
>
> (2) Mapping to our discretized surprisal process
>
> In our setting, $\{X _ n\}$ is instantiated by the discretized surprisal process $\{a _ t\}$, where $a _ t$ corresponds to $X_0$, $X _ {-K}^{-1}$ corresponds to $a_{t-K}^{t-1}==(a_{t-K},\cdots, a _ {t-1})$, $X _ {-K-1}$ corresponds to $a _ {t-K-1}$. For the case $K=1$, the gain from first-order to second order is precisely $I(a _ t;a _ {t-2}|a _ {t-1})$. We directly estimate the relevant conditional mutual information term on our data. We first fit a first-order $\hat{P} _ 1(a _ t|a _ {t-1})$ and a second-order model $\hat{P} _ 2(a _ t|a _ {t-1},a _ {t-2})$ from transition counts on the reference set. We then compute plug-in estimates on test set
> \begin{align}
> \hat{H} _ 1 = -\frac{1}{n-1} \sum _ {t=2}^{n} \log _ 2 \hat{P} _ 1(a _ t|a _ {t-1}), \\
> \hat{H} _ 2 = -\frac{1}{n-2} \sum _ {t=3}^{n} \log _ 2 \hat{P} _ 2(a _ t|a _ {t-1},a _ {t-2})
> \end{align}
> Their difference is the plug-in estimate of the conditional mutual information
> \begin{align}
> \hat{I}& = \hat{H} _ 1 - \hat{H} _ 2\\
> & = I(a _ t;a _ {t-2}|a _ {t-1})
> \end{align}
>
> in bits per token, i.e., the extra predictive information contributed by the second-order context beyond the immediate past. On our data, we obtain empirical estimates of conditional mutual information and perplexity. In our experiments, $\hat I$ is at most 0.0076 bits/token, which corresponds to a perplexity reduction around 0.5%.  Thus, in terms of average predictive performance, the second-order Markov model brings only a sub-percent improvement over the first-order model. Combined with Gray's best Markov approximation theory, this indicates that a first-order Markov chain already captures most useful temporal dependence in the discretized surprisal dynamics, and provides a theoretically justified and empirically sufficient model for the discretized surprisal dynamics in our detector.
>
> | Source          | Order pair      | $\hat H_K$ (bits/token) | $\hat H_1 - \hat H_2$ (bits/token) | Perplexity | Rel. PP change vs 1st |
> |-----------------|-----------------|--------------------:|-------------------------:|-----------:|----------------------:|
> | GPT-5-chat      | 1st (baseline)  | 2.7882              | 0.0000                   | 6.9075     | 0.000%                |
> | GPT-5-chat      | 2nd order       | 2.7805              | 0.0076                   | 6.8711     | +0.528%               |
> | Human           | 1st (baseline)  | 2.8089              | 0.0000                   | 7.0074     | 0.000%                |
> | Human           | 2nd order       | 2.8043              | 0.0045                   | 6.9854     | +0.314%               |
>
>
>
>
>
> [1] Entropy and Information Theory, 2nd ed., Springer, 2011, Gray

---

### Official Review · Reviewer_gbAS · 2025-11-01

**Soundness:** 2
**Presentation:** 3
**Contribution:** 2
**Rating:** 4
**Confidence:** 4

**Summary:**

The paper proposes SurpShark, where next-token log-likelihoods of a text under a proxy model are collected (surprisals), quantized and modeled as Markov chains. By comparing the transition kernel between a reference collection of texts and a testing text, SurpShark effectively detects whether the testing text is machine-generated. The paper also provides theoretical justification for the method.

**Strengths:**

1. The presentation of the methodology of SurpMark (Section 3) is straigtforward and easy to understand, with good discussions and intuitions to drive the construction.
2. The main experiment results show significant improvement of SurpMark over existing methods.
3. Detailed and extensive ablations are performed to understand the effects of hyperparameter k, reference length, etc. on model performance, backing intuition behind the method and strongly supporting its effectiveness.

**Weaknesses:**

1. My biggest problem with the paper is the theoretical section, which in my current opinion greatly weakens the paper because it: a) is founded on a false assumption, b) does not provide theorems that go beyond what's readily intuitive, and c) is loosely connected to experiments. Unless there's some gross misunderstanding on my part, I cannot comfortably recommend acceptance given the current state of how the theory is presented, especially since the results are presented as a main contribution. See below for detailed justifications for these arguments:

    a) The theory section bases its analysis on a single false assumption: that there is an underlying Markov transition kernel for the surprisal sequences (Page 5, Line 216). Realistically, the surprisal sequence follows a partially observed Markov chain, where the underlying transition is powered by the ground truth generation process $P(x_n | [x_t]_{1}^{n-1})$, and a hidden state is the collection of texts up to a certain token. It is fairly unreasonable to assume surprisals can be simplified into a Markov process, since by themselves the surprisals hide significant semantic information. I personally would allocate SurpMark's superior performance to its first-order **approximation** of the underlying multi-order transition kernel, and that this approximation is sufficient in differentiating between texts generated from machines vs. humans. Related to this, the authors attempted to justify this first-order choice empirically by Figure 2(b) and the argument "The first-order Markov assumption suits our setting because LM predictions rely mainly on short range context (Khandelwal et al. (2018))." on Lines 177-178, but the figure has no experimental details attached as far as I can find (it's also very questionable why the AUROC would decrease as the order increases), and the quoted 2018 paper does not support first-order assumptions, rather that shallow contexts are more "sharply distinguished".

    b,c) For discussion's sake let's say here the first-order Markov assumptions are true. In Section 4.2, the theoretical suggestion for the choice of k has a constant term with unknown magnitude from both Proposition 4.1 and Theorem 4.2, reducing the reliability of this theoretical suggestion $k^*$. Just to be clear, I'm not arguing that the whole analysis is pointless; I think it provides a good connection between theory and empirics regardless, but I would suggest not framing the theorem's prediction as a good basis for empirical studies (as in Section 5.2), but a validation or sanity check that the intuitions behind the method is correct (supported by the optimal k values in Figure 4(a)). In Section 4.3, the main result is the asymptotic normality of $\Delta GJS_n$, which despite the technical difficulty is quite straightforward. It would be better if the exact convergence terms (namely $sigma_H^2$) can be analyzed and compared to empirical results. More importantly, in my view these technically impressive results unfortunately do not answer the most important question: can first-order Markov chain approximate the underlying partially observed Markov chain?

2. There are some other minor writing issues, see Questions 2-3.

**Questions:**

Q1. Can you explain how you obtained Figure 2(b) and why the performance gets worse with larger order?

Q2. The definition of the surprisal sequence goes from text t to token sequence x. To clarify, shouldn't x be the exact same as t? Saying the proxy model $F_{\theta}$ performs inference make it seem like x is obtained as a rollout.

Q3. The text in Lines 198-204 is a bit rough. The main issue is that $\Delta GJS_n$ is referenced before defined.

Q4. [**important**] Can you discuss the assumption of first-order Markov chain, both its theoretical and empirical implications?

Q5. For the optimal $k^\*$ determined through Section 4.2, how would you handle the constant C present in the theories when applying the suggested $k^\*$ to empirical studies?

Q6. What can be implied quantatively by Theorem 4.4 beyond a qualitive assessment of asymptotic normality?

---

> ### Author Response · Authors · 2025-11-20
> **Response to Question 1 to 3**
>
> We appreciate the reviewer for the invaluable suggestions to improve the quality of our work. We have addressed all concerns as below.
>
> **Q1: Explaination for Fig.2(b)** We apologize for not providing enough background on how Figure 2(b) was constructed. All AUROC values in Figure 2(b) are computed using exactly the same amount of reference data and test data. As in our main experiments, we use 300 human paragraphs and 300 machine-generated paragraphs, each with length about 100-200 tokens as reference data. We use 150 human paragraphs and 150 machine paragraphs as test data. Intuitively, increasing the Markov order makes the state space explode while the amount of reference data is fixed, so transition estimates become very sparse and noisy.
>
> The degradation with larger order is a variance / sparsity effect that arises both on the reference side and on the test side: (i) **State-space explosion**: A first-order chain with $k$ bins has $k^2$ transitions; an order-$l$ chain effectively has $k^{l+1}$ transitions. With only 300 human + 300 machine paragraphs of 100-200 tokens, many high-order contexts in the reference data are observed only a few times or not at all, so the estimated transitions become extremely noisy. (ii) **Short test sequences.** Each test paragraph is itself only 100-200 tokens long. Even if the reference transitions were perfectly estimated, an order-$l$ model on a 100-200-token sequence can observe only a very small number of distinct $l$-length contexts. The higher-order model is severely under-sampled on each individual test example.
>
> To isolate **the effect of reference sparsity**, in the XSum@GPT-J-6B dataset, we fixed $k=6$ and the test set (150 human + 150 machine paragraphs, 100-200 tokens each), and increased the reference size from 300 to 1200 paragraphs per side. AUROC for higher-order models improves only slightly and remains clearly below the first-order model.
>
> | Order \ Reference size | 300   | 600   | 900   | 1200  |
> |------------------------|-------|-------|-------|-------|
> | 1                      | 86.51 | 86.17 | 86.69 | 86.89 |
> | 2                      | 81.49 | 82.08 | 82.77 | 82.80 |
> | 3                      | 74.98 | 74.74 | 75.84 | 75.85 |
> | 4                      | 64.72 | 69.51 | 71.20 | 73.00 |
>
> To further evaluate **the effect of test sparsity**, in another WritingPrompts@Genmma-7B dataset, we vary the test passage length from 150 to 300 tokens while keeping the reference size fixed (300 passages per side, each with fixed 300 tokens). AUROC consistently increases for all orders, but the first-order model remains clearly best, and higher orders still lag behind by several points.
>
> | Order \ Test Length | 150   | 200   | 250   | 300   |
> |----------------|-------|-------|-------|-------|
> | 1              | 90.59 | 92.66 | 94.60 | 94.58 |
> | 2              | 89.42 | 90.67 | 92.21 | 92.60 |
> | 3              | 88.67 | 90.57 | 91.40 | 91.40 |
> | 4              | 62.29 | 65.59 | 68.46 | 68.33 |
>
> Taken together, these ablations reflect practical text-detection settings with limited reference data and short passages. In this regime, the first-order model offers the best bias-variance tradeoff, so we believe it is the most reasonable default choice.
>
> **Q2: Definition for t and x**
>
> In our paper, $t$ is a fixed passage from the corpus. We simply tokenize it with the proxy LM's tokenizer and run a single forward pass to obtain conditional probabilities along this given sequence. The symbol $x=(x_1,...x_n)$ in Section 3 is intended to denote the tokenization of $t$ under the proxy model, not a rollout produced by autoregressive sampling. We will revise the text to make clear that we are only scoring a fixed passage, not generating new tokens.
>
> **Q3: ΔGJS Referenced Before Defined** We will revise Lines 198-204 and improve the wording accordingly.

---

> ### Author Response · Authors · 2025-11-20
> **Response to Question 4 and Weakness 1**
>
> **Q4 & W1: Motivation, Role of theorems** We thank the reviewer for the detailed comments on the theory section. Our intent was not to use a first-oder markov model to present a fully realistic generative LM, but to analyze the behavior of our statistic (GJS) under a simple, empirically motivated approximation.
>
> **(a) On Markov Assumption** We agree that the current phrasing about first-order Markov assumption  (line 167, 177) is too strong and can be read as a false generative assumption. In the revision, we will make it explicit that we model the discretized surprisal state sequence as a first-order Markov chain for the purpose of analyzing the statistic $\Delta GJS$. That is, the Markov structure is a **modeling choice** on a derived summary statistic, **not a claim about the underlying generator**.
>
> Empirically, our use of a first-order chain is motivated by the observation that, with limited reference data, first-order models already give strong detection performance, while higher-order chains tend to hurt performance due to data sparsity. This is supported by Fig. 2(b) and experiments in response to Q1. In the revision, we will add the missing experimental details for Fig. 2(b) and briefly explain why AUROC can decrease with increasing order in this regime. We will also remove the sentence referencing Khandelwal et al. (2018) as it is not needed for our argument.
>
> **(b,c) Role of the theorems, the choice of k, and asymptotic normality** We agree that, due to unknown constants, the results about scaling of $k$ in Section 4.2 should be interpreted as guidance and sanity checks rather than precise prescriptions for the optimal $k$.  In the revision, we will rephrase Section 4.2 and its discussion in Section 5.2 to emphasize that the theory predicts a reasonable scaling.
>
> The one part of the theory that we believe does go beyond what is “readily intuitive” is Proposition 4.3 showing that ΔGJS can be interpreted as a log-likelihood ratio between a “shared source” and a “separate sources” hypothesis for the state sequences. We will highlight this more clearly as the main conceptual link to classical hypothesis-testing.
>
> Finally, on the broader question "can a first-order Markov chain approximate the underlying partially observed Markov chain?", we agree that this is an important question but beyond the scope of the current work. Our focus here is more modest: we work with a simple first-order Markov model for the discretized surprisal sequence, motivated by our empirical findings, and analyze how ΔGJS behaves in this setting. A general theory of when such approximations are valid would be an orthogonal line of work.
>
> We hope these clarifications and revisions address the reviewer's concerns about overclaiming and better align the theoretical section with its intended role in the paper.

---

> ### Author Response · Authors · 2025-11-20
> **Response to Question 5 & 6, Part of Weakness 1**
>
> **Q5: Handling the Constant C** Our intention in Section 4.2 was not to claim that Theorem 4.2 fully determines the empirical optimum $k$, but rather that it justifies the scaling law $k=\Theta(N^{1/5})$. In practice, for each dataset we treat the theorem as providing the functional form $k=CN^{1/5}$ and then select $k$ by a small grid search. To directly address the reviewer's question about the constant $C$, we examined the ratio $\frac{k}{N^{1/5}}$ across several reference size and found it to be consistently around 0.8. This suggests that in our regime the implicit constant is approximately $C ≈ 0.8$, and that the empirically chosen $k$ is well aligned with the theoretical scaling law.
>
>
> | Number of ref samples | $N$ (approximate total transitions) | Empirical best $k$ | $N^{1/5}$ | $\frac{k}{N^{1/5}}$    |
> |-------------------|------------------------------------|------------------|---------------------|------|
> | 100               | 15,000                             | 6                | 6.84                | 0.88 |
> | 300               | 45,000                             | 7                | 8.52                | 0.82 |
> | 400               | 60,000                             | 7                | 9.03                | 0.78 |
> | 600               | 90,000                             | 7                | 9.80                | 0.71 |
> | 900               | 135,000                            | 9                | 10.62               | 0.85 |
>
>
> **Q6 & W1: Quantitative Implications of Theorem 4.4**
> While the asymptotic variance in Theorem 4.4 does not provide a simple closed-form expression, Appendix A2.3.3 (line 1545-1604) along with Lemma A2.13 give an explicit numerical procedure to solve it. To quantitatively compare this theoretical variance with empirical fluctuations, we proceed as follows. We first compute the theoretical variance using the estimated Markov kernels from reference data. Then we estimate the empirical variance of $\Delta GJS$ in detection procedure. Table below reports theoretical variance and empirical variance with test length 250. Overall, the theoretical variance captures the right order of magnitude $\Delta GJS$ fluctuations, so we interpret it as a conservative asymptotic scale parameter rather than a precise finite-sample variance estimator.
>
> | Model        | Emp σ² (Human) | Emp σ² (LM) | Th σ² (Human) | Th σ² (LM) | Ratio Th/Emp (Human) | Ratio Th/Emp (LM) |
> |-------------|----------------|-------------|---------------|------------|-----------------------|-------------------|
> | Llama3-8B   | 1.15e-05       | 1.06e-05    | 2.48e-05      | 7.50e-05   | 2.16                  | 7.08              |
> | Llama3.2-3B | 1.12e-05       | 9.93e-06    | 2.73e-05      | 9.11e-05   | 2.44                  | 9.17              |
> | Gemma-7B    | 1.50e-06       | 5.96e-07    | 3.56e-06      | 2.43e-06   | 2.37                  | 4.08              |
>
> Finally, to assess the distributional shape, we ran Shapiro-Wilk tests on the obtained $\Delta GJS$ score, as shown in table below, the Shapiro-Wilk statistics are close to 1 and the p-values are not small (larger than 0.05). This indicates no evidence against normality and empirically supports the central-limit-theorem-based approximation in Theorem 4.4, consistent with the variance comparison above.
>
> | Setting          | SQuAD@GPT-5-chat | WritingPrompts@Llama3-8B | XSum@Qwen3-8B |
> |------------------|-----------------:|--------------------------:|--------------:|
> | **H1 (LM text)** stat    | 0.9952            | 0.9856                   | 0.9974       |
> | **H1 (LM text)** p-value | 0.9078            | 0.1203                   | 0.9969       |
> | **H0 (human text)** stat    | 0.9876            | 0.9854                   | 0.9929       |
> | **H0 (human text)** p-value | 0.2032            | 0.1143                   | 0.6632       |

---

> ### Author Response · Authors · 2025-11-26
> **First-order Markov chain is a theoretically justified and empirically sufficient approximation to the discretized surprisal process**
>
> Regarding your question “can a first-order Markov chain approximate the underlying partially observed Markov chain?”, we would like to point you to our more detailed discussion in our response to Reviewer ex8a (Round 2 Response (1/2)).
>
> Very briefly, we use Gray’s Markov approximation theory, which applies to any stationary discrete-time source (including observation sequences of partially observed Markov chains), to justify that our first-order chain is the best first-order Markov approximation to the discretized surprisal process in the relative-entropy-rate sense. Empirically, the gain we refer to is the reduction in Markov approximation error for the discretized surprisal process. The measured gain from moving from first order to second order is very small, indicating that a first-order model already captures almost all useful temporal dependence in our setting. Please refer to that response for the full derivation.
>
> [1] Entropy and Information Theory, 2nd ed., Springer, 2011, Gray

---

### Comment · Area_Chair_RPqs · 2025-11-21

Dear Reviewers,

We kindly encourage you to review and respond to the authors’ rebuttals. Your timely feedback is important for ensuring a fair and thorough review process. Thank you for your contributions to ICLR 2026.

AC

---

### Author Response · Authors · 2025-11-29
**Rebuttal Summary – Clarifications and Key Theoretical & Empirical Updates**

Dear Area Chair,

We would like to briefly summarize how our rebuttal addresses the main concerns behind the initial low scores.

Three reviewers (rzH5, ex8a, EQL7) explicitly recognize our framework as **novel**; Even Reviewer rzH5 (Score 2), describes the “core idea” as “**novel and interesting**.” The low scores are driven not by lack of novelty, but by concerns about (i) the theoretical setup, (ii) generator dependence, (iii) baselines, and (iv) robustness/statistical validation. Below we summarize how we have satisfactorily addressed the main concerns:

1. Addressing Reviewer rzH5 (Score 2): Resolving Misunderstandings and Generalization Concerns


**Clarifying the "White-Box" Misunderstanding (Critical)**: The reviewer’s main objection was based on a factual misunderstanding that our theory assumed a "white-box" setting that the generator's parameters are known and used in the detector. **We clarified that our method is strictly black-box and likelihood-free**: the Markov chains model the proxy LM's discretized surprisals, not the generator's internal mechanism . The reviewer has since acknowledged our responses were "very comprehensive".



**Resolving Generator Dependency (SurpMark-MC)**: The reviewer questioned the method's reliance on knowing the generator identity to build references . In Round 2 response, we addressed this by proposing SurpMark-MC, a multi-class variant using a shared quantizer and nearest-neighbor decision rule. Experiments show it generalizes effectively to unseen models without recomputing transitions, achieving performance better than or comparable to oracle baselines .


**Completing Baseline Comparisons**: To address the request for more SOTA baselines , we added comparisons with **Binoculars, FourierGPT, and R-Detect** (also in response to EQL7).

2. Addressing Reviewer gbAS, ex8a (Score 4): Justification for first-order modeling


**Theoretical Grounding (Gray’s Approximation Theory)**: In our Round 2 response to Reviewer ex8a, we introduced Gray’s Markov approximation theory. We utilized this theory to rigorously justify that the canonical first-order chain is the mathematically optimal approximation for the discretized surprisal process in the relative-entropy-rate sense.


**Empirical Validation (Sparsity & Information Gain)**: We provided new ablation studies showing that higher-order models (order $\ge 2$) **suffer from severe data sparsity and variance issues**, which degrade detection performance compared to the first-order model. Furthermore, we explicitly measured the **conditional mutual information gain from adding second-order context** and found it to be **negligible** ($< 0.0076$ bits/token), empirically confirming that a first-order model captures the necessary signal for text detection.



3. Addressing Reviewer gbAS, rzH5: Clarification for potential overclaiming regarding its implications

**Clarifying Scope (Addressing Overclaims)**: We revised the manuscript to explicitly frame the original theoretical results as an analysis of the decision statistic under an idealized model, rather than a claim about the true generative process of LLMs. This addresses the concern about potential overclaiming while preserving the interpretive value of the theory.

4. Addressing Reviewers ex8a: Robustness

**Adversarial Robustness**: Responding to Reviewer ex8a's concern about adversarial vulnerability, we tested against prompt-engineered adversarial attacks. SurpMark demonstrated superior resilience compared to baselines like Fast-DetectGPT.

5. Addressing Reviewers gbAS, EQL7: Statistical Verification for Asymptotic Normality

**Statistical Validation (Normality & Variance)**: We empirically validated Theorem (Asymptotic Normality of test statistics) (Theorem A3.14 in revision) through two rigorous checks:

- Shape: Shapiro-Wilk tests confirmed the score distribution is statistically indistinguishable from normal ($p > 0.05$).

- Scale: We calculated the theoretical asymptotic variance derived from our theorem and verified that it aligns with the empirical variance in order of magnitude ($10^{-5}$), confirming the theory provides a valid conservative estimate for finite samples.

**Current Status**

We have uploaded a revised PDF that incorporates these theoretical clarifications, new experiments (Gray’s theory–based analysis, SurpMark-MC, higher-order ablations), additional baselines, and robustness and normality checks.

Recent reviewer responses have been positive about both the novelty of the core idea (e.g., k-means-based surprisal quantization, Reviewer rzH5) and the rigor of the new experiments (Reviewer ex8a). Our latest Round 2 updates (Gray-based derivations and SurpMark-MC results) directly target the remaining theoretical and generalization concerns.

We hope this summary is helpful in your final assessment, and we are grateful for your time and consideration.

Thanks,
The Authors of Submission 7932

---

### Meta-Review · Area_Chair_hX93 · 2026-01-07

**Summary:**

This paper proposes SurpMark, a reference-based detector for machine-generated text that models token surprisal dynamics as first-order Markov chains and uses Generalized Jensen-Shannon divergence for classification. Reviewers acknowledge the novelty of the core idea—discretizing surprisals into states and comparing transition patterns—and the extensive empirical evaluation. However, significant concerns remain regarding the theoretical foundations. The first-order Markov assumption, while empirically motivated and now supplemented with Gray's approximation theory, still represents a substantial simplification whose adequacy for capturing the distinction between LLM and human text generation is not rigorously established. The authors provided comprehensive rebuttals including new ablations and baseline comparisons, yet the fundamental question of whether first-order Markov models can reliably approximate the underlying text generation processes remains insufficiently answered. Additionally, the method's reliance on generator-specific references limits practical deployment; while SurpMark-MC partially addresses this, performance on unseen generators shows notable variability. Given these unresolved concerns about theoretical grounding and generalization, I recommend rejection.

**Reviewer Concerns:**

Several concerns were adequately addressed through the rebuttal: the white-box versus black-box misunderstanding was clarified, additional baselines (Binoculars, FourierGPT, R-Detect) were included with SurpMark outperforming in most settings, k-means versus alternative binning ablations were provided, and statistical validation of asymptotic normality was conducted via Shapiro-Wilk tests. The adversarial robustness experiments demonstrated resilience compared to baselines.

Key concerns remain outstanding. The theoretical justification for first-order Markov modeling, despite the introduction of Gray's approximation theory and empirical measurements showing small conditional mutual information gain, does not establish that this approximation suffices for reliable detection across diverse generators and domains. The empirical observation that higher-order models perform worse due to sparsity is a practical limitation rather than a theoretical validation of first-order sufficiency. The generator-dependency issue was partially addressed by SurpMark-MC, but performance varies significantly across unseen generators (e.g., GPT-5-chat at 64.27 AUROC versus 83.56 with self-ref), raising concerns about real-world applicability where the generator is unknown.

**Reviewer Scores:**

Reviewer gbAS (Score 4): The authors clarified the scope of theoretical claims and provided additional justification, but the core concern about the Markov assumption's validity was not fully resolved. The score would likely remain at 4.

Reviewer ex8a (Score 4): The reviewer explicitly appreciated the comprehensive responses and rigorous experiments. Given the positive acknowledgment and addressed concerns regarding adversarial robustness, this reviewer might increase to 5.

Reviewer EQL7 (Score 6): Most concerns were satisfactorily addressed with ablations, baseline comparisons, and clarifications. The score would likely remain at 6.

Reviewer rzH5 (Score 2): Despite acknowledging the comprehensive responses and appreciating the k-means contribution, the reviewer maintained concerns about generator-dependency. With SurpMark-MC results provided, the score might increase to 3-4, though substantial reservations about practical applicability would remain.

---

### Decision · Program_Chairs · 2026-01-26

Reject